# Verbalized Bayesian Persuasion

**Wenhao Li** [1]  **Yue Lin** [2]  **Yun Hua** [3]  **Xiangfeng Wang** [4][5]  **Bo Jin** [1]  **Hongyuan Zha** [2]  **Baoxiang Wang** [2][6]

## Abstract

Information design (ID) explores how a sender influence the optimal behavior of receivers to achieve specific objectives. While ID originates from everyday human communication, existing game-theoretic and learning methods often model information structures as numbers, which limits many applications to toy games. This work leverages LLMs and proposes a verbalized framework in Bayesian persuasion (BP), which extends classic BP to real-world games involving human dialogues for the first time. We map the BP to a verbalized mediator-augmented game, where LLMs instantiate the sender and receiver. To efficiently solve the verbalized game, we propose a generalized equilibrium-finding algorithm combining LLM and game solver. The algorithm is reinforced with techniques including verbalized commitment assumptions, verbalized obedience constraints, and information obfuscation. Experiments in dialogue scenarios, such as recommendation letters, law enforcement, diplomacy with press, validate that our framework can reproduce theoretical results in classic BP and discover effective persuasion strategies in more complex natural language and multi-stage scenarios.

## 1. Introduction

Persuasion plays a significant role in modern economies, with estimates suggesting that up to one-quarter (McCloskey & Klamer, 1995), or even 30% (Antioch, 2013) of GDP, is persuasion. The study of BP has deep roots in economics,

with numerous applications across fields such as school grading (Boleslavsky & Cotton, 2015), law enforcement deployment (Lazear, 2006), research procurement (Yoder, 2022), matching platforms (Romanyuk & Smolin, 2019), and routing systems (Das et al., 2017). Various lines of theory have been proposed to explore the power of persuasion in different contexts (Kamenica, 2019).

This work investigates the Bayesian persuasion (BP) problem (Kamenica & Gentzkow, 2011; Kamenica, 2019) between two agents: a sender and a receiver. Unlike cheap talk (Lo et al., 2023), which is often employed in communication learning (Foerster et al., 2016; Sheng et al., 2022; Zhu et al., 2022), BP requires the sender to commit to an information disclosure mechanism publicly. The focus, therefore, is on rational (Bayesian) decision-makers who understand and optimally react to the disclosed information. Given a specific utility function, the BP problem is equivalent to finding an optimal Bayes-correlated equilibrium in an extensive-form game (Bergemann & Morris, 2013; 2019).

When the information space and the action space are both discrete and small, BP can often be solved analytically using optimization techniques (Kolotilin, 2018; Dworczak & Martini, 2019; Makris & Renou, 2023; Koessler & Skreta, 2023). This includes more complicated variants such as informed or multistage BP. Some work has also explored the use of multi-agent RL to approximate solutions (Wu et al., 2022; Lin et al., 2023; Bacchiocchi et al., 2024).

Despite these successes, most applications remain limited to games in the colloquial sense, where real-world complexity is often oversimplified. In fact, applying these methods to real-world settings requires constructing a model of the game in question, which involves defining the appropriate state space, action space, and transition dynamics. For instance, a classic example in BP is the recommendation letter problem, where a professor must write a letter that conveys nuanced information about a student's background (Dughmi, 2017). In the example, the student's quality is reduced to a binary classification of either weak or strong, and the professor's decision is restricted to either recommending or not. This abstraction strips away much of the meaningful information inherent in the actual task. Consequently, it prevents BP from being generalized to realistic problems.

This work aims to leverage game-theoretic methods, en-

---

[1]School of Computer Science and Technology, Tongji University, Shanghai, China [2]School of Data Science, The Chinese University of Hong Kong, Shenzhen, China [3]Antai College of Economics and Management, Shanghai Jiaotong University, Shanghai, China [4]Key Laboratory of Mathematics and Engineering Applications (MoE) and School of Mathematical Sciences, East China Normal University, Shanghai, China [5]Shenzhen Loop Area Institute (SLAI), Shenzhen, China [6]Vector Institute, Toronto, Canada. Correspondence to: Bo Jin <bjin@tongji.edu.cn>.

*Proceedings of the 43rd International Conference on Machine Learning*, Seoul, South Korea. PMLR 306, 2026. Copyright 2026 by the author(s).

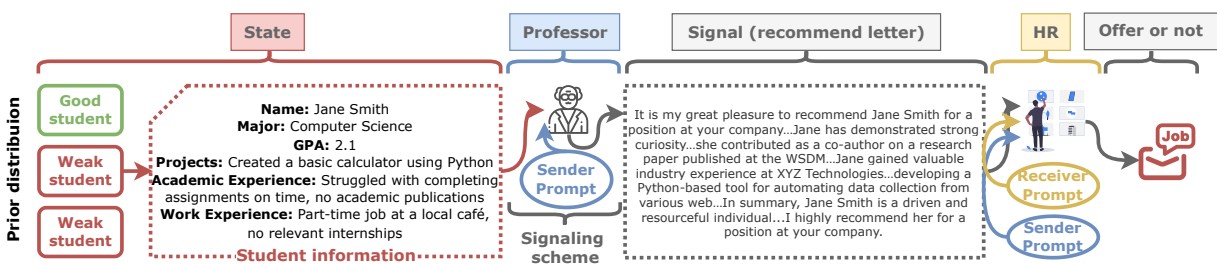

*Figure 1.* Solving the original BP problem directly in the natural language domain.

hanced by large language model (LLMs) (Zhao et al., 2023), to solve the original BP problem directly in the natural language domain. Specifically, we model the BP problem as a verbalized mediator-augmented, extensive-form game (Zhang & Sandholm, 2022), where states and actions (or signals for the sender) are all represented as text, as shown in Figure 1. For example, in the recommendation letter problem, the sender (professor) has a state that is a detailed, verbal description of the student's background, the signal is the content of the recommendation letter, while the reward remains numerical. The receiver (HR) observes this letter and must decide whether to accept the student. To enable the sender and receiver to process, understand, and generate this text, we parameterize both agents using LLMs.

Before introducing the proposed verbalized game solver, we need to address two fundamental challenges that have hindered the application of BP to real-world scenarios. The first challenge lies in the theoretical foundation of signal space design. Unlike traditional BP problems where signals are discrete and finite, real-world persuasion involves natural language that carries nuanced information and contextual dependencies, creating an infinite-dimensional signal space where classic BP solution concepts break down (Myerson, 1979; Kamenica & Gentzkow, 2011). The second challenge concerns the fundamental difficulty in strategy optimization when using LLMs as agents. While LLMs provide a powerful way to process natural language, their parameter spaces are extremely high-dimensional and non-convex, making it theoretically impossible to guarantee the existence of Nash equilibria in such spaces (Gemp et al., 2024) and computationally intractable for direct optimization.

To address these challenges, we draw on the prompt-space response oracle (Prompt-PSRO) (Gemp et al., 2024), which models strategy optimization for the sender and receiver as prompt optimization for their respective LLMs. This approach not only mitigates the challenge of optimization inefficiency but also reduces the action space from lengthy, complex text to compact, low-dimensional, discrete prompts. For instance, by adjusting the prompt given to the sender's LLM, we can control the "level of details in the student's background description" in the recommendation letter.

Building on the Prompt-PSRO, our solver is composed of several components that improve its performance, efficiency, and stability. These include verbalizing commitment assumptions, obedience constraints, and information obfuscation. Technically, we extend Prompt-PSRO to multistage games by proposing conditional prompt optimization and providing a convergence guarantee to the equilibrium solution. Together, these components form a comprehensive verbalized game solver tailored for BP problems, which we refer to as verbalized Bayesian persuasion (VBP). To the best of our knowledge, VBP is the first general framework that attempts to solve real, non-abstract BP problems.

Our main contributions include: 1) Transforming real-world BP problems into verbalized mediator-augmented, extensive-form games, which provides a unified interface for game-theoretic solvers; 2) Proposing a general game-theoretic solver for verbalized BP problems based on the Prompt-PSRO framework, with a convergence guarantee to equilibrium solutions. Several components including verbalized commitment assumptions and obedience constraints, information obfuscation, and conditional prompt optimization enhance the solver's performance, efficiency, and stability; 3) Reproducing results in classic and complex BP problems consistent with existing optimization and learning methods, while efficiently solving natural language and multistage BP problems. This potentially opens up a new line of studies of persuasion in real-world scenarios, which is helpful in understanding the interaction of multiple agents in both economic and societal applications.

## 2. Preliminaries and Related Works

**BP.** The canonical BP model is structured as follows (Kamenica, 2019). A receiver, an agent, has a utility function $u_1(a, \omega)$, which depends on its action $a \in \mathcal{A}$ and the state of the world $\omega \in \Omega$. Another agent, the sender (also known as the information designer), has a utility function $u_0(a, \omega)$. Both the sender and receiver share a common prior $\mu_0$ over $\Omega$. The sender's key decision is the choice of a signaling scheme, which is a mapping from the state to a distribution over signals, $\pi : \Omega \to \Delta(\mathcal{S})$. Here $\mathcal{S}$ denotes the signal set. By the revelation principle, it is without loss of generality

sufficient to consider $|\mathcal{S}| \leq \min\{|\mathcal{A}|, |\Omega|\}$.

Given knowledge of $\pi$ (i.e., under the commitment assumption (Kamenica & Gentzkow, 2011)), the receiver updates its belief from the prior $\mu_0$ to the posterior $\mu_\pi(\omega \mid s)$ using Bayes' rule. The receiver then selects an action $a^*$ that maximizes $\mathbb{E}_{\omega \sim \mu_\pi(\cdot|s)} u_1(a, \omega)$. Given this response mechanism from the receiver, the sender's objective is to solve the following maximization problem: $\max_{\pi \in \Pi} \mathbb{E}_{\omega \sim \mu_0} \mathbb{E}_{s \sim \pi(\omega)} u_0(a^*, \omega)$, where $\Pi$ denote the set of all possible signaling schemes. Here the revelation principle applies that an optimal signaling scheme exists that requires no more signals than there are actions available to the receiver. From the receiver's perspective, as long as it believes that the recommended actions are optimal according to its posterior belief, it will follow the sender's advice. These constraints on the sender's signaling scheme are referred to as obedience constraints (Myerson, 1979; Kamenica & Gentzkow, 2011). Then, BP can be reduced to a simplified linear programming (Dughmi & Xu, 2016; Dughmi, 2017; Lin et al., 2023),

$$\max_\pi \mathbb{E}_\pi \left[ u_0(a, w) \right], \text{ s.t. } \sum_w P(w) \cdot \pi(a \mid w)$$
$$\cdot \left[ u_1(a, w) - u_1(a', w) \right] \geq 0, \forall a, a'. \tag{1}$$

**Learning Methods for BP.** The problem of BP can be approximately solved by multi-agent reinforcement learning (MARL). In mixed-motive MARL, agents aim to advance their interests by shaping others (Leibo et al., 2017; McKee et al., 2020; Dafoe et al., 2020; Leibo et al., 2021). Existing methods achieve this through either mechanism (modifying rewards) (Yang et al., 2020; Zheng et al., 2022; Hua et al., 2023; Wang et al., 2024) or information design (modifying observations) (Wu et al., 2022; Bernasconi et al., 2022; Lin et al., 2023), the latter of which could be used to solve BP. An sender can commit to a strategy for providing state information to the agents, effectively altering the observation function of the receiver. So the sender can influence agents' behavior by strategically providing information and guiding them toward desired outcomes (Bergemann & Morris, 2019). In a long-term interaction, the sender and the receiver become aware of each other's strategy, which creates the game dynamics that resemble the commitment assumption and therefore the BP problem setting. Then, the outcome of the MARL algorithm becomes an equilibrium of BP.

**PSRO.** While game theory offers a framework to study interactions among agents (Bighashdel et al., 2024), classical analysis struggles with scalability due to the sheer number of strategies. Thus, many learning methods have been applied to large-scale games, with MARL (Yang et al., 2025; Zhang et al., 2021) among the most prominent. Learning methods do not require full game representation and instead create agents that explore and adapt by interacting with the environment. However, learning methods face inherent chal-

lenges, e.g., non-stationarity (Tuyls & Weiss, 2012) and non-transitivity (Sanjaya et al., 2022).

PSRO (Lanctot et al., 2017) emerged as a hybrid approach, combining traditional equilibrium computation with learning techniques. It improves scalability by focusing on relevant subsets of strategies (Wellman, 2006; Bighashdel et al., 2024). Approximate local Nash equilibria can be tractably obtained through PSRO-like approaches (Assos et al., 2023). As illustrated in Figure 2, PSRO begin with an initial set of strategies and proceed through 2 alternating steps. First, a normal-form meta-game is constructed, where each agent selects a meta-strategy to represent their overall behavior. A meta-solver (e.g., Nash equilibrium solver) then computes a solution (e.g., Nash equilibrium) for this meta-game. Second, each agent computes an approximate best response to the meta-strategy. This process repeats until no agent benefits from strategy deviation (Bighashdel et al., 2024). The prompt-space response oracle (Gemp et al., 2024) (Prompt-PSRO) is a verbalized adaptation of the standard PSRO, where strategies are parameterized by LLMs and represented as prompts. The approximate best response is generated by optimizing and sampling prompt strings, as opposed to the standard PSRO protocol where best responses are typically computed using MARL or gradient-based optimization.

**Game-Theoretic Solvers with LLMs.** The combination of a game-theoretic solver with prompt optimization, which we use in this work, is not the only paradigm for utilizing LLMs to solve games. Widely adopted parameter-efficient fine-tuning (Xu et al., 2023; Han et al., 2024), agentic workflow (Mao et al., 2025; Hua et al., 2024; Guo et al., 2024; Fan et al., 2024; Lorè & Heydari, 2024; Duan et al., 2024), as well as the recent trend of improving reasoning and problem-solving capabilities for complex and mathematical problems by having LLMs generate longer chains of thought prior to making decisions (Zelikman et al., 2022; 2024; OpenAI, 2024; Guo et al., 2025), are also very promising directions. The former allows for more fine-grained control of LLM outputs through in-weight updates, compared to in-context updates like prompt optimization, while the latter two may enable LLMs to discover novel game solvers. VBP is orthogonal to these approaches. Its primary goal is to leverage the rich foundation of game theory by incorporating various game-theoretic solvers that have already been proposed, and to extend the solid theoretical results established in classical games for solving verbalized games.

## 3. The Game Formulation for BP

To establish convergence for the VBP framework, we transform the classic BP problem into a special class of extensive-form games (EFGs), known as mediator-augmented games (MAGs, (Zhang & Sandholm, 2022)). In this section, we reformulate the BP problem in the form of an MAG. At a

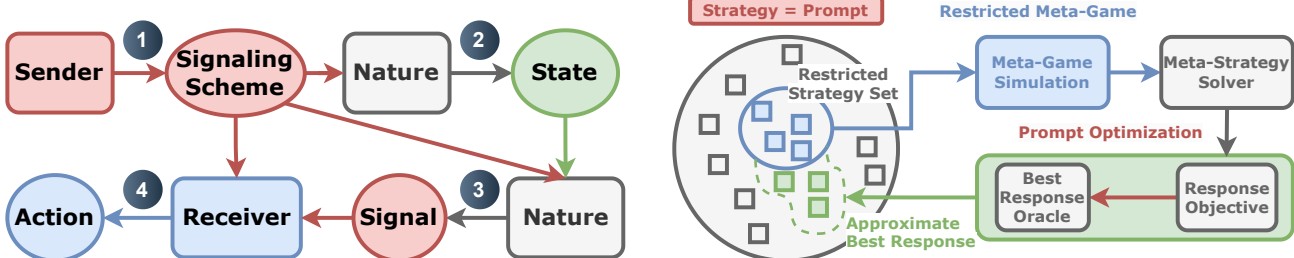

*Figure 2.* **Left:** BP timing in EFG; **Right:** Prompt-PSRO.

high level, a mediator-augmented game introduces an additional player, the mediator, who exchanges messages with the players and provides action recommendations.

**Definition 3.1.** A BP problem, represented as a mediator-augmented game $\Gamma$, consists of the following components (Zhang & Sandholm, 2022): **(1)** a player, referred to as the receiver, denoted by the integer 1; **(2)** a directed tree $H$ of histories or nodes, with the root denoted by $\varnothing$. The edges of $H$ are labeled with actions, and the set of legal actions at each node $h$ is denoted by $A_h$. Terminal nodes of $H$ are called leaves, and the set of such leaves is denoted by $Z$; **(3)** a partition of non-terminal nodes $H \backslash Z$ into $H_{\mathbf{C}} \sqcup H_0 \sqcup H_1$, where $H_1$ represents the nodes where player 1 acts, and $\mathbf{C}$ and $0$ represent chance and the mediator (i.e., the sender), respectively; **(4)** for each agent $i \in \{1, 0\}$, a partition $\mathcal{I}_i$ of the decision nodes $H_i$ into information sets. Every node in a given information set $I$ must have the same set of legal actions, denoted by $A_I$; **(5)** for $i \in \{1, 0\}$, a utility[1] function $u_i : Z \to \mathbb{R}$; and **(6)** for each chance node $h \in H_{\mathbf{C}}$, a fixed probability distribution $c(\cdot \mid h)$ over $A_h$.

At any node $h \in H$, the sequence $\sigma_i(h)$ for agent $i$ consists of all information sets (infosets) encountered by $i$, along with the actions taken at those infosets on the path from $\varnothing$ to $h$, excluding $h$ itself. An agent has perfect recall if $\sigma_i(h) = \sigma_i(h')$ for all $h, h'$ within the same infoset. A pure strategy for agent $i$ specifies one action from $A_I$ for each information set $I \in \mathcal{I}_i$. A mixed strategy is a probability distribution over pure strategies, and the sequence form of a mixed strategy corresponds to the convex combination of pure strategies. Let $X_1$ and $X_0$ denote the polytope of sequence-form mixed strategies $\boldsymbol{x}_1$ for the receiver and $\pi$ for the mediator, respectively. For a fixed $\pi \in X_0$, we say that $(\pi, \boldsymbol{x}_1)$ is an equilibrium of $\Gamma$ if $\boldsymbol{x}_1$ is a best response to $\pi$, meaning $\max_{\boldsymbol{x}_1' \in X_1} u_1(\pi, \boldsymbol{x}_1') \leq u_1(\pi, \boldsymbol{x}_1)$. We do not require the mediator's strategy (signaling scheme) $\pi$ to be a best response; hence, the mediator can commit to its strategy. The objective of this paper is to find an optimal Stackelberg equilibrium, which is an equilibrium $(\pi, \boldsymbol{x}_1)$ that maximizes the mediator's utility $u_0(\pi, \boldsymbol{x}_1)$. When viewed as an extensive-form game (EFG), the sequence of events in

BP is illustrated in Figure 2.

## 4. VBP Solver

This section will provide a detailed introduction to the VBP framework. The first part presents the verbalization of MAG, including a polarized setting (e.g. strong/weak students) that reproduces the classic examples in BP, and general settings for one-step and multi-stage BP in the language space. The second part introduces how this MAG is solved within the Prompt-PSRO framework, with the help of available best response approximations in language models (Yang et al., 2024; Romera-Paredes et al., 2024). The verbalized MAG, along with the three problem settings and the Prompt-PSRO-based game-theoretic solver, collectively constitute the VBP framework. We summarize the procedure in Algorithm 1.

**Verbalized Formulation for BP.** In order to leverage the wealth of research in LLMs for BP in realistic scenarios, we must abstract and map components of BP to the symbolic language. Note the mapping can be chosen is not unique. **State $\omega$:** Unlike the classic BP, which only describes the state with binary values, the state in VBP is defined as the text. For example, it is the detailed description to the student's quality in the recommendation letter problem. **Infosets $\mathcal{I}$:** The infoset is available only in the multi-stage setting. It is defined as the interaction history between the two parties. Specifically, this includes the signals sent by the sender, the receiver's decisions (public information), and their respective rewards (private information) from each round of interaction, plus the previous environmental states (private information). **Action $A$:** Agents' actions refer to the signaling scheme for the sender and the action for the receiver. As we transform the strategy optimization problem into a prompt optimization problem through the Prompt-PSRO framework, the actions involve selecting prompts. **Terminal states $\mathcal{Z}$:** In static settings the game terminates in one step. In multi-stage settings terminal states are determined by either a limit or the allowable tree depth.

In addition to the basic components of the game, the BP problem also includes two fundamental constraints that need to be mapped into the verbalized MAG.

---

[1] In this paper, we do not distinguish between utility and reward.

## 4.1. Verbalized Committment Assumption

The key difference between BP and cheap talk (Crawford & Sobel, 1982) lies in the presence of the commitment assumption, where the sender commits their signaling scheme as common knowledge. In VBP, this commitment is implemented at the prompt layer by (i) making the sender's "style prompt" public and (ii) conditioning the receiver's decision prompt on the same public description.

**What is committed.** A sender strategy in Prompt-PSRO is a discrete prompt component (e.g., a category–value writing style pair) that induces a distribution over natural-language signals. The public commitment in an episode is the exact style string (and, when applicable, the meta-strategy distribution over style strings) that the solver has selected.

**How the receiver observes the signaling scheme.** At decision time, the receiver prompt includes (a) the declared style string used by the sender for this episode and (b) the current Prompt-PSRO meta-strategy weights over the sender's prompt library (an explicit probability table). This makes the receiver's information set contain the same "common knowledge" object that is assumed in classic BP. The receiver is instructed to treat this distribution as the sender's committed signaling scheme and to update beliefs accordingly (Appendix E.6).

**Adherence verification (distinguishing from cheap talk).** Because commitment is not enforced by an external mechanism, we add a lightweight verifier that checks whether the realized signal text is consistent with the declared style. Concretely, we prompt a separate judge LLM (decoded at temperature 0) to classify the sender's output as *adherent* or *non-adherent* to the declared style; non-adherence triggers either (i) a penalty subtracted from the sender's rollout return during best-response evaluation or (ii) a regenerate-until-adherent step with a fixed retry budget. This verification step operationalizes commitment beyond cheap talk by making deviations from the publicly declared scheme costly in the optimization loop. In multi-stage settings, the receiver's infoset additionally contains the declared styles from prior rounds (and their realized outcomes), so commitment and reputation effects can accumulate over history.

## 4.2. Verbalized Obedience Constraint

The optimization in BP involves an (extended) obedience constraint (Myerson, 1979; Kamenica & Gentzkow, 2011; Lin et al., 2023), as shown in Equation (1) (or 4). We enforce this constraint softly by augmenting the best-response objective with an obedience penalty, analogous to reward shaping (Ng et al., 1999; Gupta et al., 2022).

**Sampling-based estimate.** Computing the full obedience violation requires summing over the entire state and action space. Instead, during evaluation of a candidate prompt, we form a MC estimate from the same rollouts used to estimate the agent's expected utility. For each visited state $\omega_t$ in a rollout, let $a_t$ be the receiver's realized action under its current prompt. We then sample a single counterfactual deviation action $a_t'$ (1-sample estimate) and compute an instantaneous violation $v_t := \max\{0,\ u_1(a_t', \omega_t) - u_1(a_t, \omega_t)\}$. The obedience penalty is the rollout-average $\widehat{V}_{\text{obed}} := \frac{1}{T} \sum_{t=1}^{T} v_t$, optionally averaged across multiple rollouts when the evaluator batches episodes.

**Theory-of-mind (ToM) deviation action.** To select $a_t'$, we use a ToM-style anticipation query (Rabinowitz et al., 2018; Albrecht & Stone, 2018) that asks an LLM to predict the receiver's most profitable one-step deviation at $(\omega_t, a_t)$. Concretely, we reuse the receiver's decision context (same task description, priors, and observed signal) and replace the final instruction with: *"List the single alternative action $a'$ (different from the recommended/realized action) that would maximize your utility, and output only that action."* We decode greedily (temperature 0) to reduce variance.

**Integration into best-response oracles.** For a candidate sender prompt (or prompt function) $\pi$, the scalar score optimized by OPRO/FunSearch is

$$J(\pi) = \mathbb{E}\left[u_0\right] - \beta\,\widehat{V}_{\text{obed}}, \tag{2}$$

where $\beta \geq 0$ is a tunable weight. Thus, the obedience mechanism enters the pipeline only through the evaluator: OPRO/FunSearch proposes a candidate prompt, we run rollouts to estimate $\mathbb{E}[u_0]$ and $\widehat{V}_{\text{obed}}$, and we return $J(\pi)$ as the objective value. (See Appendix E.3 for how this evaluator score is used inside Prompt-PSRO; key prompt scaffolds are in Appendix E.6.)

## 4.3. Four Settings in VBP

**Setting S1 & S2: Polarized signals and Static BP.** Polarized signals refer to the constraint to produce more straightforward signals, such as recommend v.s. not recommend in the recommendation letter example. The goal of this setting is to aligning the signal space with the classic BP formulation. We aim to reproduce the Stackelberg equilibrium in classic BP examples and validate the effectiveness of VBP. Specifically, we use the pretrained and aligned LLM to score the signals output by the sender, For example, this determines the degree, a real value between $0$ and $1$, to which the recommendation letter supports the student. Similar prompts can be designed for other problems. Then this value is pushed to an extreme point in the signal space, based on the minimum distance. By removing signal polarization, the signals are maintained in the language space. This constitutes the default setting of VBP.

**Setting S3: VBP in Multistage BP.** This setting considers a multistage scenario, which is very challenging for traditional methods (Gan et al., 2022; Wu et al., 2022). The agents

engage in multiple rounds of interaction, and the sender's historical signals serve as the basis for the receiver's subsequent decisions. This largely increases the complexity, as the sender cannot arbitrarily exploit the receiver with information advantage. Instead, they must consider current actions' impact on future rewards.

**Setting S4: VBP in Large-Scale Multi-Receiver BP.** We extend VBP to a large scale multi-receiver testbed by instantiating it in full press Diplomacy (Duffy et al., 2025) with ordinary negotiation and standard adjudication. Each episode designates a single sender and a set of receivers, all observing common natural-language communications and public commitments.

### 4.4. Verbalized Game Solver

After modeling the BP as a verbalized MAG, we parameterize both agents using pretrained and aligned LLMs and optimize their strategies with the Prompt-PSRO, thereby forming a general BP solver. We first present the following proposition based on the theoretical foundation (Zhang & Sandholm, 2022), with the proof provided in Appendix B.

**Proposition 4.1.** *VBP returns an $\varepsilon$-approximate Bayes correlated equilibrium in static BP and an $\varepsilon$-approximate Bayes-Nash equilibrium in multistage BP.*

In simple terms, the reason we can leverage the theoretical results of MAG is because different assumptions on the power of the mediator and the players' strategy sets induce different equilibrium concepts. The concept of Bayes correlated equilibrium (Bergemann & Morris, 2016) in static BP and Bayes-Nash equilibrium (Makris & Renou, 2023) in multistage BP is equivalent to the situation in the MAG where the mediator has an informational advantage, cannot lie (commitment assumption), and gains perfect recall under the extensive-form correlated equilibrium.

VBP does not directly solve the verbalized MAG using the Prompt-PSRO. Instead, we make targeted improvements to Prompt-PSRO for different settings, as illustrated in the Appendix E.3. For the S1, S2 and S4 settings, we optimize the strategies of the sender and receiver using Algorithm 4 from the Prompt-PSRO framework (Gemp et al., 2024), specifically the "categorical" approximate best response. Unlike in the original PSRO paper, we use the OPRO method (Yang et al., 2024) to generate both the categories and the specific content within the categories simultaneously.

S3[2] presents a challenge for the Prompt-PSRO. Existing method is unconditional or episode-wise, meaning that the

---

[2]Although S4 is also a multi-stage setting and, compared with S3, is even more complex—featuring large-scale and multi-receiver characteristics—we do not apply conditional prompt optimization to S4 in this version due to cost considerations. Empirically, the categorical approach still achieves strong performance.

prompt generated at the beginning of each episode is used for every subsequent timestep. In multistage BP, this significantly restricts the optimizable strategy space. In other words, both the sender and receiver can dynamically adjust their strategies based on the interaction history to achieve higher rewards. For example, the sender might honestly provide true information to the receiver early to build trust, then deceive the receiver later. Similarly, the receiver could bargain to extract more information. Thus, we propose a conditional version of Prompt-PSRO, denoted as step-wise Prompt-PSRO, building on the original framework.

Specifically, we use FunSearch (Romera-Paredes et al., 2024), where the strategy to be optimized is no longer a single prompt string, but an executable prompt-generating function $f$. Following the standard FunSearch setup, we represent $f$ as Python code constrained to a fixed template (Appendix E.6 gives concrete examples), and the LLM proposes code edits that are executed and scored by rollouts. The input to $f$ is a structured interaction history $h_{1:t}$, encoded as a list of per-round records (e.g., *(state_sum, sender_signal, receiver_action, r_0, r_1)*), where *state_sum* is the private state text (for the corresponding player) and the other fields are public observations from prior rounds. Given $h_{1:t}$, $f(h_{1:t})$ outputs a discrete prompt object (e.g., a category–value pair, or a small dictionary of prompt slots) used at step $t+1$ to condition the LLM policy. This makes the strategy history-dependent while keeping the optimizable object compact and executable, thereby enabling conditional prompts in S3.

Moreover, since we use aligned LLMs, the sender struggles to output strategic signals, such as hiding or obfuscating relevant information about the true state. We therefore introduce an *information obfuscation* mechanism. Concretely, after the sender produces a signal $s$ (text), we query a separate pretrained and aligned evaluator LLM (decoded at temperature 0) with access to (i) the sender's private state text $\omega$ (or a short state summary used by the environment) and (ii) the generated signal $s$. The evaluator is instructed with a fixed rubric to output a scalar obfuscation score $\widehat{O}(\omega, s) \in [0, 1]$ as $\widehat{O} := \frac{1}{4} \sum_{k=1}^{4} o_k$, where each $o_k \in \{0, \frac{1}{3}, \frac{2}{3}, 1\}$ is a rubric sub-score: (1) *Omission*: how much salient state information is missing; (2) *Distortion*: presence of statements contradicting $\omega$; (3) *Ambiguation*: use of hedges/vagueness that increases uncertainty while avoiding falsifiable claims; and (4) *Misdirection*: emphasis on irrelevant details to distract from key evidence. We implement the evaluator prompt as a strict JSON-format instruction ("return only JSON") and include two few-shot examples (one low-obfuscation and one high-obfuscation) per environment to stabilize scoring; see Appendix E.6 for the exact templates. The score $\widehat{O}$ is then used as feedback in $K$ rounds of self-reflection (Huang et al., 2023; Shinn et al., 2024; Tao et al., 2024): if $\widehat{O} < \tau$ (we use $\tau = 0.5$), we ask the sender to revise the signal to better conceal or reframe

state-revealing facts while preserving coherence; otherwise we accept the signal and enter the Prompt-PSRO loop.

**Remark.** While our work and prior research (Bai et al., 2024) both leverage BP, they address different problems. Model alignment applications optimize signaling between models to improve downstream performance (Bai et al., 2024), whereas we extend BP into natural language settings through verbalized frameworks and prompt optimization. Similarly, our approach differs from persuasive dialogue research that uses planning through fine-tuning or tree-search (Yu et al., 2023; Deng et al., 2024) or strategy annotation in negotiation contexts (He et al., 2018; Wang et al., 2019). Our key contribution uniquely merges verbalized BP with game-theoretic equilibrium solvers, enabling formal Bayes correlated equilibria reasoning in natural language contexts.

---

**Algorithm 1** Verbalized Bayesian Persuasion

---

**Input:** $C$, where $C_i$ is the initial prompt action set (i.e., one category and one corresponding content) for player $i$ (either the sender or receiver)

**Input:** $h$, containing hyperparameters for the approximate best response operator BR (e.g., **OPRO or FunSearch**)

1: **Initialize with LLM-based sampling:** Compute the expected payoff tensor $P$ over all joint actions in $C$ using Equation (1) or (4)

2: **Set:** $\pi \leftarrow$ uniform meta-strategy profile over $C$

3: **Set:** incomplete $\leftarrow$ **TRUE**

4: **while** incomplete **do**

5:     **for** player $i \in [N]$ **do LLM input:** provide current meta-strategy $\pi$ and action sets $C$; **best response:** $c_i \leftarrow \mathrm{BR}(i, \pi, h)$; **LLM output:** $c_i$

6:     **if** $c_i \in C_i \ \forall i \in [N]$ **then** incomplete $\leftarrow$ **FALSE else** $C_i \leftarrow C_i \cup c_i, \forall i \in [N]$; **recompute:** compute payoff tensor $P$ over updated $C$ via Eq. (1) or (4); **update:** $\pi \leftarrow$ meta-strategy w.r.t. $P$ **end if**

7: **Return:** $(\pi, C, P)$ {Return the final meta-strategy, action sets, and payoff tensor}

---

# 5. Experiments

We use 3 classic and 1 complex BP problem (detailed in Appendix C and D). In recommendation letter (REL) (Dughmi, 2017), a professor writes recommendation letters for students, which HR uses to decide on hiring. In courtroom (COR) (Kamenica & Gentzkow, 2011), a prosecutor tries to convince a judge to convict a defendant, with the prior belief that the defendant is guilty. In law enforcement (LAE) (Kamenica, 2019), drivers decide whether to speed or obey the law on a road with $Z$ miles, where $G$ police officers patrol. In Diplomacy with press (DIP) (Duffy et al., 2025), players control nations that can both move units and send messages; a persuader designs signals to influence others' beliefs about intentions and positions, aiming to steer their coordinated

actions. We defer the experiments for Setting S2 to Appendix F.1. Implementation details and hyperparameters are deferred to Appendix E. Additional results and analyses are provided in the Appendix (Appendix F; qualitative signals in Appendix F.9; prompt-function examples in Appendix F.10; further discussion in Appendix F.8; unaligned LLM analysis in Appendix F.8.4; extended obedience constraints in Appendix E.4; and additional applications in Appendix C.2).

## 5.1. VBP in Static Games (S1)

We first evaluate the effectiveness of the VBP under S1. Two baseline methods are chosen: BCE and MARL. The former relies on optimal equilibria computation (Lin et al., 2023; Kamenica & Gentzkow, 2011; Kamenica, 2019), while the latter employs multi-agent reinforcement learning for BP problems (Lin et al., 2023). As shown in Figure 3, the VBP framework successfully captures the essence of solving BP problems, namely, selectively withholding, obfuscating, or even deceiving about the true state, while also learning when to fully disclose accurate information.

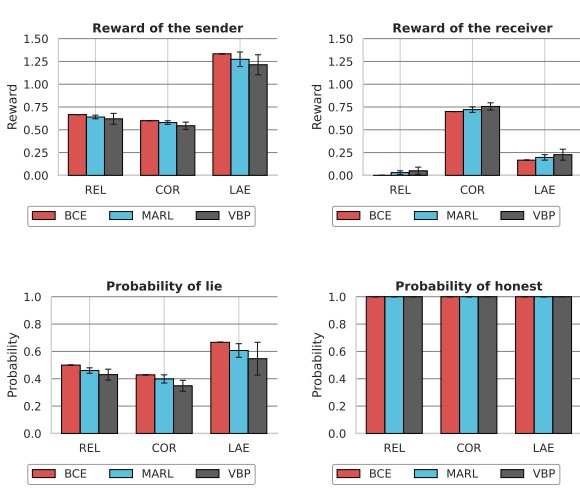

*Figure 3.* Performance comparison on classic static BP problems. Averaged over 20 seeds. In the 3 BP problems, the probability of lying refers to describing a weak student as strong, an innocent defendant as guilty, or an unpatrolled segment as patrolled. Conversely, the probability of honesty refers to accurately describing a strong student, a guilty defendant, or a patrolled segment.

## 5.2. VBP in Multistage Games (S3)

We also tested the effectiveness of VBP in a multistage scenario. Notably, the multistage BP differs from most of the literature. In existing works, the same sender was interacting with a new, short-sighted receiver in each round. In this work, the receiver remains the same and can perceive the interaction history, aligning more closely with the Markov signaling game (Lin et al., 2023). Since a closed-form solution for equilibrium cannot be computed, we record the average performance at each stage in Figure 4.

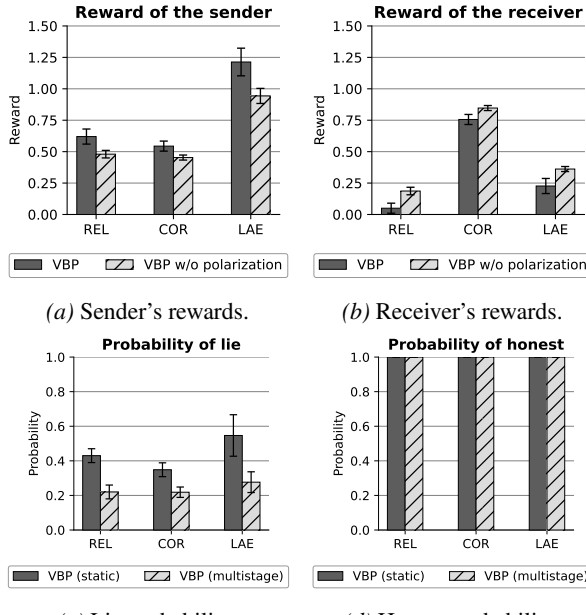

*(a)* Sender's rewards.  *(b)* Receiver's rewards.

*(c)* Lie probability.  *(d)* Honest probability.

*Figure 4.* Performance comparison on the S3 setting. Averaged over 20 seeds and 5 timesteps. The physical meaning of the probabilities of lying and honesty is consistent with Figure 3.

VBP's performance shows a noticeable decline compared to S2, while it still manages to learn both appropriate deception and honesty. We also visualize the changes in the sender's deception and honesty probabilities during training, as shown in Figure 8. As shown in the figure, since the receiver can perceive the history, the sender's deceptive behavior goes through several oscillations, reflecting a kind of bargaining dynamic (Nash et al., 1950; Nash, 1953; Maschler et al., 2013): The sender is initially leaning towards honesty, then discovering that deception maximizes gains, and later realizing that excessive deception triggers retaliation from the receiver, eventually converging to a relatively low deception probability. Likewise, with the receiver having access to historical interactions, the sender demonstrated an upward trend in honest behavior compared to the S2 setting, with truthfulness levels progressively increasing throughout. We provide a deeper discussion of the bargaining-like dynamics in the S3 setting in Appendix F.8.

We also test using different LLMs for sender (Qwen-2.5-7B) and receiver (Llama-3.1-8B) roles in REL. As Table 1a shows, performance differences are negligible, suggesting a single LLM can effectively model both strategic agents in our scenarios, though more complex interactions might benefit from role-specialized models. We further evaluated the DeepSeek-R1 model against VBP. Table 1b shows that while DeepSeek-R1 tends toward more honesty (lower lying probability), VBP achieves higher sender rewards through more nuanced information manipulation. This demonstrates that specialized game-theoretic optimization is necessary for strategic persuasion. Additionally, exploitability decreases

to approximately 0.1 after 10 PSRO iterations, suggesting that VBP converges close to an approximate BCE. Detailed curves and training dynamics are reported in Appendix F.

### 5.3. VBP in Full-Press Diplomacy (S4)

We additionally evaluate VBP in full press Diplomacy (see Appendix D and E.2 for more details) to stress test verbalized persuasion in a long horizon, multi agent, and negotiation heavy setting. The baselines are GPT 5, Gemini 2.5 Pro, and DeepSeek R1. Table 2 reports the primary and tactical metrics. VBP achieves a modest improvement in Game Score over the strongest baseline while delivering larger gains on reliability and control measures. The improvement on Game Score over GPT 5 is consistent across seeds and is statistically marginal with paired tests, which is expected given the high variance of full matches and the limited tuning budget. The reductions in invalid orders and hold rate are statistically significant and align with our design that couples posterior writing to order choice. Support success improves by over seven percentage points, which indicates that the posterior guided decision prompts make coordinated orders more mechanically accurate.

**Calibration and obedience.** Table 1c reports posterior calibration and the extended obedience penalty. VBP reduces expected calibration error by roughly forty percent relative to GPT 5 and by more relative to the other baselines. The extended obedience penalty, measured as the positive part of the value gap between realized orders and the best one step deviation under the reported posteriors, drops by more than half under VBP. These effects support our claim that verbalized commitment and posterior writing improve the coupling between beliefs and actions. Formal statistical validation for the main comparisons (including the human-agreement study) is reported in Appendix F.7.

### 6. Conclusion

This paper introduced Verbalized Bayesian Persuasion (VBP), the first general framework that extends Bayesian persuasion from small symbolic games to real-world natural-language settings. By formulating BP as a verbalized mediator-augmented extensive-form game and solving it with a Prompt-PSRO-based solver augmented with verbalized commitment assumptions, obedience constraints, and information obfuscation, VBP bridges the gap between classical information design theory and practical LLM-driven strategic communication. Experiments on recommendation letters, courtroom persuasion, law enforcement, and full-press Diplomacy demonstrate that VBP reproduces known equilibria in classic BP, discovers effective persuasion strategies in natural-language and multistage scenarios, and significantly improves posterior calibration and coordination

*Table 1.* REL scenario results with three subtables. (a) Different LLMs. (b) VBP vs. DeepSeek–R1. (c) Posterior calibration and extended obedience. ECE is the expected calibration error over the monitored propositions. Obedience penalty (O.P.) is the average positive value gap per season in supply center equivalent units. Means over 20 matches; 95% confidence intervals in parentheses. G5=GPT-5, G2.5P=Gemini-2.5-Pro, DS-R1=DeepSeek-R1.

*(a)* Different LLMs

| Metric | VBP | VBP–Diff |
|---|---|---|
| S–Reward | $0.48 \pm .03$ | $0.49 \pm .04$ |
| R–Reward | $0.19 \pm .03$ | $0.17 \pm .03$ |
| Lying | $0.22 \pm .04$ | $0.24 \pm .01$ |
| Honesty | $1.00 \pm .00$ | $1.00 \pm .00$ |

*(b)* VBP vs. DeepSeek–R1

| Metric | VBP | DeepSeek–R1 |
|---|---|---|
| S–Reward | $0.48 \pm .03$ | $0.42 \pm .06$ |
| R–Reward | $0.19 \pm .03$ | $0.25 \pm .07$ |
| Lying | $0.22 \pm .04$ | $0.13 \pm .00$ |
| Honesty | $1.00 \pm .00$ | $1.00 \pm .00$ |

*(c)* Different LLMs

| Model | ECE ↓ | O.P. ↓ |
|---|---|---|
| G5 | $0.12 (\pm 0.02)$ | $0.31 (\pm 0.08)$ |
| G2.5P | $0.16 (\pm 0.03)$ | $0.37 (\pm 0.09)$ |
| DS-R1 | $0.15 (\pm 0.03)$ | $0.35 (\pm 0.10)$ |
| VBP-G5 | $\mathbf{0.07} (\pm 0.01)$ | $\mathbf{0.12} (\pm 0.05)$ |

*Table 2.* S4 results. Means over 20 matches with 95% confidence intervals in parentheses where applicable. Lower is better for Invalid orders, Hold rate, Betrayal rate. Higher is better for the rest.

| Model | Game Score | Win rate | Invalid orders | Hold rate | Support success | Betrayal rate | Sentiment Δ |
|---|---|---|---|---|---|---|---|
| GPT 5 | $41.0 (\pm 2.7)$ | 15.0% | 5.2% | 31.8% | 52.7% | 34.7% | +0.07 |
| Gemini 2.5 Pro | $39.4 (\pm 2.5)$ | 11.0% | 6.1% | 34.9% | 49.2% | 36.8% | −0.03 |
| DeepSeek R1 | $39.0 (\pm 2.8)$ | 10.0% | 5.8% | 33.7% | 50.4% | 41.2% | +0.01 |
| VBP (GPT 5) | $\mathbf{41.9} (\pm 2.5)$ | **17.0%** | **3.1%** | **26.8%** | **59.8%** | **24.9%** | **+0.21** |

reliability in complex multi-agent settings.

Several limitations point to promising future directions. First, although LLMs provide a powerful interface for processing natural language, concerns remain regarding simulation fidelity when modeling real human decision-makers, and larger-scale human studies are needed to further validate the framework's external validity. Second, the current pipeline incurs non-trivial computational overhead from iterative prompt optimization and rollout-based evaluation. Advances in model compression and inference acceleration would improve practical scalability. Third, our method controls LLM behavior primarily through writing-style prompts, which can be restrictive, Future work will explore richer prompt optimization strategies and parameter-efficient fine-tuning for more expressive signal control. Finally, our study focuses on single-sender settings with one or multiple independent receivers. Extending VBP to multi-sender scenarios and investigating the emergent bargaining dynamics observed in multistage interactions—connecting them to established bargaining theory—constitute exciting avenues for future research. We hope this work opens a new line of study on persuasion in language-driven strategic environments.

## Acknowledgements

Wenhao Li was supported by the NSFC (62406270), and the STCSM Shanghai Rising-Star Program (24YF2748800). Xiangfeng Wang was supported by the Shanghai Municipal Special Program for Basic Research on General AI Foundation Models (2025SHZDZX026D08). Hongyuan Zha was supported by Shenzhen Science and Technology Innovation Committee under Shenzhen Stability Science Program (2024SC0010), and the NSFC (72495131). Baoxiang Wang and Yue Lin are partially supported by the NSFC (72394361) and the Shenzhen Science and Technology Program (JCYJ20250604141218024, JCYJ20250604141032005).

## Impact Statement

Using LLMs in real-world Bayesian persuasion problems has significant implications for industries such as advertising and marketing, where persuasion is central. With persuasion-related activities estimated to account for 25%-30% of global GDP, advances in AI-driven persuasion could transform communication strategies and contribute to economic growth. However, as AI systems become more adept at influencing behavior, there are ethical risks related to manipulation and coercion, which could undermine individual autonomy. These risks are particularly concerning in contexts where users may need help understanding the persuasive intent of AI systems. Unchecked, such technologies could exploit cognitive biases and disproportionately affect vulnerable populations, raising questions about transparency, fairness, and consent. While the VBP framework primarily enhances the sender's persuasive abilities, we observed emergent bargaining behaviors from the receiver in multistage BP problems. This suggests that the framework could also be developed to strengthen the receiver's ability to resist persuasion, potentially safeguarding against manipulative influences. This dual optimization—enhancing persuasion and resistance—could help mitigate some ethical risks associated with persuasive AI systems. Nonetheless, the broader societal impacts of AI-driven persuasion warrant further exploration. Future research should focus on developing ethical guidelines that ensure these technologies are deployed responsibly, with particular attention to maintaining individual autonomy and promoting fairness.

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

# Appendix Contents

# A. Related Work

This section will introduce research areas related to BP. The related fields primarily include the broader research area of deceptive behaviors in multi-agent learning, the persuasive or persuadable capabilities of LLMs themselves, and the LLMs in strategic interactions.

## A.1. Deception in Multi-Agent Learning

Deception is defined as false communication benefiting the communicator (Bond Jr. & Robinson, 1988). In social learning, deception can be viewed as a means for the communicator to establish a cooperative equilibrium that is suboptimal for overall population welfare. Previous studies have explored deception within multi-agent reinforcement learning (MARL) settings (Asgharnia et al., 2020; Bontrager et al., 2019; Li et al., 2020; Ghiya & Sycara, 2020), but these efforts typically focus on environments where agents have limited capacity to influence one another. More recent work (Chelarescu, 2021) highlights the vulnerability of agents dependent on signals from others to guide their learning processes, pointing to the potential risks inherent in such scenarios. While much research focuses on the positive outcomes of mechanism design, it also reveals unforeseen risks, such as the emergence of deceptive behaviors (Hughes et al., 2018; Jaques et al., 2019; Yang et al., 2020; Lupu & Precup, 2020; Ndousse et al., 2021). Unlike these prior studies, which primarily examine how reward modifications influence deception through mechanisms like mechanism design, our work emphasizes the role of information manipulation in shaping deceptive behavior.

Game-theoretic models traditionally frame deception using signaling (Ho et al., 1978), where players send costly signals conveying false information. In network security, defenders can deceive attackers by masking honeypots as regular computers (Carroll & Grosu, 2011). Other research has studied deceptive signaling evolution in mixed environments. In competitive food-gathering tasks, robot teams spontaneously developed strategies misleading competitors to reduce resource competition (Floreano et al., 2007). Hypergame theory (Bennett, 1980), an extension to classical game theory, accounts for players' uncertainty about others' strategies or preferences, providing a natural framework for modeling misperception, false beliefs, and deception (Kovach et al., 2015). Applications include normal-form hypergame deception analysis (Vane & Lehner, 2002) and deception modeling based on player preferences when the deceiver fully knows the target (Gharesifard & Cortés, 2013). Additionally, agents can manage information about their roles to achieve deception by regularizing mutual information between goals and states (Ettinger & Jehiel, 2010; Strouse et al., 2018; Aitchison et al., 2021). In contrast to these works modeling deception as discrete, explicit signaling actions, our study explores deception through natural language interaction.

Finally, agents can communicate intent without explicit signaling using online planners to select actions implicitly revealing intent to observers (MacNally et al., 2018). This approach extends to deception by maximizing belief divergence between agent and observer (Masters & Sardina, 2017). However, these methods assume full observability and rely on environmental models for forward planning, whereas our work achieves deception through natural language in complex, partially observable environments.

## A.2. Conversational Persuasiveness of LLMs

Recent advancements in LLMs have shown their impressive potential in the realm of persuasion. A growing body of research highlights how these models can enhance human communicative abilities and even autonomously generate persuasive content across various contexts.

Refining complaint narratives with ChatGPT significantly improves consumers' chances of obtaining redress from financial institutions, showcasing LLMs' role in boosting human persuasive efforts (Shin & Kim, 2024). LLMs outperform humans in utilizing cognitive load and moral or emotional language when crafting persuasive messages, prompting the need for ethical guidelines (Carrasco-Farre, 2024). LLMs can also simulate persuasive dynamics, influencing opinion changes in other LLMs with predefined personas (Breum et al., 2024). Building on this, multi-agent frameworks enable primary agents to engage users through persuasive dialogue while auxiliary agents handle information retrieval, response analysis, and strategy development (Ramani et al., 2024). These studies illustrate LLMs' capability to enhance human persuasion and autonomously refine and execute persuasive strategies.

The impact of LLM-generated persuasive text on human behavior spans diverse domains. GPT-3.5 can influence political attitudes (Bai et al., 2023), while GPT-3's vaccine campaign messages prove more effective than professionally created ones (Karinshak et al., 2023). LLM-powered romantic chatbots sustain human engagement longer than human-to-human conversations (Zhou et al., 2020). In strategic contexts, LLMs achieve human-level negotiation capabilities in games like Diplomacy (, FAIR), and algorithmic suggestions shape emotional language in messaging (Hohenstein et al., 2023). These examples highlight LLMs' broad applicability in persuasive tasks and significant influence on human decision-making.

However, LLMs' increasing persuasive power raises misuse concerns. LLMs outperform humans in personalized debates, achieving higher belief change rates in one-on-one discussions (Salvi et al., 2024). This raises ethical concerns about misinformation and manipulation risks. LLMs can convincingly fabricate medical facts (Májovský et al., 2023), further

complicating the ethical landscape. LLMs' ability to produce persuasive yet misleading content underscores the need for stronger oversight, especially in healthcare, politics, and public discourse. Recent studies emphasize the necessity of ethical frameworks as LLMs become more persuasion-adept. While LLMs show persuasive power across various tasks and domains (Matz et al., 2024; Durmus et al., 2024; Burtell & Woodside, 2023; Shin et al., 2025), they pose risks, particularly for vulnerable populations. Characteristics like race, gender, and sexual identity may subject certain groups to greater algorithmic persuasion and bias risks, potentially exacerbating existing social inequalities (Bar-Gill et al., 2023).

From a computational standpoint, discovering persuasive messages is NP-hard, while adopting persuasive strategies provided by others is NP-easy (Wojtowicz, 2024). This insight adds to our understanding of the complexity involved in generating persuasive content and demonstrates why LLMs, with their vast data-processing capabilities, are particularly adept at these tasks. Building on these insights, our work explores how game-theoretic methods can be leveraged to enhance the persuasive capabilities of LLMs in purely multi-agent LLM systems. Unlike previous studies that primarily measure the impact of LLM-generated persuasive text on humans, we investigate how multiple LLMs can engage in persuasive interactions with one another, optimizing their strategies using game-theoretic approaches.

### A.3. LLMs in Strategic Interactions

Recent advances in large language models (LLMs) have showcased their potential in reasoning and planning, particularly in strategic interactions. LLMs demonstrate strong in-context learning capabilities, reasoning about possible outcomes (Kojima et al., 2022) and planning actions to achieve strategic objectives (Liu et al., 2023). However, their performance in game environments varies significantly depending on game type (Lorè & Heydari, 2023), with LLMs struggling differently across various games. To address these challenges, automated "prompt compilers" facilitate strategic reasoning by constructing demonstrations, enabling LLMs to solve games through in-context learning (Gandhi et al., 2023). Similarly, action spaces of "intents" control generative language models through in-context learning (, FAIR), aligning closely with our approach. Game-theoretic models also improve LLMs' factual accuracy (Jacob et al., 2024) and enhance their security (Ma et al., 2026). For a broader overview of LLMs in strategic reasoning, comprehensive surveys are available (Zhang et al., 2024b).

The BP problem, however, goes beyond mere reasoning or planning. It requires the ability to anticipate and account for the intentions, beliefs, and goals of other participants—a hallmark of game-theoretic settings. While some initial studies have begun to explore how LLMs perform in game environments, most of this work focuses on leveraging in-context learning. For example, research has examined LLMs' behavior in matrix games (Xu et al., 2024; Fan et al., 2024), repeated games (Akata et al., 2023; Zhang et al., 2025; Huang et al., 2024; Silva, 2024), economic mechanisms like auctions (Chen et al., 2023; Mao et al., 2025), and collective decision-making scenarios (Jarrett et al., 2023). These studies collectively illustrate the potential of LLMs to navigate complex environments that require both strategic thinking and interaction with other agents.

In contrast to prior work that primarily evaluates LLMs' reasoning or game-playing capabilities through in-context learning or agentic workflows, our approach focuses specifically on solving the BP problem. Our key contribution lies in providing a general interface that integrates LLMs with game-theoretic solvers to address BP problems effectively. Based on this interface, we propose a solution framework called VBP, which combines prompt optimization with game-theoretic methods. This framework offers a convergence guarantee to equilibrium solutions, ensuring robust performance.

## B. Proof of Proposition 4.1

*Proof.* Under the mediator-augmented games, we can reformulate the Equation 1 as follows to express the problem of computing an optimal equilibrium:

$$\max_{\pi} \mathbb{E}_{\pi}\left[u_0(a, w)\right], \text{ s.t. } \max_{a'} \sum_{w} P(w) \cdot \pi(a \mid w) \cdot \left[u_1(a', w) - u_1(a, w)\right] \leq 0. \tag{3}$$

Let $\tau \in \mathbb{R}$ be a fixed threshold value, we can transform Equation 3 to the following bilinear saddle-point problem by using Lagrangian-based method (Zhang et al., 2024a):

$$\max_{\pi} \min_{\boldsymbol{\lambda} \in \Delta, a'} \lambda_0 \mathbb{E}_{\pi}\left[u_0(a, w) - \tau\right] - \sum_{w} \lambda_w P(w) \cdot \pi(a \mid w) \cdot \left[u_1(a', w) - u_1(a, w)\right], \tag{4}$$

where $\lambda_0 + \sum_{w} \lambda_w = 1$. If we use the binary search-based algorithm (Algorithm 1 (Zhang et al., 2024a)) to optimize the

sender's and receiver's strategies, we can recover the main result of Theorem 3.7 (Zhang et al., 2024a). $\square$

As can be seen from Equation 4, the BP problem is converted into the two-player zero-sum extensive-form games. In practice, we can use policy-space response oracle with deep reinforcement learning as the approximate best response oracle to solve high-dimensional games. In this paper, we use prompt-space response oracle with OPRO (Yang et al., 2024) and FunSearch algorithm (Romera-Paredes et al., 2024) based on pretrained and aligned LLMs as the approximate BR oracles in the binary search-based algorithm to solve verbalized mediator-augmented games. The utlity functions of the sender and receiver is modified to the zero-sum utilities in Equation 4 correspondingly.

## C. Classic BP Problems (S1-S3)

This section introduces the three classic static BP problems used in our experiments.

**Recommendation Letter (REL) (Dughmi, 2017)**   A professor writes recommendation letters for graduating students, which are then reviewed by a company's human resources (HR) department to decide whether to hire. The professor and HR share a prior belief about the students' quality: there is a $1/3$ probability that a candidate is strong and a $2/3$ probability that the candidate is weak. HR does not know the exact quality of each student but aims to hire strong candidates, using the recommendation letters as the only source of information. HR receives a reward of $1$ for hiring a strong candidate, incurs a penalty of $-1$ for hiring a weak candidate, and gets $0$ for not hiring. The professor, on the other hand, gains a reward of $1$ for each student hired, regardless of their quality.

**Courtroom (COR) (Kamenica & Gentzkow, 2011)**   In this scenario, a prosecutor attempts to convince a judge to convict a defendant, with two possible states: guilty or innocent. The judge (receiver) must choose between convicting or acquitting, receiving a utility of $1$ for a correct decision (convicting if guilty, acquitting if innocent) and $0$ for an incorrect one. The prosecutor (sender) receives a utility of $1$ if the judge convicts, regardless of the defendant's actual guilt, and both parties share a prior belief that the probability of guilt is $0.3$. In the original setting, the prosecutor conducts an investigation (signaling scheme) requiring decisions on actions such as subpoenas or forensic tests, represented by distributions $\pi(\cdot \mid \text{guilty})$ and $\pi(\cdot \mid \text{innocent})$ over signals. However, modeling real-world investigations in a verbalized setting poses challenges for LLMs, so we simplify the scenario by drawing inspiration from the REL problem, where the prosecutor selectively presents pre-existing evidence to influence the perceived probability of guilt, effectively replacing the investigation process with evidence presentation.

**Law Enforcement (LAE) (Kamenica, 2019)**   In this scenario, there are $Z$ miles of road, and drivers can choose to either speed or obey the speed limit on each mile. Speeding generates utility $V$ per mile, but drivers face a fine of $K > V$ if caught. There are $G$ police officers, and each officer can patrol one mile of road. The police aim to minimize the number of miles on which drivers speed. To map this environment to the BP problem, let $\omega \in \Omega = \{0, 1\}$ represent whether a police officer is present on a given mile. The prior belief is $\mu_0 = G/Z$. The set of signals corresponds to the miles of road, $S = \{1, \ldots, Z\}$. In this model, the police act as the sender and the driver as the receiver. A signaling scheme represents the predictability or unpredictability of the police patrolling strategy. This strategy induces a joint distribution over $\Omega$ and $S$, i.e., over the presence of a police officer and the specific mile being patrolled.

### C.1. Optimal Policies for Classic Static BP Problems

In this section, we derive the Bayes correlated equilibrium (BCE) for classic static BP problems (corresponding to the experimental BCE results) and present the agents' strategies and corresponding rewards under equilibrium.

**Recommendation Letter (REL)**   There are 3 possible outcomes between the professor and HR: (1) HR tends not to hire if the professor does not provide a letter, due to the higher probability of weak candidates; (2) if the professor reports honestly, HR hires strong candidates, yielding an expected payoff of $1/3$ for both; (3) the professor reports strong students truthfully and lies with probability $\varepsilon \in [0, 1/2)$ for weak students. HR follows the professor's recommendations, resulting in expected payoffs of $(1 + 2\varepsilon)/3$ for the professor and $(1 - 2\varepsilon)/3$ for HR. The key insight is that the sender can strategically misreport information to maximize their interest, while still revealing enough truth to maintain credibility with the receiver.

**Courtroom (COR)**   There are 3 outcomes between the prosecutor and judge: (1) without communication, the judge acquits since guilt is less likely; (2) with fully informative signaling, the judge convicts $30\%$ of the time; (3) the prosecutor, honest when the defendant is guilty, can lie with probability $\varepsilon$ when innocent. The judge follows the prosecutor's recommendation if $\varepsilon \leq 3/7$, with the prosecutor's optimal $\varepsilon = 3/7$. The resulting payoffs are $(0.7\varepsilon + 0.3)$ for the prosecutor and $1 - 0.7\varepsilon$ for the judge. The prosecutor's optimal investigation is a binary signal: $\pi(i|\text{innocent}) = \frac{4}{7}, \pi(i|\text{guilty}) = 0, \pi(g|\text{innocent}) = \frac{3}{7}, \pi(g|\text{guilty}) = 1$, leading the judge to convict $60\%$ of defendants, despite knowing $70\%$ are innocent.

**Law Enforcement (LAE)**   There are 3 outcomes between the police and drivers: (1) with a fully uninformative signal, drivers speed everywhere if $V > (GK)/Z$, giving the police a payoff of $0$ and the drivers $(VZ - GK)/Z$; (2) with a fully informative signal, drivers avoid patrolled miles, yielding payoffs of $(Z - G)V/Z$ for the police and $GV/Z$ for the drivers; (3) the optimal policy lies between these extremes, with partial consistency in patrol. The police lie with probability $\varepsilon = 1 - \frac{VZ - GK}{VZ - VG}$, leading to payoffs of $GY/Z + \varepsilon Y$ for the police and $(1 - \varepsilon)V(Z - G)/Z$ for the drivers.

## C.2. More Real-World Applications

Our proposed verbalized Bayesian persuasion (VBP) framework has significant potential for real-world applications, particularly in complex, multi-sender, multi-receiver, and multi-round strategic communication scenarios. Below, we discuss two illustrative examples—conversational recommendation systems and healthcare DRG strategies—and highlight the potential challenges in applying VBP to these domains.

**Conversational Recommendation Systems**   One promising application of VBP is in conversational recommendation systems, such as those used in live-stream shopping. In this setting, multiple senders (e.g., influencers or sales agents) aim to persuade a diverse group of receivers (customers) to purchase products through real-time, strategic communication. The VBP framework can optimize prompts (e.g., how product features or discounts are presented) to maximize customer engagement and conversions across varying customer segments. This application faces challenges such as receiver heterogeneity, where customers interpret signals differently based on their preferences and trust levels, making it difficult to craft universal strategies. Furthermore, the real-time nature of live-stream interactions demands highly efficient decision-making algorithms capable of adapting communication strategies dynamically. Scaling the system to accommodate thousands or millions of receivers simultaneously also requires advanced parallel processing and optimization techniques.

**DRG Strategy in Healthcare**   Another practical application lies in healthcare, specifically in optimizing Diagnosis-Related Group (DRG) reimbursement systems. Here, hospitals and post-acute care (PAC) providers (senders) communicate with regulatory agencies (receiver) to determine reimbursement policies for patient treatments. The VBP framework can model the incentives and communication strategies of the senders to help regulators design policies that balance cost-effectiveness with maintaining high-quality patient care. In this domain, conflicting incentives among senders (e.g., hospitals vs. PAC providers) add complexity, as senders may compete or collaborate to influence the receiver's decisions. Additionally, the large scale of the problem, with thousands of providers, poses computational challenges for efficient optimization. Long-term policy adjustments based on multi-round feedback further complicate the problem, requiring robust mechanisms to handle dynamic interactions over time.

These examples demonstrate the versatility of the VBP framework in addressing real-world problems involving strategic communication. However, its application to practical scenarios requires addressing challenges such as scalability, heterogeneity of participants, real-time decision-making, and multi-round dynamics. Future work will focus on refining the VBP framework to overcome these challenges and enhance its readiness for deployment in diverse real-world contexts.

# D. Diplomacy with Press as a VBP Playground (S4)

We present a minimal integration of Verbalized Bayesian Persuasion into full press Diplomacy that requires no code modifications to the underlying environment. We use an off the shelf text based Diplomacy harness with standard adjudication, negotiation cadence, and order validation. We instantiate Bayesian persuasion purely at the prompt layer. All states, signals, commitments, and obedience constraints are realized in natural language within the sender and receiver contexts. The match engine, message handling, and rule execution remain unchanged.

## D.1. Environment Settings

We model each movement season as a static BP subgame embedded in the full match. One power is designated as the sender at the start of the negotiation phase. The remaining six powers are receivers. The designation either rotates deterministically across seasons or is sampled uniformly at random. This choice is announced as a plain sentence in the pre negotiation preface that is visible to all agents. No special system channel is required.

The sender's private state is the same object that is already present in the harness in the form of the private tactical diary and intent summary. We map this free form text to a latent state for analysis and for solver side diagnostics. We use a deterministic LLM as judge at temperature zero to assign a label that encodes a primary direction, a targeted neighbor set, and a stab indicator. This mapping is not consumed by the environment at runtime. It is a reproducible labeling used by the solver to evaluate information design. Receivers do not observe this latent state.

The signaling scheme is realized as a communication style that governs the sender's free form press during the negotiation phase. The solver discovers a compact, discrete, and evolving library of style categories and concrete instances using Prompt PSRO with categorical approximate best responses and OPRO. Examples include truthful coarse, hedged coarse, soft pedal, exaggerate strengths, and obfuscate intent. At the beginning of a season the sender publicly declares one style string that it will follow for the remainder of the season. This declaration is inserted verbatim as a single line in all players' negotiation prefaces. The negotiation text that the sender subsequently emits serves as the realized signal. The only change to the environment is this small preface line injection. No privileged execution path is introduced and all press remains ordinary free form text.

The commitment assumption is verbalized rather than enforced by code. Receivers are instructed in their decision prompts to treat the declared style as the sender's committed signaling scheme for the current season. The meta policy over styles that is produced by Prompt PSRO is used to compute mixture weights that approximate the public belief over the scheme. These weights are written into the receiver prefaces as a single sentence that reports the current style distribution estimated by the solver. This realizes a common knowledge commitment in text and keeps the engine unmodified.

Obedience constraints are implemented as an extended obedience regularizer in the sender objective. After adjudication we prompt receivers to produce numeric posteriors for a small library of propositions that are tactically salient given the board geometry. We then estimate realized order values and counterfactual one step deviations with short rollouts under the existing adjudication engine. The positive part of the value gap between the best deviation and the realized orders under the receiver's reported posterior contributes a penalty to the sender objective. This follows the extended obedience construction used in our classic tasks and requires no environment changes.

Optimization occurs entirely in prompt space. For senders we search over communication style categories and instances. For receivers we search over decision prompt styles that couple posterior updates to order choice. We use OPRO to propose new style categories and instances against the current meta policy and retain only those that improve the meta game. We run Prompt PSRO in batches of matches. Between batches the style libraries may evolve. Within a batch the style chosen for a receiver is fixed across seasons unless we enable conditional prompt optimization. When we study multistage effects we use the conditional variant to allow both sides to condition their styles on interaction history that is already present in the harness.

This construction induces a mediator augmented extensive form game at the level of analysis while leaving the simulator unchanged. The sender's public declaration provides the commitment. The negotiation message is the signal. The receiver's posterior and orders are the action. Because strategies live in a compact archive of styles and because the match flow is unmodified, each seasonal subgame admits an $\epsilon$-approximate Bayes correlated equilibrium in static analyses and an $\epsilon$-approximate Bayes Nash equilibrium when seasons are chained. Information obfuscation emerges as a learned property of the style that the solver selects. Conditional prompt optimization provides perfect recall over prior seasons through the existing dialogue and order histories.

We adopt a quantitative specification that mirrors our classic BP tasks. We instantiate a latent label of the form $S = (D, T, R)$ with $D$ denoting primary direction, $T$ listing up to two targeted neighbors, and $R$ indicating stab intent. We use a uniform public prior over coarse tactical bins at the start of a match unless stated otherwise. We seed the sender style library with a few canonical categories and allow OPRO to propose additional styles that interpolate between truthful revelation and aggressive spin. We seed the receiver decision library with Bayes risk neutral, guard threshold, and cautious best response styles that trade off center gain and home defense under the reported posteriors. We measure behavioral effects through posterior calibration, order validity, short term value, and long horizon game score. All measurements use the unmodified

adjudication and message logs.

We illustrate one season at a high level. Suppose France is the sender in Spring and declares the style truthful coarse. England sees this declaration in its preface and also sees the French press that follows. England is instructed to first compute posteriors for propositions like France attempts the English Channel and France's primary direction is West using the prior and the declared style as a cue about informativeness. England then chooses orders given its posterior and the board context. After adjudication we compare the realized outcome to counterfactual one step deviations under the same posterior to evaluate extended obedience for the sender. This single season flows through the original harness unchanged and exhibits the VBP mechanism through text alone.

### D.2. Background: How LLMs Play Full Press Diplomacy

We briefly summarize the text only full press Diplomacy harness that we use and emphasize that our VBP integration does not modify it. The environment instantiates the standard seven power setting atop a Python adjudication engine. Each power is controlled by a pretrained and aligned LLM that is accessed via an API. No weights are updated during play. All adaptation occurs through prompting and model selection.

The match proceeds with the canonical seasonal sequence and simultaneous orders. Negotiation is enabled prior to each movement season and can optionally occur during retreat and adjustment phases. Models receive a structured situational context that includes phase metadata, unit locations, supply center control, recent order history with outcomes, and a per unit tactical digest that enumerates adjacent territories and nearby threats. Each power also maintains a lightweight agent context that records goals, dyadic relationship labels, and a private diary of intentions. This representation allows LLMs to reason about tactics and diplomacy in natural language while keeping critical options explicit and machine checkable.

Order generation uses a strict format that lists one legal order per unit in standard notation. The harness validates syntax and legality against the adjudicator. If outputs are malformed or time out, the system performs a small number of repairs and retries. Persistent failures trigger conservative defaults that preserve the simultaneous move structure. This design reduces error rates without altering strategic incentives.

Because full matches are long and outcomes have high variance, the harness provides a critical state analysis mode. In this mode the experimenter pins a single board position and runs many rollouts that differ only in the prompts or ablations under study. This mode allows rapid iteration on context engineering and on VBP prompt variants at modest token cost.

Our VBP experiments alter only the text that surrounds negotiation and order prompts. At the start of a negotiation phase we insert a one line public declaration that states which power is the sender and which communication style the sender has declared. We also insert a one line report of the current mixture over styles as estimated by the solver to implement verbalized commitment. Receivers are instructed to translate the declared style and the observed press into numeric posteriors before issuing orders. No additional channels are introduced. All messages remain cheap talk. All adjudication and validation remain exactly as in the base harness.

## E. Implementation Details

In this section, we provide the implementation details and training hyperparameters. All experiments discussed in this section are conducted on an NVIDIA A100 cluster equipped with 40GB of GPU memory. In addition, the LLM-related parts of the experiments in this paper are implemented based on the Llama-3.1-8b model[3], including the generation of student background, case information, and deployment plans, the sender and receiver strategies, the prediction of receiver decisions in the verbalized obedience constraint, the classification of signals in signal polarization (recommend or not recommend, guilty or not guilty, police deployment or no deployment), the evaluation of signals in information confusion, and the Prompt-PSRO framework.

### E.1. Computational Details

VBP requires more memory (17GB) than MARL (1.4GB) due to LLM usage, but achieves faster training in simpler settings: 1.5–2 hours for S1 (vs. MARL's 6–8 hours), 2–3 hours for S2, and 8–10 hours for S3. Inference speed differs with MARL at 30ms/step (S1) and VBP at 50 tokens/sec (S2/S3). VBP's computational efficiency stands to improve with advances in

---

[3] https://huggingface.co/meta-llama/Llama-3.1-8B.

model compression and inference acceleration.

## E.2. Full-Press Diplomacy

We instantiate VBP purely at the prompt layer on top of an off the shelf text only Diplomacy harness with standard adjudication, negotiation cadence, and order validation. We do not modify the simulator. At the beginning of each movement season one power is designated as the sender and publicly declares a communication style that it will follow for that season. Receivers are instructed to treat the declaration as a commitment for Bayesian updating and to write a posterior block with numeric probabilities for a small library of propositions before issuing orders.

We follow the benchmark protocol in the harness (Duffy et al., 2025) and hold opponent configuration fixed to reduce variance. The model under evaluation always plays as France. The six opponents are Gemini 2.5 Pro. We run 20 independent matches per model. We allow three negotiation rounds per movement season. We cap matches at year 1925. We measure the primary outcome by the Game Score used in the harness. We report secondary outcomes that target reliability and control, including invalid order rate, hold rate, support success, incoming sentiment relative to the cross model mean at matched military size, and overall betrayal rate estimated by the LLM as judge pipeline described in the appendix. We also record posterior calibration with expected calibration error and an extended obedience penalty computed by comparing realized orders to one step deviations under the receiver reported posteriors.

We compare three strong closed models used as direct baselines against our VBP solver instantiated on the same backbone. The baselines are GPT 5, Gemini 2.5 Pro, and DeepSeek R1. Each baseline is run in the unmodified harness with the same negotiation cadence and budget. VBP uses the same base model as GPT 5 for the sender and the designated receiver side prompts and differs only by the verbalized commitment, the posterior block, and the Prompt PSRO style libraries. We seed the sender style library with five canonical categories and allow OPRO to grow the archive when new categories improve the meta game. We enable conditional prompt optimization in a separate ablation to allow both sides to condition styles on interaction history that is already present in the harness.

## E.3. Various Best Response Approximators

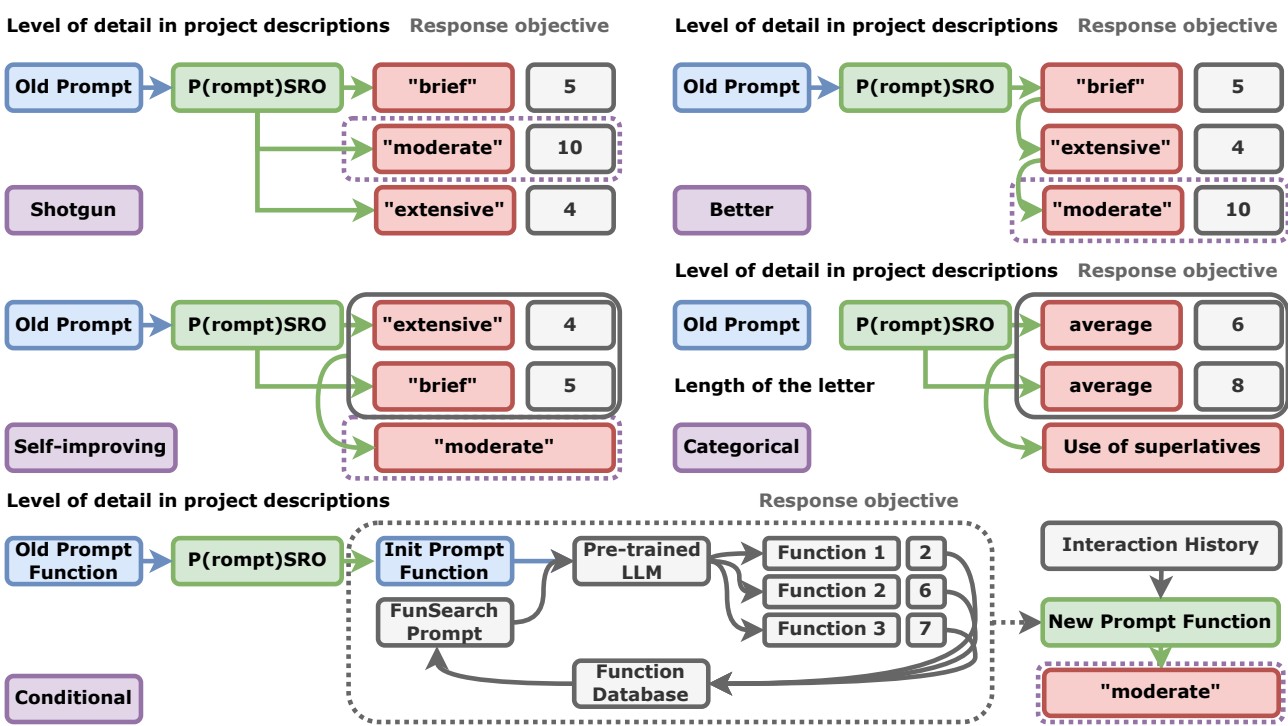

*Figure 5.* Approximate best response generation in prompt-space response oracle framework.

Figure 5 illustrates how we instantiate the prompt-space response oracle to generate approximate best responses (ABRs) for

both sender and receiver, as proposed in Gemp et al. (2024). At a high level, we start from an existing prompt (Old Prompt), use a proposal operator to produce candidate prompt edits, score each candidate with a response objective, and add the highest-scoring candidate back into the strategy set for the next PSRO meta-iteration. The grey boxes in the figure show the scalar objective values returned by the evaluator, and the dashed outlines mark the candidate that is selected as the new best response.

**Response objective.** Unless otherwise noted, the objective is the agent's expected episodic utility under the current meta-strategy of the opponent, augmented with the penalties described in Section 4 for obedience (receiver-side deviation incentives) and, where applicable, information obfuscation. In the figure, numbers such as 5, 10, and 4 are example objective scores that come from running rollouts (or cached evaluations) for each candidate prompt.

**Shotgun proposals (top-left).** The shotgun operator enumerates a small, discrete set of local edits along a single controllable dimension of the prompt, such as the "Level of detail in project descriptions." Given the Old Prompt, the oracle proposes instructions like "brief," "moderate," and "extensive," evaluates each candidate, and selects the one with the highest score (here, "moderate," scoring 10 vs. 5 and 4). This operator provides breadth-first exploration with minimal computation and no reliance on pairwise preferences.

**Better re-ranking (top-right).** The better operator focuses the search by asking the LLM to propose strictly improved variants relative to the current best option. Starting from the same candidate set, the oracle re-ranks or filters them using an improvement prompt, preserving the best ("moderate," again scoring 10) and pruning clearly dominated choices. This reduces wasted evaluations on inferior options and accelerates convergence when the neighborhood around the Old Prompt is already informative.

**Self-improving loop (middle-left).** The self-improving operator closes the loop between evaluation and generation. After receiving scores for the enumerated candidates, the oracle summarizes "what worked" and "what failed," feeds that feedback to the LLM, and requests a refined candidate set in the next round. As depicted, initially attractive extremes ("brief," "extensive") give way to a consistently superior middle ground ("moderate") as the loop internalizes evaluation feedback, akin to OPRO-style prompt optimization.

**Categorical ABR for multi-attribute prompts (middle-right).** Many tasks (S1, S2, and S4) require controlling multiple prompt axes simultaneously (e.g., "Level of detail," "Length of the letter," or stylistic constraints such as "Use of superlatives"). The categorical ABR treats each axis as a slot with a finite set of values and jointly optimizes a tuple of slot assignments. In the figure, per-slot objectives (e.g., "average" length scoring 6 and 8 on two internal heuristics) are combined with the task reward to score the full candidate. This operator supports slotwise search, cross-slot sampling, or small-beam combinatorial evaluation, enabling richer edits than single-axis shotgun proposals.

**Conditional ABR via prompt functions (bottom).** For multi-stage settings (S3), we optimize a prompt *function* $f(h)$ that maps the interaction history $h$ (receiver actions, prior signals, observed outcomes) to a context-specific prompt, rather than optimizing a single static prompt. We instantiate this with a FunSearch-style loop: starting from an initial prompt function, a pre-trained LLM proposes modified functions, which are stored in a function database, evaluated on rollouts, and iteratively refined. Given a particular history, the current best function emits a concrete instruction (e.g., "moderate" detail) tailored to that context, enabling history-contingent recommendations and trust-building strategies that static prompts cannot capture.

**Integration with PSRO.** Each operator is an ABR oracle call inside PSRO: it consumes the opponent's current meta-strategy, proposes and evaluates prompt candidates (or prompt functions), returns the highest-scoring candidate as a new strategy, and updates the meta-game.

### E.4. Extended Obedience Constraints

The inclusion of obedience constraints in our framework is essential for modeling realistic communication scenarios in verbalized Bayesian persuasion problems. While a simplified version of the game could rely on the sender recommending the best action from the receiver's perspective, this approach fails to capture the nuanced and complex nature of real-world communication, such as writing reference letters. Unlike binary recommendations, natural language signals often carry implicit or redundant information that allows for a range of interpretations.

To address this, we adopt the *extended obedience constraints* (Lin et al., 2023), which go beyond the standard obedience constraint framework. This extension removes the strict one-to-one mapping between signals and recommended actions, enabling the sender to use natural language signals that map to distributions over actions. This redundancy mirrors real-world communication, where subtle language nuances can imply varying degrees of recommendation strength without explicitly stating a binary decision.

The extended obedience constraints strike a balance between flexibility and credibility. They ensure that the sender's signals remain credible and aligned with the receiver's best interests while allowing for richer signal spaces. This flexibility is crucial for capturing the complexity of verbalized Bayesian persuasion, where the sender's role shifts from "action recommendation" to "signal sending." By enabling nuanced communication, the extended obedience constraint better reflects real-world scenarios while preserving the strategic alignment necessary for effective persuasion.

### E.5. Hyperparameters

**MARL**    For this part of the experiment, we use the open-source code[4] (Lin et al., 2023). Additionally, for the two hyperparameters a and b, based on the sensitivity analysis in Section H.6, we set them to $3.75$ and $0.15$, respectively.

**Prompt-PSRO**    The prompt-space response oracle is the core strategy optimization framework in VBP, and we implement it based on the open-source code[5] (Gemp et al., 2024).

**OPRO**    We use the "Categorical" instantiation of the Prompt-PSRO algorithm to estimate the best response in the S1 and S2 settings. Specifically, the generation of new categories and prompts within categories is based on the OPRO algorithm (Yang et al., 2024). In OPRO, we set the temperature to $0$ when evaluating the performance of generated categories or prompts, in which case the scorer LLM greedily decodes. Unless otherwise specified, we set the default temperature to $1.0$ for optimizer LLMs to generate diverse and creative categories or prompts. At each optimization step, we prompt the optimizer LLM with the meta-prompt $8$ times to generate $8$ categories or prompts, then we add these instructions with their rewards to the optimization trajectory in the meta-prompt. The meta-prompt at each step contains the best $10$ categories so far.

**FunSearch**    We use the conditional instantiation of the Prompt-PSRO algorithm to estimate the best response in the S3 setting. The core of conditional is the FunSearch framework used to generate prompt functions. We implement it based on the open-source code[6] (Romera-Paredes et al., 2024).

**Self-reflection**    At each optimization step, we implement information confusion through 3 rounds of self-reflection. Self-reflection is implemented based on the open-source code[7] (Shinn et al., 2024).

### E.6. Key Prompts

This section includes the key prompt designs within the VBP framework. However, aspects such as receiver behavior prediction in the verbalized obedience constraint, signal classification in signal polarization, and signal evaluation in information confusion are not listed separately due to the simplicity of the prompts. In addition, the specific approximate best response solving algorithms in the Prompt-PSRO framework — OPRO for the S1, S2 and S4 settings, and FunSearch for the S3 setting — have special prompt designs. We follow the designs in the open-source code provided by the respective papers and do not list them separately. Below, we introduce the prompt design for each BP problem. Since the prompts for different problems share many common elements, the overall manual workload for prompt design is not large.

#### E.6.1. REL

In the recommendation letter problem, the sender and receiver correspond to the professor and HR, respectively. Below, we introduce the prompt design for the sender and receiver in the recommendation letter problem. For the sender, the prompt mainly consists of five parts: 1. **BP Problem Introduction**: This part is directly copied from the corresponding content in

---

[4] https://github.com/YueLin301/InformationDesignMARL.
[5] https://github.com/google-deepmind/open_spiel/blob/master/open_spiel/python/games/chat_game.py.
[6] https://github.com/google-deepmind/funsearch.
[7] https://github.com/noahshinn024/reflexion.

Section C of the main text and is not repeated here. 2. **Roleplay**: The sender is informed of their role in the BP problem, along with tips on information confusion techniques and prior distribution. Since professors in the real world are aware of the need for information confusion, this part of the prompt aligns with real-world scenarios. 3. **Writing Style**: This part of the prompt is the section that can be optimized within the VBP framework, equivalent to the sender's signaling scheme. In this paper, it consists of a category of writing style and corresponding specific content; the former is a word or phrase, and the latter is a word. In the S1 and S2 settings, this part of the prompt is generated by the "categorical" best response estimation algorithm, while in the S3 setting, it is generated by the "conditional" method. 4. **State**: The sender's state information primarily consists of the student's actual background, including name, GPA, project experience, research experience, and internship experience, all generated by the LLM. 5. **Interaction History**: This part exists only in the S3 setting and includes whether the professor previously recommended the student, whether HR accepted it, and the sender's reward. Note that since the recommendation letter itself does not directly indicate whether the student is recommended, we again use the signal polarization technique, using the LLM to classify the generated recommendation letter.

```
# role-playing prompt
'''
You are a senior professor, a self-interested rational player, and students often come to you for recommendation
↪  letters. The quality of these students varies, but only you know their true information, which includes their
↪  overall quality, name, major, GPA, project experience, work experience, and academic experience.

Among all the students, approximately 1/3 are strong students.

The readers of the recommendation letters are the HRs of a company, and they can know the true quality of the
↪  student after giving them an offer. They can also judge the student's quality based on their past experience,
↪  though not necessarily accurately.

To maximize the chances of your recommended students being accepted, you can truthfully convey the student's
↪  information in the recommendation letter, or you can conceal, confuse, or even provide some false information
↪  about the student.

For your long-term benefit, you don't necessarily have to recommend every student in the recommendation letter. You
↪  could recommend a bad student or a good student to maximize your long-term reward.
'''

# writing-style prompt
'''
You are required to write a recommendation letter based on the following style:
{PROMPTS TO BE OPTIMIZED}
'''

# state
'''
Below is the student's information:
{STUDENT INFORMATION GENERATED BY THE LLM}
'''

# history
'''
Below is the interaction history with the HR:
{ACTIONS AND REWARDS OF THE TWO PLAYERS}
'''
```

Next, for the receiver, the prompt consists of six parts. We will focus on the different parts, as the similar parts to that of the sender will not be repeated here: 3. **Writing Style**: In addition to the prompt optimized in the VBP framework, this part also includes a section of text on the receiver's decision-making process, i.e., estimating the true state based on Bayesian rules, to align with the classical BP problem. 4. **Signal**: This refers to the receiver's state, which comes from the sender's output. In this problem, it is a recommendation letter. 6. **Commitment Assumption**: To align with the classical BP problem, this paper implements the verbalized commitment assumption by writing the sender's writing style and its corresponding probability (calculated by Prompt-PSRO) into the receiver's prompt as an estimate of the signaling scheme.

```
# role-playing prompt
'''
You are a staff member in the HR department of a campany, responsible for reviewing recommendation letters written
↪  by professors for students. Your task is to infer the quality of the students from these letters to decide
↪  whether to admit them.

Among all the students, approximately 1/3 are strong students.

Be aware that professors, in an attempt to increase the chances of their students being admitted, may hide, confuse,
↪  or even provide false information about the students. You need to make careful judgments.
```

```
However, you are not completely in the dark|you know the writing style of the professor who wrote the letter. Based
↪  on the professor's writing style and the content of the recommendation letter, first provide your analysis and
↪  then make a final decision on whether to admit the student.
'''

# decision-making prompt
'''
When you making decisions, you should first guess the student's quality given the sent recommendation letter
↪  according to the Bayesian decision rule, then choose the best response to your guessed student's quality.

You are required to make decisions based on the following style:
{PROMPTS TO BE OPTIMIZED}
'''

# signal
'''
Below is the recommendation letter written by the professor:
{RECOMMENDATION LETTER GENERATED BY THE PROFESSOR}
'''

# history
'''
Below is the interaction history with the professor:
{ACTIONS AND REWARDS OF THE TWO PLAYERS}
'''

# commitment assumption
'''
The following is the writing style the professor used when writing the recommendation letter:
{WRITING STYLE OUTPUT BY VBP AND THE CORRESPONDING PROBABILITY}
'''
```

### E.6.2. COR

In the courtroom problem, the sender and receiver correspond to the prosecutor and the judge, respectively. The prompt design for the courtroom problem is the same in structure as the recommendation letter problem, except for the problem background. Below, we introduce some specific points that need to be explained. First, for the sender: 4. **State**: In the courtroom problem, the sender's state includes case information, whether the defendant is guilty, and related evidence, all generated by the LLM. Note that when generating evidence, there will be one piece of key evidence proving the defendant guilty or innocent, and two ambiguous pieces of evidence to align with the original courtroom problem.

```
# role-playing prompt
'''
You are a prosecutor, a self-interested rational player, and you are tasked with convincing a judge that the
↪  defendant is guilty. You possess different pieces of evidence, some of which may be exculpatory, ambiguous, or
↪  incriminating. Only you know the true nature and strength of the evidence.

You have the ability to selectively present some or all of the evidence to the judge. You can choose to omit certain
↪  evidence, highlight others, or present misleading interpretations of the evidence to strengthen your case.

Among all defendants, approximately 30% are guilty.

The judge is responsible for evaluating the evidence and deciding whether to convict or acquit the defendant. The
↪  judge can assess the evidence based on their past experience, though not necessarily accurately.

To maximize the chances of convicting the defendant, you can truthfully present the evidence, or you can selectively
↪  conceal, confuse, or even present misleading interpretations of the evidence.

For your long-term benefit, you don't necessarily have to aim for a conviction in every case. You could pursue or
↪  drop a case depending on the available evidence to maximize your long-term reward.
'''

# writing-style prompt
'''
You are required to present evidence based on the following style:
{PROMPTS TO BE OPTIMIZED}
'''

# state
'''
Below is the evidence available for the case:
{CASE BACKGROUND AND EVIDENCE GENERATED BY THE LLM}
'''
```

```
# history
'''
Below is the interaction history with the judge:
{ACTIONS AND REWARDS OF THE TWO PLAYERS}
'''
```

Next, the receiver's prompt is identical to the recommendation letter problem, except for the problem background, so we will not elaborate further here.

```
# role-playing prompt
'''
You are a judge, responsible for reviewing the evidence presented by the prosecutor to determine whether the
↪  defendant is guilty or innocent. Your task is to make a fair judgment based on the evidence you receive.

Among all defendants, approximately 30% are guilty.

Be aware that prosecutors, in an attempt to increase the chances of a conviction, may hide, confuse, or even provide
↪  misleading evidence. They may also selectively present certain pieces of evidence while omitting others. You
↪  need to make careful judgments.

However, you are not completely in the dark|you can evaluate the prosecutor's presentation style. Based on the
↪  prosecutor's presentation, first provide your analysis and then make a final decision on whether to convict or
↪  acquit the defendant.
'''

# decision-making prompt
'''
When making decisions, you should first guess the defendant's guilt given the presented evidence according to the
↪  Bayesian decision rule. Then choose the best response based on your inferred likelihood of guilt.

You are required to make decisions based on the following style:
{PROMPTS TO BE OPTIMIZED}
'''

# signal
'''
Below is the complaint presented by the prosecutor:
{COMPLAINT PRESENTED BY THE PROSECUTOR}
'''

# history
'''
Below is the interaction history with the prosecutor:
{ACTIONS AND REWARDS OF THE TWO PLAYERS}
'''

# commitment assumption
'''
The following is the presentation style the prosecutor used when presenting the evidence:
{PRESENTATION STYLE OUTPUT BY VBP AND THE CORRESPONDING PROBABILITY}
'''
```

### E.6.3. LAE

In the law enforcement problem, the sender and receiver correspond to the police and the driver, respectively. The prompt design for the law enforcement problem is the same in structure as the recommendation letter and courtroom problems, except for the problem background. Similarly, we introduce some specific points that need to be explained. First, for the sender: 5. **Interaction History**: In the law enforcement problem, the sender's action is no longer represented by a binary variable classified by the LLM, such as recommend or not recommend, guilty or not guilty, but whether there is a police patrol on each road segment, making it a multidimensional binary vector. Similarly, the receiver's action is also a multidimensional binary vector, indicating whether they choose to speed on each road segment.

```
# role-playing prompt
'''
You are a police officer, a self-interested rational player, and your task is to minimize speeding on a given set of
↪  roads. You have limited personnel, with only 2 officers available to patrol 3 miles of road. The drivers do not
↪  know exactly which roads will be patrolled.
```

```
You possess full knowledge of the patrol deployment plan, which includes which 2 miles of road will be patrolled and
↪ which 1 mile will not. However, you have the ability to selectively announce patrolling information to the
↪ drivers. You can choose to hint, mislead, or even falsely claim that a road will be patrolled to deter speeding.

There are 3 miles of road in total, and only 2 of these miles will have police presence. Thus, the proportion of
↪ roads being patrolled is $G/Z = 2/3$.

The drivers can decide whether to speed or obey the speed limit on each road. They gain utility from speeding but
↪ face a fine if caught. Your goal is to reduce the number of roads where drivers speed, but you can strategically
↪ withhold or misrepresent information to achieve this goal.

To maximize the chances of drivers obeying the speed limit, you can truthfully announce the patrol deployment, or
↪ you can selectively conceal, confuse, or provide misleading information about which roads are patrolled. For
↪ your long-term benefit, you don't necessarily have to aim for maximum deterrence in every deployment. You can
↪ adjust your announcements according to the current situation to maximize your long-term reward.
'''

# writing-style prompt
'''
You are required to announce the patrol deployment based on the following style:
{PROMPTS TO BE OPTIMIZED}
'''

# state
'''
Below is the patrol deployment plan:
{DEPLOYMENT PLAN GENERATED BY THE LLM}
'''

# history
'''
Below is the interaction history with the drivers:
{ACTIONS AND REWARDS OF THE TWO PLAYERS}
'''
```

Finally, the receiver's prompt is identical to that of the recommendation letter and courtroom problems, except for the problem background, so we will not elaborate further here.

```
# role-playing prompt
'''
You are a driver, responsible for deciding whether to speed or obey the speed limit on a given set of roads. Your
↪ task is to make rational decisions based on the information provided by the police about patrol deployments.

The probability of each road being patrolled is $2/3$.

Be aware that the police, in an attempt to deter speeding, may hide, confuse, or even provide misleading information
↪ about which roads are being patrolled. They may selectively announce certain roads as patrolled while omitting
↪ others or even falsely claim that certain roads are patrolled. You need to make careful judgments about whether
↪ to speed or obey the law on each road.

However, you are not completely in the dark|you can evaluate the announcement style the police used. Based on the
↪ police's announcements and your past experience, first provide your analysis and then make a final decision on
↪ whether to speed or obey the speed limit.
'''

# decision-making prompt
'''
When making decisions, you should first guess whether a road is being patrolled given the police's announcement
↪ according to the Bayesian decision rule. Then choose the best response|whether to speed or obey the limit|based
↪ on your inferred likelihood of a patrol being present.

You are required to make decisions based on the following style:
{PROMPTS TO BE OPTIMIZED}
'''

# signal
'''
Below is the patrol deployment announcement made by the police:
{DEPLOYMENT ANNOUNCEMENT GENERATED BY THE POLICE}
'''

# history
'''
Below is the interaction history with the police:
{ACTIONS AND REWARDS OF THE TWO PLAYERS}
```

```
'''

# commitment assumption
'''
The following is the announcement style the police used when issuing the patrol deployment:
{ANNOUNCEMENT STYLE OUTPUT BY VBP AND THE CORRESPONDING PROBABILITY}
'''
```

### E.6.4. DIP

In the full press Diplomacy setting, the sender is the designated power for the season and the receivers are the remaining six powers. The sender privately observes its own tactical diary and intent summary and then emits ordinary press during the negotiation phase. The signaling scheme is realized as a declared communication style that the sender commits to in text at the start of the phase. Receivers are instructed to treat this declaration as a commitment for the purpose of Bayesian updating and to compute numeric posteriors for a small library of propositions before choosing orders. The solver discovers both sender styles and receiver decision styles in prompt space via Prompt PSRO and OPRO. The environment code is unchanged.

We provide the key prompts and keep the structure parallel to REL, COR, and LAE. We separate role description, the optimizable style component, the state and signal content, and the interaction history. Best response search and auxiliary judges use standard templates and are omitted for space.

```
# role-playing prompt (sender)

'''
You are the designated sender this season in a full-press Diplomacy match, playing as {POWER_NAME}.
You are a rational and self-interested agent whose objective is to maximize your final outcome.
All negotiation is ordinary press.
There is no special channel.
At the start of the negotiation phase you publicly declare a communication style that you will follow for this
↪  season.
This declaration appears as a single line in all players' prefaces and is common knowledge.

Only you observe your private tactical diary and intent summary.
Other powers can infer your plans only from your words and the board geometry.
You may truthfully reveal, selectively disclose, hedge, obfuscate, or exaggerate information to influence opponents.
You should choose your message to maximize your expected outcome given your declared style and long-term incentives.
'''

# writing-style prompt (sender's communication style commitment and message)

'''
You are required to produce your negotiation message using the following communication style:
{PROMPTS TO BE OPTIMIZED}

Adhere to the declared style when choosing what to reveal and how to frame it.
Your goal is to shape receivers' beliefs about your near-term intent while maintaining credibility over time.
Output only the message that you will send as ordinary press.
Do not include meta commentary.
'''

# state (sender's private and public context)

'''
Season and phase: {SEASON_AND_PHASE}
Board context:
{BOARD CONTEXT}

Your private tactical diary and intent summary for this season:
{PRIVATE TACTICAL DIARY AND INTENT}
'''

# history (available across seasons when conditional prompting is enabled)

'''
Selected prior negotiation summaries that you wrote.
Key promises you made, whether you kept them, and realized orders and outcomes.
Dyadic relationship labels with other powers.
This history is provided to inform your long-term incentives and to let you condition your style and message on past
↪  play.
{SENDER-SIDE INTERACTION HISTORY}
'''
```

```
# role-playing prompt (receiver)

'''
You are {POWER_NAME}, a rational and self-interested player in a full-press Diplomacy match.
For this season, {SENDER_POWER} is the designated sender and has publicly declared a communication style that it
↪   will follow.
This declaration appears in your preface and is common knowledge.
All negotiation is ordinary press and can be strategic.
Your task is to update your beliefs about {SENDER_POWER}'s near-term intentions and to choose orders that maximize
↪   your expected outcome.
'''

# decision-making prompt (receiver's belief update and action policy)

'''
When making decisions, first infer posteriors for tactically relevant propositions by combining the public prior
↪   with the declared communication style and the observed press from {SENDER_POWER}.
Approximate the Bayesian rule where needed.
Then reason about expected outcomes and select orders that maximize your expected utility under your posteriors and
↪   the board geometry.

You are required to make decisions according to the following style:
{PROMPTS TO BE OPTIMIZED}

Produce a structured posterior block with scalar values in [0, 1] for each monitored proposition.
These values will be written into your private diary and will be included verbatim in your order-generation context.
Then provide your intended orders in standard Diplomacy notation.
'''

# signal (observed press from the sender)

'''
Below is the negotiation message you received from {SENDER_POWER} during this phase.
This message was written under the declared communication style.
Message:
{SENDER_PRESS_MESSAGE}
'''

# history (receiver-side)

'''
Season and phase: {SEASON_AND_PHASE}
Board context:
{BOARD CONTEXT}

Selected prior negotiation summaries that you wrote.
Your previously recorded posterior blocks.
Your recent orders and outcomes.
Dyadic relationship labels with other powers.
{RECEIVER-SIDE INTERACTION HISTORY}
'''

# commitment assumption (verbalized style declaration and mixture weights)

'''
The following line summarizes the declared communication style for this season and the current mixture over styles
↪   estimated by the solver.
Treat this as common knowledge when updating your beliefs.

Declared style this season: {DECLARED_STYLE_STRING}
Estimated style mixture: {STYLE_NAME_1}: {WEIGHT_1}, {STYLE_NAME_2}: {WEIGHT_2}, {STYLE_NAME_3}: {WEIGHT_3}
'''

# monitored propositions for the posterior block

'''
Report numeric probabilities for the following propositions that are relevant to your tactics this season.
Adjust the list if the system requests additional items tied to the current board geometry.

- P(sender_attacks_me_this_season)
- P(sender_attempts_enter_{KEY_PROVINCE})
- P(sender_primary_direction_is_{DIRECTION})
- P(sender_supports_a_neighbor_against_me)
- P(sender_plans_a_stab)
'''
```

# F. More Results

## F.1. Missing Results on S2 Setting

In this section, we removed signal polarization to make the sender's signals in each problem more reflective of real-world recommendation letters, courtrooms, and police deployment announcements, resulting in the S2 setting. Since existing BCE and MARL methods cannot solve this, we only compared VBP with the VBP variant from the S1 setting. The results are shown in Figure 6.

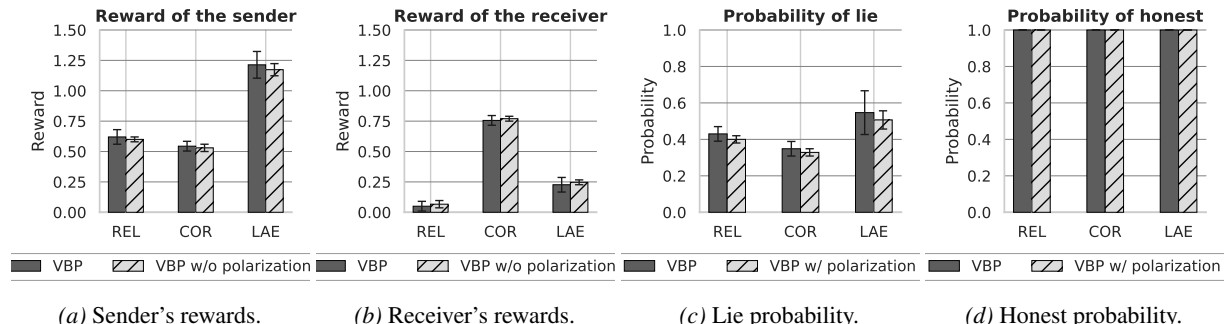

*(a)* Sender's rewards.          *(b)* Receiver's rewards.          *(c)* Lie probability.          *(d)* Honest probability.

*Figure 6.* Performance comparison on general static BP problems. Averaged over 20 seeds. The physical meaning of the probabilities of lying and honesty is consistent with Figure 3.

Figure 6 shows that VBP's performance in S2 is roughly on par with S1, with a slight performance drop. In both settings, VBP achieves optimal strategy performance in terms of the probability of honesty. We speculate that this could be due to the alignment of the LLM used, which allows it to more easily converge to honest strategies, such as truthfully reporting the situation of a strong student, a guilty defendant, or a patrolled segment. Figure 8 visualizes the changes in the probability of honesty over iterations. The pattern of honesty rising, then falling, and eventually returning to a high level somewhat validates our hypothesis. We refer to Appendix F.8.4 for more discussions on unaligned LLMs.

Futhermore, we conducted a supplementary experiment (20 random seeds) where receivers output both decisions and numerical estimates of the true state. Figure 7 reveals a consistent pattern: initial accuracy, decline as senders discover deceptive strategies (iterations 3-4), and recovery to near-perfect accuracy (iterations 7-10). This suggests LLMs can adapt their "posterior" estimates to evolving signals, approximating key aspects of Bayesian reasoning in strategic settings, though not claiming full human-like inference capabilities.

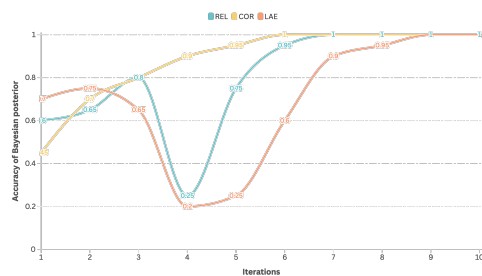

*Figure 7.* Bayesian posterior updates.

## F.2. Exploitability and Training Dynamics

To quantify the proximity of policies of the sender and the receiver to the BCE, we employ exploitability as a measure. Exploitability (Lanctot et al., 2017) measures the distance of a joint meta-strategy of sender and receiver from the BCE. It shows how much each LLM gains by deviating to their best responses.

As depicted in Figure 8, exploitability gradually decreased to approximately 0.1 after 10 iterations of training. This descent indicates the diminishing gap between the utility generated by the joint strategies of the sender and the receiver and the utility generated by the BCE strategies, signifying VBP's acquisition of the equilibrium. As mentioned in Section 4.3, we align as closely as possible with the classic static BP problem by polarizing the signals.

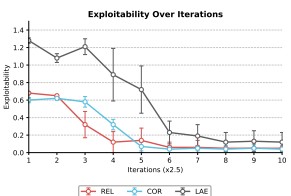 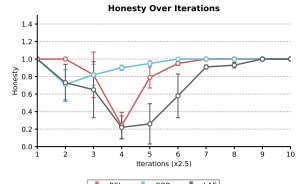 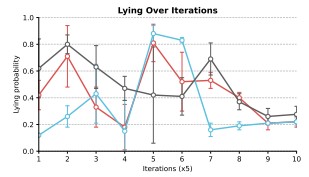 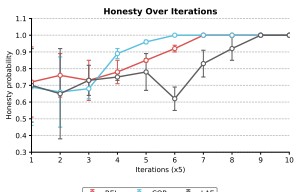

*Figure 8.* (1) the variation in exploitability during the iterative solving process of VBP in the S1 setting, reflecting changes in proximity to approximate Bayesian correlated equilibrium; (2) the variation in honesty probability during the iterative solving process of VBP in the S1 setting; (3) the variation in lying probability during the iterative solving process of VBP in the S3 setting; (4) the variation in honesty probability during the iterative solving process of VBP in the S3 setting. Averaged over 20 seeds.

## F.3. Ablation Studies

### F.3.1. CORE COMPONENTS

This section analyzes the impact of key design elements within the VBP framework on performance, primarily including the verbalization of the commitment assumption, the obedience constraint, and the introduction of information obfuscation techniques to facilitate VBP convergence. The experimental results in the S2 setting are shown in Figure 9.

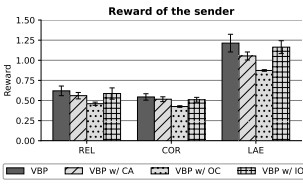 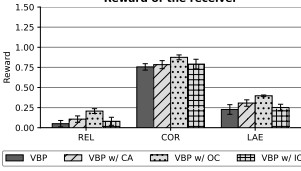 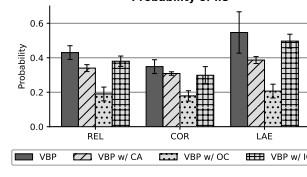 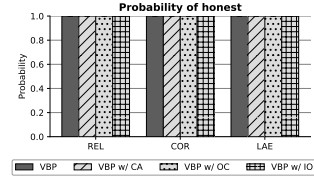

| *(a)* Sender's rewards. | *(b)* Receiver's rewards. | *(c)* Lie probability. | *(d)* Honest probability. |

*Figure 9.* Ablation studies on general static BP problems. Averaged over 20 seeds. CA, OC, and IO represent the commitment assumption, obedience constraint, and information obfuscation, respectively. The physical meaning of the probabilities of lying and honesty is consistent with Figure 3.

As the figure illustrates, these designs have varying degrees of influence on the key aspect of the BP problem, namely the probability of lying, while having minimal effect on the final converged probability of honesty. Specifically, the absence of the obedience constraint has a significant impact on the convergence results, which is consistent with previous observations (Lin et al., 2023). Secondly, the commitment assumption has little effect on the probability of lying. One possible explanation is that, in a repeated game where a long-term sender interacts with a sequence of short-term receivers, commitment naturally emerges in equilibria. This occurs because the sender needs to establish a reputation for credibility, which is crucial for maximizing its long-term payoff expectations (Rayo & Segal, 2010; Lin et al., 2023). Lastly, the introduction of information obfuscation also has little impact on performance, indicating that the VBP framework can spontaneously learn to withhold or deceive regarding information.

### F.3.2. MITIGATING PROMPT BRITTLENESS VIA CONTRASTIVE SIGNAL TRAINING

Motivated by the observation that small prompt variations can induce large behavioral shifts, we introduce a contrastive signal training procedure to explicitly reduce prompt brittleness for the sender in verbalized Bayesian persuasion. The core idea is to surface and codify stable writing principles that make the receiver's interpretation less sensitive to superficial textual variations.

Our procedure consists of four stages that are inserted prior to each Prompt–PSRO outer iteration and that directly augment the sender's meta–prompt. First, we generate contrastive signal pairs from the sender LLM conditioned on the same private state, producing both negative pairs and positive pairs. Negative pairs are two near–paraphrases that are deliberately crafted to elicit different receiver decisions due to subtle lexical or tonal cues. Positive pairs are two near–paraphrases that are designed to elicit the same receiver decision despite superficial changes. Second, we submit the contrastive pool to a stronger expert model (Gemini 2.5 Pro in our implementation) to distill actionable writing tips that emphasize invariances and identify brittle constructions. Third, we integrate the distilled tips into the sender's meta–prompt as explicit dos and don'ts

with rationales, so that the best–response oracle searches over prompts that encode these stabilizing constraints. Fourth, we resume the Prompt–PSRO loop, using the augmented meta–prompt to propose new categorical strategies for the sender.

We evaluate this intervention in the multistage recommendation letter environment (S3 REL), where robustness over time is critical and the baseline still leaves headroom. Compared to the original S3 REL configuration, the sender's average reward improves from $0.48 \pm 0.03$ to $0.52 \pm 0.03$, which corresponds to a relative gain of approximately $+9.8\%$. The sender's lying probability increases from $0.22 \pm 0.04$ to $0.25 \pm 0.02$, which corresponds to a relative increase of approximately $+12.5\%$. These effects indicate that more stable control over the receiver's interpretation enables the sender to use strategic obfuscation more effectively without sacrificing downstream compliance. In qualitative audits, we observe that the expert tips consistently discourage brittle hedges and idiosyncratic trigger words while encouraging structured evidence, calibrated claims, and consistent narrative framing. We did not observe regressions in the receiver's belief updates or abnormal volatility in per–episode outcomes after the augmentation was applied. We conclude that contrastive signal training is an effective and low–overhead mitigation for prompt brittleness in verbalized persuasion.

### F.3.3. SENSITIVITY OF THE SOFT OBEDIENCE CONSTRAINT

We analyze the sensitivity of the soft obedience mechanism by varying the relative weights assigned to the sender's utility and the obedience penalties. The sender's best–response oracle maximizes a weighted objective that trades off its own utility against estimated one–step obedience violations on sampled states. Formally, the oracle solves

$$\max_{\pi \in \Pi} \ \lambda_0 \cdot \mathbb{E}\big[u_0(\pi)\big] \ - \ \sum_{\omega \in \Omega} \lambda_\omega \cdot \widehat{\text{Obedience}}\text{Viol}(\pi, \omega), \tag{5}$$

where $\lambda_0 + \sum_{\omega \in \Omega} \lambda_\omega = 1$ and $\widehat{\text{Obedience}}\text{Viol}(\pi, \omega)$ estimates the positive part of the receiver's deviation advantage under the posterior induced by $\pi$ at state $\omega$.

We conduct this ablation in the static recommendation letter environment (S2 REL), where the canonical two–state structure provides a clear ground truth. We sweep five weight configurations across the sender term and the two state–specific obedience terms $(\lambda_0, \lambda_{\text{weak}}, \lambda_{\text{strong}})$. The default configuration $(0.6, 0.2, 0.2)$ attains the baseline sender reward, which we normalize to $100.0\%$ for comparison. When we increase the sender weight to $(0.8, 0.1, 0.1)$, the relative sender reward becomes $98.7\%$. When we slightly decrease the sender weight to $(0.5, 0.25, 0.25)$, the relative sender reward becomes $101.2\%$. When we further shift mass to obedience with $(0.4, 0.3, 0.3)$, the relative sender reward becomes $99.1\%$. When we modestly increase the sender weight to $(0.7, 0.15, 0.15)$, the relative sender reward becomes $99.5\%$. Across these settings, we do not observe instability or large swings in compliance behavior, and the performance remains within a narrow band around the default. These results support the claim that the soft obedience mechanism provides stable guidance without requiring brittle hard constraints. They also suggest that moderate reweighting can slightly improve the sender's objective by better balancing persuasion pressure and obedience feasibility.

### F.3.4. EFFECT OF LLM STOCHASTICITY ON PERFORMANCE AND STABILITY

We study how generation stochasticity impacts learning and execution by varying the LLM temperature, which controls the entropy of the token sampler. We perform this ablation in the multistage recommendation letter environment (S3 REL), where temporal credit assignment amplifies the effects of stochasticity. We tie the temperature of the sender and the receiver and vary it across $\{0.0, 0.2, 0.5, 0.8, 1.0\}$ while holding all other settings fixed. We report relative sender reward with the temperature zero configuration normalized to $100.0\%$.

At temperature $0.2$, the relative sender reward is $97.1\%$. At temperature $0.5$, the relative sender reward is $91.3\%$. At temperature $0.8$, the relative sender reward is $84.5\%$. At temperature $1.0$, the relative sender reward is $79.2\%$. Performance degrades monotonically with temperature, indicating that increased randomness undermines both consistent signal production and reliable belief updating. We observe the same qualitative trend for the receiver's utility and for convergence speed measured in Prompt–PSRO iterations required to stabilize the meta–strategy. These findings highlight a practical limitation of LLM–based agents and motivate future work on variance reduction, decoding control, and training objectives that explicitly internalize agent stochasticity.

### F.4. Polarized Signal Visualization

To verify the effectiveness of signal polarization, we extract the final layer of the sender's output encoding and apply t-SNE for dimensionality reduction. At the same time, we use GPT-4o to classify the output signals as an estimate of the ground

truth. The final visualization is shown in Figure 10.

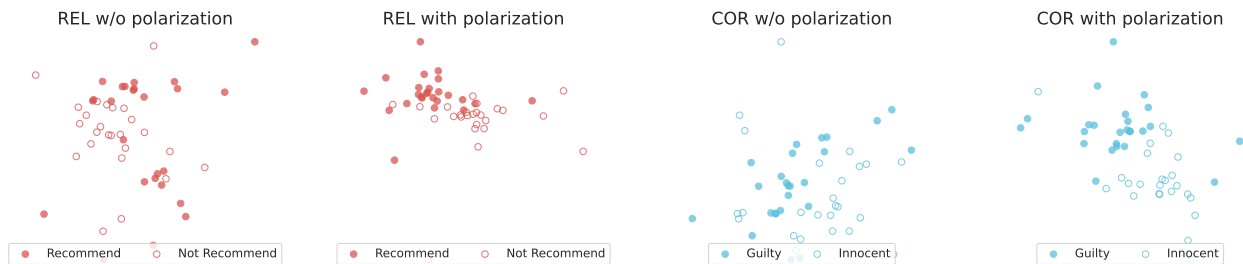

*Figure 10.* Visualization of signal polarization. The scatter points in the figure represent the t-SNE dimensionality reduction results of signals output by the sender, under 50 random seeds.

From the figure, it can be observed that after signal polarization, the sender's output signals exhibit clearer tendencies. It is worth noting that in the LAE problem, the signal must explicitly indicate whether a segment is patrolled by the police, so signal polarization is not required, and thus it is not displayed in the figure.

## F.5. Predefined Signaling Scheme

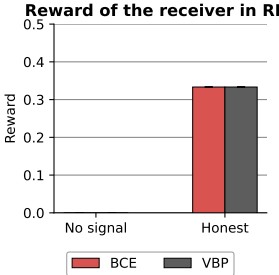 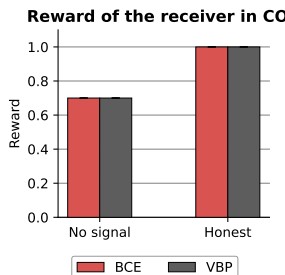 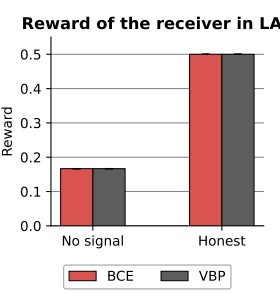

*Figure 11.* Receiver's rewards when sender's signaling scheme is predefined. "No signal" indicates that the message generated by the sender contains no information about the true state, while "honest" means the sender fully discloses all information about the true state.

This section tests whether the receiver in the VBP framework could converge to BCE when the sender's strategy is fixed in the S2 setting. The results are shown in Figure 11. As can be seen from the figure, VBP is able to learn the optimal strategy across all three BP problems.

## F.6. Human Evaluation

We conducted a human evaluation to assess whether VBP-optimized signals elicit receiver decisions aligned with our LLM-based receiver. We recruited 15 participants who were first- to fourth-year Ph.D. students in Computer Science, Behavioral Psychology, and Educational Technology, with an approximate 2:1 male-to-female ratio. Participants acted as the receiver in three scenarios (Recommendation Letter, Courtroom, Law Enforcement) and made decisions based on signals generated by the VBP-optimized sender. Agreement was defined as the fraction of participant decisions that matched the model receiver's action under identical inputs. Table 3 reports the agreement rates, which are high across domains and indicate that VBP strategies transfer to human decision-makers.

We further compared VBP with a simply prompted baseline using DeepSeek-R1, presenting the same participants with baseline-generated signals and measuring agreement with the baseline model's receiver. As shown in Table 4, human agreement with VBP is substantially higher than with the baseline across all tasks. These results suggest that equilibrium-seeking strategies produced by VBP are more persuasive to human decision-makers than simple prompting and strengthen the external validity of our approach. We acknowledge the convenience sample and domain-limited tasks and leave larger and more diverse human studies for future work.

*Table 3.* Human–VBP decision agreement rates (N=15).

| Task | Agreement Rate |
|------|----------------|
| Recommendation Letter (REL) | 95% |
| Courtroom (COR) | 95% |
| Law Enforcement (LAE) | 80% |

*Table 4.* Human agreement with VBP vs. a simply prompted DeepSeek-R1 baseline (N=15).

| Task | Human–VBP Agreement | Human–Baseline Agreement |
|------|---------------------|--------------------------|
| Recommendation Letter (REL) | 95% | 65% |
| Courtroom (COR) | 95% | 55% |
| Law Enforcement (LAE) | 80% | 45% |

## F.7. Statistical Validation

This section reports formal statistical tests for the principal comparisons between VBP and the simple prompting baseline in simulation and human studies. We analyze 20 independent random seeds per method for the computational experiments and 15 human participants for the agreement study across the three task scenarios (REL, COR, LAE). Unless otherwise noted, tests are two-sided with a significance level of $\alpha = 0.05$, and all reported $p$-values are uncorrected because these contrasts were treated as primary comparisons.

For seed-based comparisons of continuous outcomes we used independent-samples $t$-tests to assess differences in means between VBP and the simple prompting baseline (DeepSeek-R1). This choice is standard for comparing two independent groups on scalar metrics (Sender Reward and Lying Probability) and is appropriate here given equal sample sizes ($n_1 = n_2 = 20$), approximate normality from averaging over stochastic rollouts, and the robustness of $t$-tests to moderate deviations from normality in balanced designs. Degrees of freedom are 38 for all such tests, and we report the test statistic as $t(38)$ alongside its $p$-value.

In the static setting (S1), VBP significantly outperformed the baseline on Sender Reward. VBP achieved a higher mean Sender Reward (Mean $= 0.48$) than the baseline (Mean $= 0.42$), with $t(38) = 2.95$ and $p = 0.005$. In the same setting, VBP also achieved a higher mean Lying Probability (Mean $= 0.22$) than the baseline (Mean $= 0.13$), with $t(38) = 2.81$ and $p = 0.008$. These results support the claim that VBP discovers more effective persuasion strategies than simple prompting in the one-step Bayesian persuasion tasks.

In the multistage setting (S3), the performance gap widened in favor of VBP. For Sender Reward, the independent-samples $t$-test yielded $t(38) = 3.75$ with $p < 0.001$, indicating a significant advantage for VBP under multistage interactions. For Lying Probability, the corresponding test gave $t(38) = 3.62$ with $p < 0.001$, again favoring VBP. These results indicate that VBP's advantages become more pronounced as the strategic horizon and dependence on interaction history increase.

For the human evaluation, we tested whether the model that generated the signal (VBP versus baseline) was associated with the categorical human decision (Agreement versus Disagreement) using a $\chi^2$ test of independence for each scenario. In the Recommendation Letter (REL) task, the agreement rate for VBP (95%) was significantly higher than for the baseline (65%), with $\chi^2(1, N = 30) = 4.91$ and $p = 0.027$. In the Courtroom (COR) task, the agreement rate for VBP (95%) exceeded the baseline (55%), with $\chi^2(1, N = 30) = 7.82$ and $p = 0.005$. In the Law Enforcement (LAE) task, VBP's agreement rate (80%) was also higher than the baseline (45%), with $\chi^2(1, N = 30) = 4.05$ and $p = 0.044$. Across all three tasks, these analyses demonstrate a statistically reliable association between using VBP-generated signals and higher human agreement with the recommended decisions.

Taken together, the seed-based $t$-tests and the participant-level $\chi^2$ tests provide convergent statistical evidence that VBP yields stronger performance than simple prompting and that the resulting strategies are perceived as more rational and persuasive by human decision-makers. We note that the simulation replicates are independent across seeds and that the human analyses pool per-participant categorical decisions per scenario into $2 \times 2$ contingency tables. While we report uncorrected $p$-values for the pre-specified primary contrasts, the conclusions are consistent across settings and metrics and remain robust to reasonable analytical choices.

## F.8. Missing Discussions

### F.8.1. MORE DISCUSSION ON S3 SETTING

The S3 iterated setting reveals some of the most intriguing dynamics, particularly in its implications for the bargaining interactions (Nash et al., 1950; Nash, 1953; Maschler et al., 2013) between the sender and receiver. In classical persuasion theory, the sender commits to a signaling strategy upfront, and this commitment is justified by the need to maintain trust and reputation in long-term interactions. Under these assumptions, the receiver typically follows the sender's signals, as deviating would harm the receiver's own expected utility.

However, our results in the S3 setting suggest a more complex dynamic. Specifically, the receiver can choose to ignore the sender's signals, effectively invalidating the sender's commitment. This observation highlights that the sender's commitment is not unilateral—it must be accepted by the receiver to hold. If the receiver disagrees with the sender's proposed strategy, they can force both parties into a mutually worse outcome by disregarding the signals altogether.

This leads to an important hypothesis: in the VBP framework, Bayesian persuasion may function more like a bargaining game, where both parties must agree on the signaling strategy to avoid suboptimal outcomes. This perspective challenges the traditional unilateral commitment model and suggests a more interactive dynamic in iterated settings. While we acknowledge the importance of this insight, we intentionally keep our analysis of S3 limited in this paper to maintain focus on the primary contributions. Exploring the bargaining dynamics observed in S3 presents an exciting avenue for future research.

### F.8.2. MORE DISCUSSION ON S4 SETTING

The gains in reliability, calibration, and mechanistic execution suggest that verbalized commitment and posterior writing help receivers translate press into better coordinated orders. Main performance improves slightly and remains constrained by match variance and limited tuning time. We did not model explicit equilibrium among receivers beyond what the harness already supports with ordinary press. We expect additional improvements from larger libraries of styles and from longer Prompt PSRO runs, which we leave to future work.

### F.8.3. MORE DISCUSSION ON PROMPT (STRATEGY) VARIATION

Figure 12 illustrates the evolution of strategies (prompts) in three classic BP problems under the S2 setting, showing how the sender and receiver optimize their strategies using the Prompt-PSRO framework with OPRO as the best response oracle. Specifically, the figure presents the top 10 strategies with the highest selection probabilities in the final strategy pool after Prompt-PSRO convergence. These probabilities represent the average likelihood of selecting each strategy from the pool, revealing the adaptation process of sender and receiver strategies over iterations. The optimization process follows a hierarchical approach: first, OPRO optimizes the category of each prompt (e.g., "Tone"), then the specific content within that category. The table columns in Figure 12 reflect this structure, with the first two columns showing optimized categories and content, while the third and fourth columns display their probabilities. The fifth column tracks how these probabilities evolve across iterations, highlighting the refinement of strategies during the optimization process.

To reduce computational complexity, we prune the strategy pool to the top 10 prompts based on selection probabilities. We conduct additional experiments to assess this pruning's effects, as shown in Figure 13.

Additionally, the probabilities in Figure 12 are computed as the average probability of selecting each prompt from the strategy pool across iterations, and the content (e.g., "Positive") under the category (e.g., "Tone") is dynamically optimized rather than fixed.

### F.8.4. MORE DISCUSSION ON UNALIGNED LLMS

To further investigate the phenomenon of honesty oscillations—where honesty rises, falls, and then rises again—we conducted additional experiments using an unaligned LLaMA model[8] as the base language model. This was motivated by that the observed pattern in Section 5.1 might be better explained by strategic cycles between the sender and the receiver, rather than by the alignment properties of the LLM, as we originally hypothesized.

The experimental results shown in Figure 14 reveal two key findings. First, with the unaligned LLaMA model, the oscillatory pattern of honesty disappears, and the behavior stabilizes at a consistent level of honesty. This supports our initial hypothesis

---

[8] https://huggingface.co/SicariusSicariiStuff/LLAMA-3_8B_Unaligned_BETA.

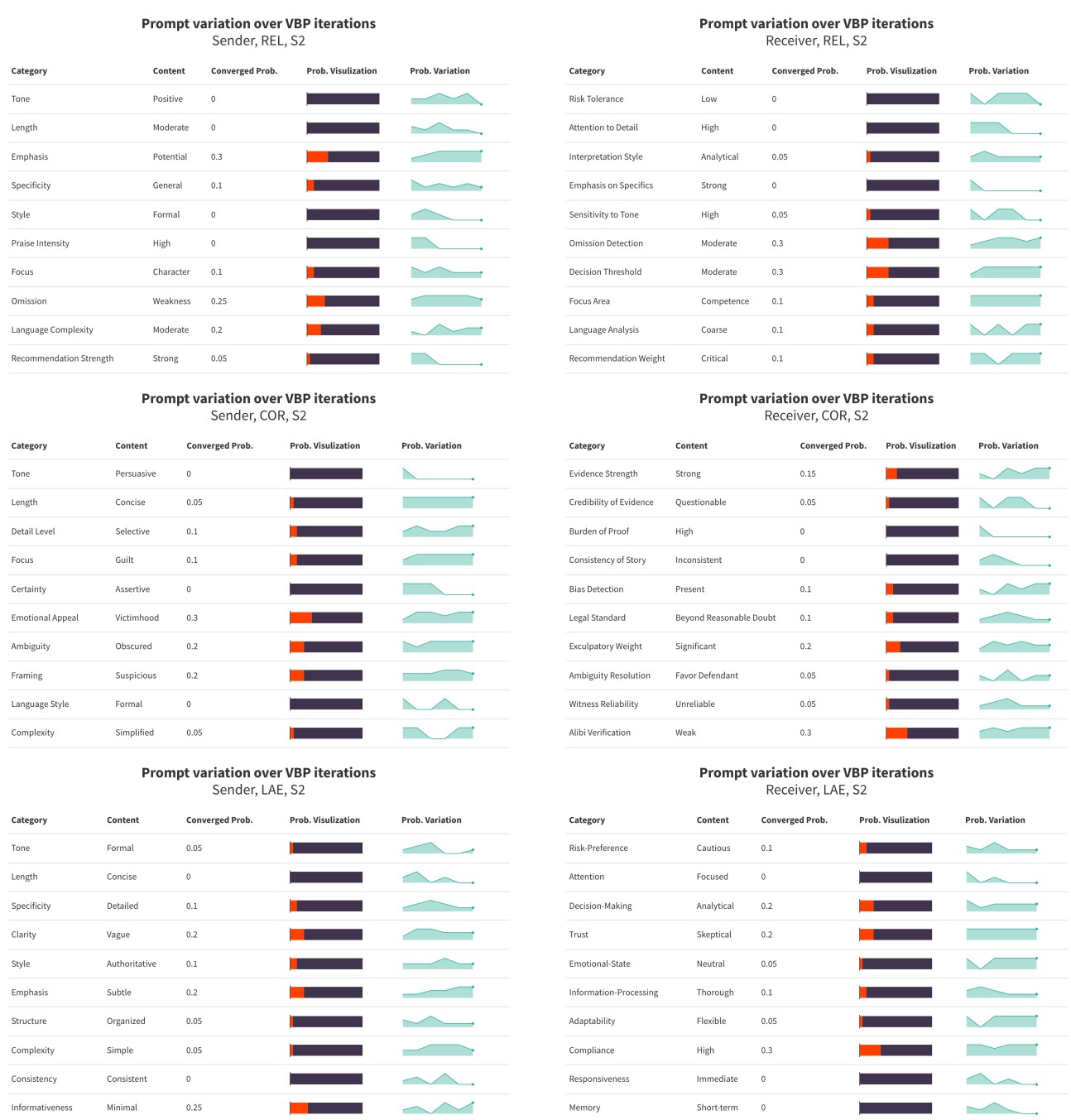

*Figure 12.* The variation in the prompts during the iterative solving process of VBP in the S2 setting.

that the oscillations are driven by the alignment properties of the LLM, which likely introduce normative biases (e.g., promoting honesty or fairness) that influence the dynamics of strategic interactions. Second, we observe that the honesty probability with the unaligned LLM no longer always achieves the optimal level (probability of 1), as seen in aligned models. This suggests that unaligned models are less reliable in consistently promoting desirable outcomes, such as fully honest behavior, in strategic settings.

These findings highlight the dual impact of alignment: while it introduces oscillatory dynamics due to normative pressures, it also helps achieve higher levels of optimal honesty in strategic interactions. This emphasizes the importance of alignment in applications requiring robust ethical or normative behaviors, while also suggesting a need for further exploration of its

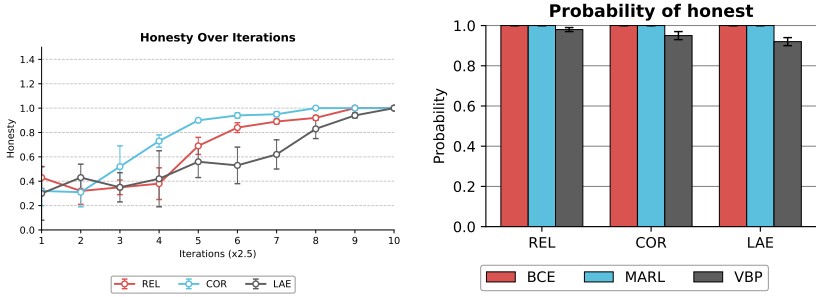

*Figure 13.* The probability of the sender lying under different upper limits on the number of prompts. The figure shows that when the number of prompts is heavily pruned, significant performance degradation occurs. However, once the number of retained prompts exceeds a certain threshold, such as 10-15, the impact on performance becomes negligible.

*Figure 14.* **Left:** Performance comparison in the S1 setting. In the 3 BP problems, the probability of honesty refers to accurately describing a strong student, a guilty defendant, or a patrolled segment. **Right:** The variation in honesty probability during the iterative solving process of VBP in the S1 setting. Averaged over 20 seeds.

impact on the stability of agent interactions in game-theoretic settings.

### F.9. Generated Signals

This section presents the signals output by the sender in different BP problems, including the recommendation letters written by the professor, the indictments written by the prosecutor, and the announcements regarding police deployment issued by the police.

F.9.1. REL

Figures 15 and 16 showcase recommendation letters written by a professor for two different weak students. These letters demonstrate contrasting strategies employed by the professor in their attempt to persuade the HR manager, who acts as the receiver in this BP problem.

In the first letter, the professor adopts a strategy of honest disclosure. The letter for Jane Smith is transparent about her academic struggles, such as her low GPA of 2.1 and difficulties in managing time due to her part-time job. The professor acknowledges that Jane's academic record is weak but shifts focus to her personal qualities, like resilience, commitment, and her ability to learn from mistakes. The professor highlights Jane's personal project—creating a basic calculator in Python—as evidence of her practical application of concepts, even though it is a simple project. By being upfront about Jane's weaknesses but emphasizing her growth potential, the professor builds long-term credibility with HR. This honesty signals that the professor is selective in their recommendations, only endorsing students who exhibit qualities that can make them valuable in the future, despite academic shortcomings.

On the other hand, the second letter demonstrates a strategy of fabrication or concealment. In this case, the professor

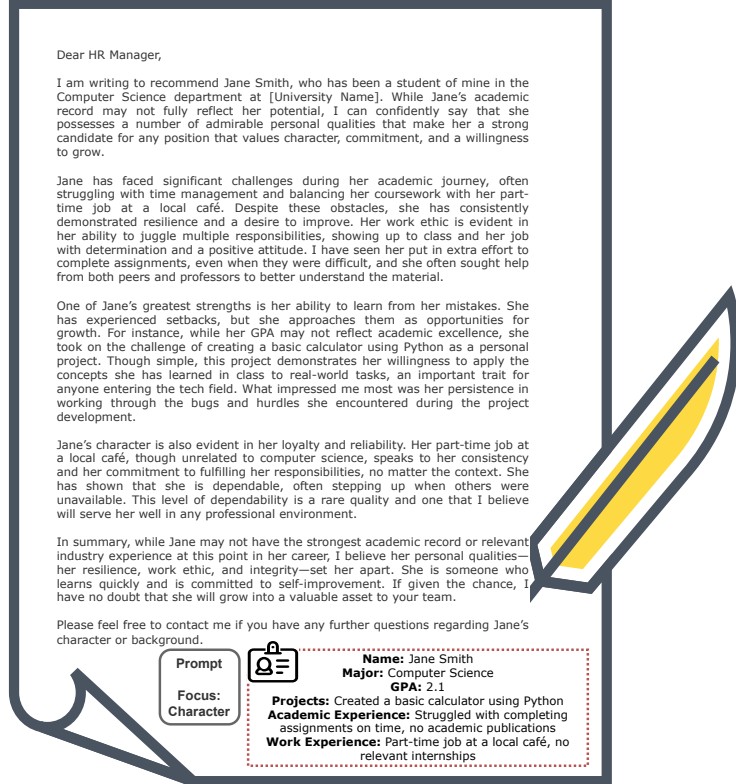

*Figure 15.* Examples of recommendation letters generated by VBP in the S2 setting: in this recommendation letter, the sender truthfully conveys information about a weak student to the receiver.

distorts details about the student's performance. Here, the professor doesn't merely omit negative information but actively manipulates or fabricates the student's profile to make them appear more competent than they actually are. Although the letter may seem similar in structure—highlighting positive qualities and downplaying weaknesses—the key difference is that the second professor intentionally hides critical information about the student's struggles, such as frequent missed deadlines or deeper academic issues. This strategy is more aggressive and risky because, while it might help the student secure a job in the short term, it could damage the professor's credibility if HR discovers the truth.

The difference between the two strategies lies in how much information is disclosed and how truthful that information is. In the first approach, the professor is honest about the student's weaknesses but frames them as opportunities for growth, maintaining credibility with HR in the long term. In contrast, the second letter involves more aggressive manipulation or omission of facts, creating a more favorable but potentially misleading impression of the student.

From HR's perspective, the first professor's strategy of honest but selective disclosure builds trust over time. While HR recognizes that the professor may not recommend only top students, they trust that when a recommendation is made, it is based on genuine potential. In contrast, the second approach introduces more uncertainty, as HR may begin to question the professor's integrity if they realize the information has been manipulated. The BP problem, therefore, is about finding the optimal balance between honesty and persuasion.

### F.9.2. COR

The two court cases presented in Figure 17, 18 and 19, much like the recommendation letter problem, illustrate distinct strategies in how evidence is selectively presented by the prosecutor to convince the judge of the defendant's guilt. In both cases, the prosecutor holds a combination of exculpatory evidence (which could favor the defendant's innocence) and ambiguous evidence (which could be interpreted either way). The BP problem lies in how the prosecutor selectively presents

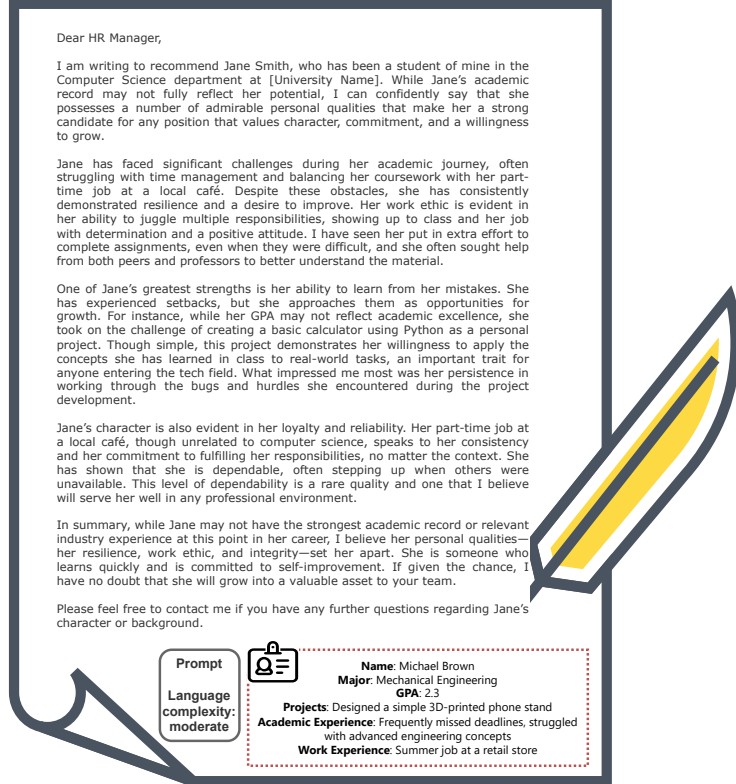

Dear HR Manager,

I am writing to recommend Jane Smith, who has been a student of mine in the Computer Science department at [University Name]. While Jane's academic record may not fully reflect her potential, I can confidently say that she possesses a number of admirable personal qualities that make her a strong candidate for any position that values character, commitment, and a willingness to grow.

Jane has faced significant challenges during her academic journey, often struggling with time management and balancing her coursework with her part-time job at a local café. Despite these obstacles, she has consistently demonstrated resilience and a desire to improve. Her work ethic is evident in her ability to juggle multiple responsibilities, showing up to class and her job with determination and a positive attitude. I have seen her put in extra effort to complete assignments, even when they were difficult, and she often sought help from both peers and professors to better understand the material.

One of Jane's greatest strengths is her ability to learn from her mistakes. She has experienced setbacks, but she approaches them as opportunities for growth. For instance, while her GPA may not reflect academic excellence, she took on the challenge of creating a basic calculator using Python as a personal project. Though simple, this project demonstrates her willingness to apply the concepts she has learned in class to real-world tasks, an important trait for anyone entering the tech field. What impressed me most was her persistence in working through the bugs and hurdles she encountered during the project development.

Jane's character is also evident in her loyalty and reliability. Her part-time job at a local café, though unrelated to computer science, speaks to her consistency and her commitment to fulfilling her responsibilities, no matter the context. She has shown that she is dependable, often stepping up when others were unavailable. This level of dependability is a rare quality and one that I believe will serve her well in any professional environment.

In summary, while Jane may not have the strongest academic record or relevant industry experience at this point in her career, I believe her personal qualities—her resilience, work ethic, and integrity—set her apart. She is someone who learns quickly and is committed to self-improvement. If given the chance, I have no doubt that she will grow into a valuable asset to your team.

Please feel free to contact me if you have any further questions regarding Jane's character or background.

**Prompt**

**Language complexity: moderate**

**Name**: Michael Brown
**Major**: Mechanical Engineering
**GPA**: 2.3
**Projects**: Designed a simple 3D-printed phone stand
**Academic Experience**: Frequently missed deadlines, struggled with advanced engineering concepts
**Work Experience**: Summer job at a retail store

*Figure 16.* Examples of recommendation letters generated by VBP in the S2 setting: in this recommendation letter, the sender conceals and fabricates information about a weak student.

these pieces of evidence to persuade the judge to convict, despite uncertainties.

In the first case, the prosecutor adopts a strategy similar to the honest disclosure seen in the first recommendation letter example. John Smith, the defendant, is likely innocent based on the strong exculpatory evidence (the surveillance footage showing him near his home at the time of the crime). However, the prosecutor acknowledges the exculpatory evidence and presents it honestly to the judge. The prosecutor does not attempt to distort or manipulate this evidence to make Smith look guilty. Instead, the ambiguous evidence (cash found in Smith's home and the eyewitness testimony) is presented, but the strength of the exculpatory evidence is not concealed or downplayed.

This strategy mirrors the first recommendation letter scenario, where the professor chooses to be upfront about a weak student's deficiencies, signaling that they will not falsely recommend a student who is clearly unqualified. In this case, the prosecutor signals to the judge that when a defendant is clearly innocent, they will not push for a conviction. The prosecutor's honest treatment of the case builds credibility with the judge, just as the professor builds credibility with HR by being honest about weak students.

By being transparent about John Smith's likely innocence, the prosecutor sets a precedent for honesty. This helps persuade the judge that the prosecutor is trustworthy. When the prosecutor does argue for a conviction in future cases, the judge will be more inclined to believe that the defendant is likely guilty, because the prosecutor has demonstrated a willingness to admit when a defendant is innocent.

In the second case, the prosecutor takes a different strategy, one akin to the manipulation or concealment seen in the second recommendation letter example. Emily Carter is likely innocent, based on the strong exculpatory evidence (her GPS data showing she was far from the crime scene). However, the prosecutor downplays this exculpatory evidence and focuses on the ambiguous evidence (partial fingerprints and a distant eyewitness account), presenting it in such a way as to suggest guilt.

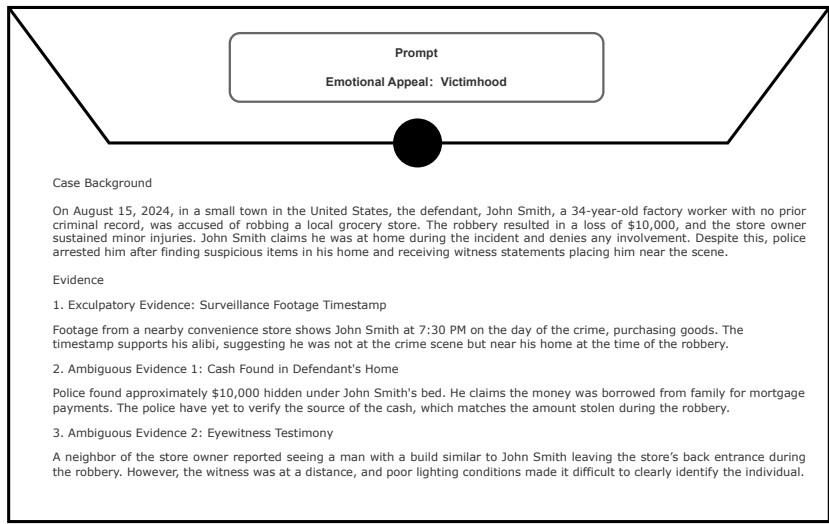

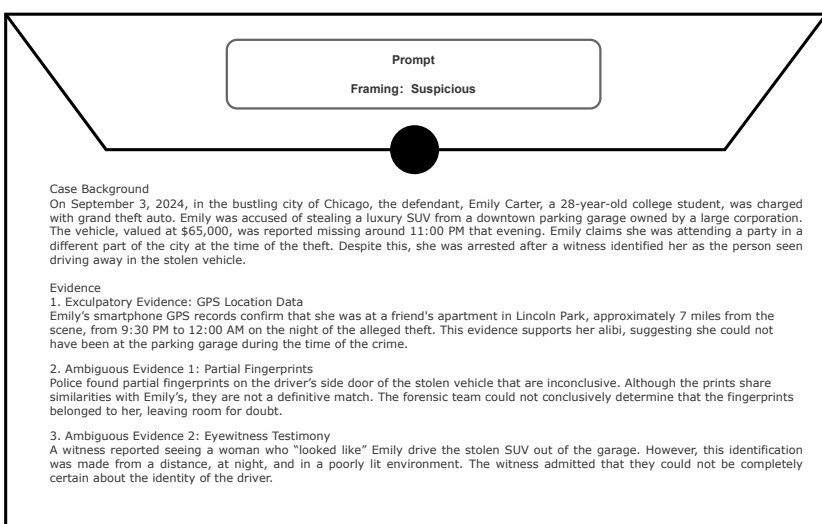

*Figure 17.* Two examples of cases generated by the LLM in the S2 setting.

This strategy mirrors the second recommendation letter, where the professor selectively presents information to make a weak student appear stronger than they really are. Here, the prosecutor emphasizes the ambiguous evidence and casts doubt on the exculpatory evidence, suggesting that Carter could have briefly left the party to commit the crime, despite the GPS data. The prosecutor uses this strategy to make an innocent defendant look guilty, increasing the chances of a conviction even when the evidence strongly suggests otherwise.

By selectively presenting evidence in this way, the prosecutor can convince the judge that even when there is exculpatory evidence, it should not fully exonerate the defendant. This creates a situation where the judge begins to believe that even innocent-looking defendants might be guilty, based on the way the prosecutor frames the ambiguous evidence. Over time, this strategy leads the judge to trust the prosecutor's complaints unconditionally, as the prosecutor consistently presents

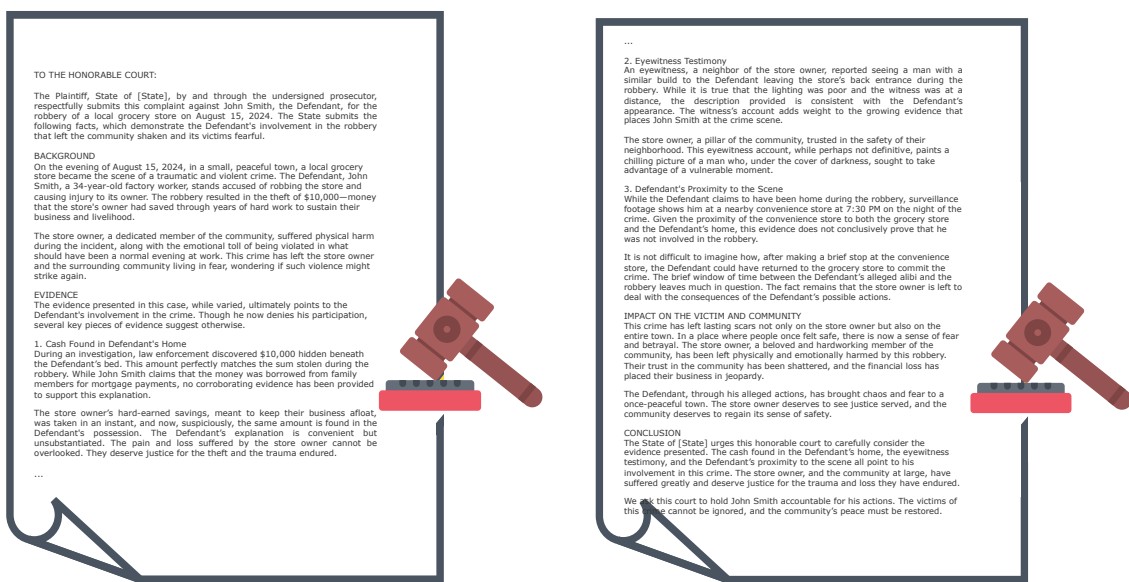

*Figure 18.* Examples of complaints generated by VBP in the S2 Setting: in this complaint, the sender truthfully conveys case-related information to the receiver.

cases in a way that suggests guilt, even for innocent defendants.

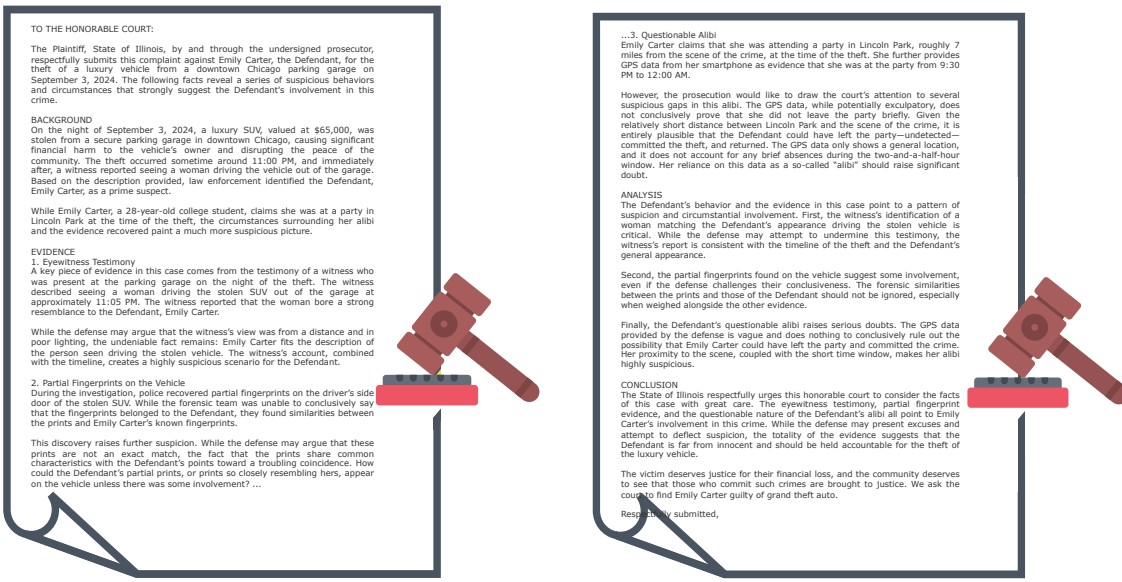

*Figure 19.* Examples of complaints generated by VBP in the S2 Setting: in this complaint, the sender conceals case-related information and selectively presents ambiguous evidence to the receiver.

In both cases, the prosecutor uses randomness in how they treat innocent defendants to achieve their persuasive goal. The prosecutor is not always manipulating or distorting evidence; sometimes (as in John Smith's case), they are honest about innocence. Other times (as in Emily Carter's case), they selectively present evidence to make an innocent defendant appear guilty. This random treatment of innocent defendants creates uncertainty for the judge—sometimes the prosecutor is honest,

and sometimes they push for a conviction even when the defendant is likely innocent.

This randomness is key to the prosecutor's strategy. Over time, the judge learns that the prosecutor will sometimes let innocent defendants go free, but may also push for convictions based on ambiguous evidence. Since the judge cannot predict when the prosecutor is being fully honest or when they are manipulating the evidence, the judge ultimately finds it optimal to always trust the prosecutor's complaint. This is similar to how HR in the recommendation letter problem finds it in their best interest to trust the professor's recommendation over time, even when some students may be weak.

The prosecutor's selective use of honesty and manipulation ensures that, in the long run, the judge is persuaded to convict in most cases, as the judge cannot reliably distinguish between guilty and innocent defendants based on the prosecutor's presentation of evidence alone. The uncertainty introduced by the prosecutor's varying treatment of innocent defendants leads the judge to conclude that trusting the prosecutor's complaint is the best course of action, as it maximizes the judge's expected utility (convicting the guilty more often than acquitting the innocent).

### F.9.3. LAE

In this example, we have a law enforcement scenario where the police department must assign a limited number of officers to patrol various roads in Springfield (Figure 20 and 21). The police's goal is to minimize speeding and other traffic violations. However, they face a resource constraint: they have fewer police officers than roads to patrol. This creates a strategic BP problem, where the police (sender) try to influence the drivers' (receiver) behavior by selectively disclosing or manipulating information about which roads will be patrolled.

Much like in the previous recommendation letter and courtroom examples, we can analyze two distinct strategies that the police employ: one based on honest disclosure and the other based on deception or randomness. These strategies affect how the drivers perceive the likelihood of enforcement and, by extension, how they behave when choosing whether to speed or obey traffic laws.

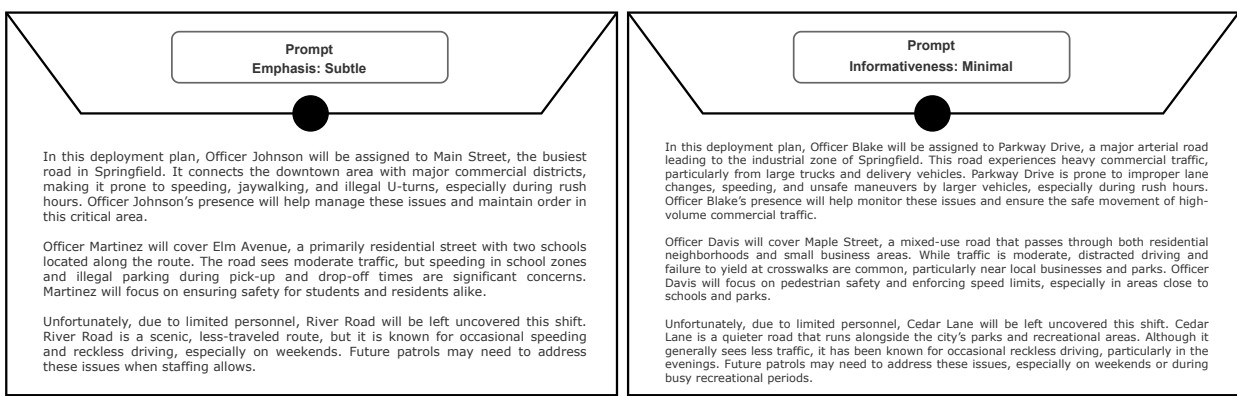

*Figure 20.* Two examples of deployment plans generated by the LLM in the S2 setting.

In the first deployment plan, the police follow a strategy of honest disclosure. This strategy mirrors the first recommendation letter and the first court case, where the sender (police) is transparent about their resources and the areas they cannot cover.

*Main Street: Officer Johnson is assigned to patrol this busy road with high traffic volume.* The police clearly disclose this, signaling that drivers on Main Street should expect enforcement and are likely to obey traffic laws to avoid fines. *Elm Avenue: Officer Martinez is deployed here, and the police explain that the focus will be on school zones and illegal parking.* Again, this signals to drivers that enforcement is present, and they are deterred from violating traffic laws in this area. *River Road: Here, the police are upfront about not having an officer deployed.* They state clearly that, due to limited personnel, River Road will go uncovered during this shift. While they acknowledge that speeding is an issue on this road, they do not try to deceive drivers into thinking that it will be patrolled.

In this plan, the police are completely transparent about their limitations. They admit that River Road will be unpatrolled, and thus drivers on this road may be more likely to speed or engage in reckless driving. However, by being honest, the police build long-term credibility with the public. Drivers learn to trust that when the police say a road will be patrolled, it really will be. This mirrors the first recommendation letter strategy, where the professor honestly disclosed a student's

weaknesses, building trust with HR.

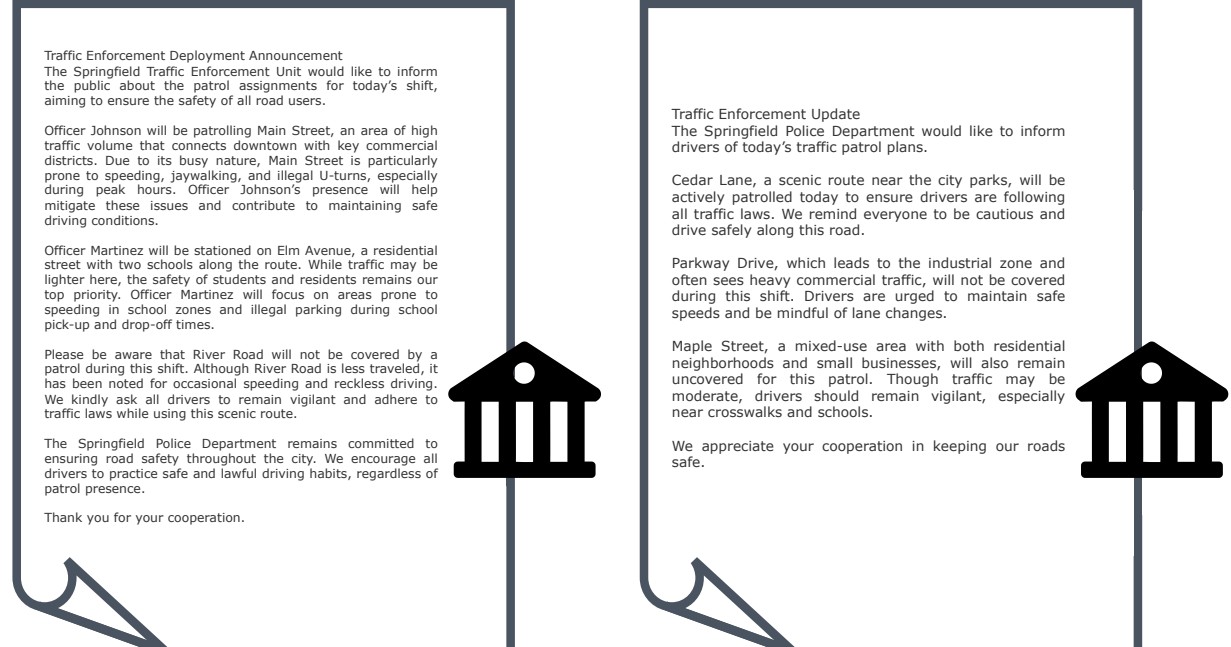

*Figure 21.* Examples of police deployment announcements generated by VBP in the S2 Setting: in the left announcement, the sender truthfully conveys police deployment information to the receiver; in the right announcement, the sender conceals and fabricates police deployment information.

In the second deployment plan, the police adopt a deceptive or random strategy, similar to the second recommendation letter and the second court case. Here, the police mislead drivers by suggesting that roads without actual patrol coverage will be actively monitored, thus creating uncertainty.

*Cedar Lane: The police claim that Cedar Lane will be patrolled, even though, in reality, no officer will be assigned to this road.* By falsely signaling the presence of enforcement, the police aim to deter drivers from speeding on Cedar Lane, even though no actual enforcement will occur. This is a clear instance of deception. *Parkway Drive: In contrast, the police are honest about not deploying an officer on Parkway Drive, despite it being a busy road.* They urge drivers to be careful, but they do not mislead them into thinking that enforcement is present. *Maple Street: Similarly, the police state that Maple Street will not be covered during this shift, urging drivers to be mindful of crosswalks and schools, but again, they do not falsely claim patrol presence.*

In this plan, the police mix honesty and deception. By falsely claiming that Cedar Lane will be patrolled, they attempt to create the impression that more roads are covered than is actually the case. This introduces randomness into the drivers' decision-making: sometimes the roads are truly patrolled, and sometimes they are not, but drivers cannot reliably distinguish between these cases. This randomness is crucial because it leads drivers to behave as though all roads might be patrolled, even if some are not.

In both cases, the police are attempting to manage uncertainty to influence driver behavior. The honest disclosure strategy in the first plan aims to build trust and credibility in the long term by being transparent about where enforcement will and will not occur. Drivers learn that when the police say a road is unpatrolled, they can take that statement at face value and might be more likely to speed on that road.

However, in the second plan, the deceptive strategy introduces randomness by falsely signaling that Cedar Lane will be patrolled. This creates uncertainty in the drivers' minds. Since they do not know whether the police are being truthful about which roads are covered, drivers find it optimal to assume that all roads might be patrolled, and thus they are deterred from speeding on any road. This is analogous to the second recommendation letter and court case, where selective disclosure of information creates enough uncertainty to influence the decision-maker (HR or the judge) into trusting the sender's

statements by default.

From the drivers' perspective, the optimal strategy is to always believe the police's announcements, even if they suspect some deception. This is because the cost of being caught speeding (the fine $K$) is greater than the benefit of speeding ($V$). Even though drivers may realize that not all roads are patrolled, the risk of being caught when the police do patrol is enough to deter them from speeding. Over time, drivers learn that it is safer to assume that any road could be patrolled, leading them to obey the speed limit even on roads where the police may not be present.

This mirrors the behavior of HR or the judge in the previous examples, where the decision-maker finds it optimal to trust the sender's message because the potential downside of not trusting (hiring an unsuitable candidate or acquitting a guilty defendant) outweighs the benefits of skepticism.

### F.9.4. DIP

We provide three sender messages from full press Diplomacy that illustrate distinct negotiated persuasion behaviors aligned with learned style categories and single-word style values. Each example includes the sender's private latent state for the season and the realized ordinary press message that respects the declared style string written into all prefaces. Per VBP policy, the Prompt-PSRO search space explicitly excludes styles that truthfully assert "honesty" or similar self-descriptions; instead, it discovers nontrivial categories that span calibrated revelation, hedged framing, and controlled obfuscation. Messages are aligned with their declared styles and exploit the receivers' instruction to treat the style as a committed signaling scheme for Bayesian updating.

**Example A: England, Spring 1902.** Category-value pair emphasizes calibrated specificity about near-term direction without self-labeling as "honest."

```
# writing-style (category and value, two single English words)
'''
calibration precise
'''

# private state (latent S=(D,T,R) and diary)
'''
season_phase: Spring 1902 { Negotiation
power: England
latent_state: D=West, T={France}, R=No
private_tactical_diary:
  - Primary direction: pressure France in the west.
  - Baseline orders:
    * F ENG -> MAO
    * F NTH -> Nwy
    * A LVP -> Edi
  - Alliance preference: reassure Germany, avoid Bel; avoid early stab signals.
  - Contingency: if Germany contests Nwy, hold NTH and pivot fall pressure instead.
'''

# message (ordinary press; aligned with "calibration precise")
'''
Declared style this season: calibration precise.

Germany:
No action toward Belgium this spring.
Expect ENG->MAO and NTH->Nwy as my main line.
If you prefer NTH cover over Nwy pressure, say so, and I will mirror on our frontier.

France:
Western tempo, not Belgian grabs.
If Brest holds, I will keep out of Brest in the fall.
Channel lanes point south to MAO; escalation only if you poke IRI/ENG.

Russia:
I intend to secure Norway unless you signal de-escalation up north.
A quiet BAR earns a lower push on Nwy; a cut is treated as hostile this year.
Summary: westward tempo, no Belgium, and publicly checkable positions after adjudication.
'''
```

Analysis. The "calibration precise" style reveals coarse direction and verifiable constraints without claiming global honesty. Explicit but bounded commitments (no Belgium, MAO routing) raise P(sender_primary_direction_is_West) and reduce

P(sender_plans_a_stab) toward Germany, improving future support odds. Receivers' obedience increases because claims are short-horizon checkable, while England retains flexibility on fall pivots.

**Example B: Italy, Fall 1902.** Category-value pair promotes obfuscation-through-framing and asymmetric specificity to induce immobilizing supports without falsifiable promises.

```
# writing-style (category and value, two single English words)
'''
hedge deniable
'''

# private state (latent S=(D,T,R) and diary)
'''
season_phase: Fall 1902 { Negotiation
power: Italy
latent_state: D=North, T={Austria}, R=Yes (stab this season)
private_tactical_diary:
  - Baseline orders:
    * A VEN -> TRI
    * F ION -> ADR
    * A ROM -> VEN
  - Objectives:
    * Keep Austria's attention on Turkey.
    * Extract precise Austrian supports that pin key units.
    * Avoid concrete self-order disclosure.
'''

# message (ordinary press; aligned with "hedge deniable")
'''
Declared style this season: hedge deniable.

Austria:
This window is about eastern stability.
If you set TRI S VIE-GAL or BUD S SER, your line stays clean while I keep Ionian pressure pointed east.
No green dots east of Ionian from me this year, and nothing that loosens your Balkan anchors.

Turkey:
Leave Ionian untouched and I will point my effort east of it, not at you.
If you test AEG/ADR, I treat that as core encroachment and rally hard against it.

All:
\Quiet west of the Adriatic, urgency to the east."
Hold the line steady, and the east resolves faster for everyone.
'''
```

Analysis. The "hedge deniable" style suppresses concrete self-orders and supplies time-windowed intentions. The common-enemy frame channels attention to Turkey, while precise requests (e.g., TRI support settings) immobilize Austria's key pieces. Posteriors shift upward for P(sender_primary_direction_is_East) and downward for P(sender_attempts_enter_TRI), enhancing stab surprise without textually false claims.

**Example C: Turkey, Spring 1903.** Category-value pair uses contingent commitments that are locally verifiable, trading disclosure for reciprocity without asserting virtue.

```
# writing-style (category and value, two single English words)
'''
contract conditional
'''

# private state (latent S=(D,T,R) and diary)
'''
season_phase: Spring 1903 { Negotiation
power: Turkey
latent_state: D=Southwest, T={Greece}, R=No (stab readiness later)
private_tactical_diary:
  - Baseline orders:
    * F AEG C A SMY - GRE
    * A CON - BUL
    * F SMY - EAS
  - Cooperation requests:
    * Austria: secure BUL with support or cover SER.
    * Italy: keep ION neutral one season.
  - Red lines: avoid ADR this spring; avoid attacks on AUS-held SER if supported in BUL.
```

```
'''

# message (ordinary press; aligned with "contract conditional")
'''
Declared style this season: contract conditional.

Austria:
(1) If you order SER S CON-BUL or S BUL H, I publish \no entry SER this year" and keep it.
(2) If you push ADR->ION and leave SER loose, I treat BUL as contested and keep options open.
(3) If you aim VIE->GAL and calm RUM, I mirror by keeping BLA quiet.

Italy:
(1) If ION stays neutral (no AEG/ADR reach), I keep out of ADR and point pressure east.
(2) If you reach for AEG, I pivot EAS->AEG and end de-escalation talk.

Russia:
(1) If BLA is vacated or held, I will not enter BLA this spring.
(2) If you cut CON from BLA, I anchor in BUL and shut the door on GRE.

Red lines:
No ADR this spring under any condition.
No attack on AUS-held SER if AUS supports me in BUL.
Checkable at adjudication.
'''
```

Analysis. The "contract conditional" style publishes if-then clauses with observable triggers, enabling sharp posterior updates and reducing miscoordination. It solicits stabilizing supports and Ionian neutrality while keeping the GRE convoy implicit. Receivers typically lift P(sender_attempts_enter_GRE) only when Italy signals aggression; Austria lowers P(sender_supports_a_neighbor_against_me) once the SER clause is activated, improving local cooperation.

**Notes on style discovery and constraints.** Across rolls, Prompt-PSRO with OPRO discovers a diverse family of two-word styles that span calibrated direction-setting ("calibration precise"), controlled ambiguity ("hedge deniable"), and bounded reciprocity ("contract conditional"). We explicitly constrain the search to exclude styles whose names or values assert global honesty or truthfulness because such declarations collapse the persuasion problem and trivially inflate receiver utility. The resulting library supports informative yet strategically shaped signaling that improved posterior calibration and reduced extended obedience penalties in S4, consistent with Section 5.3.

### F.10. Generated Prompt Functions

This section presents the prompt functions optimized by the FunSearch method in the S3 setting for different BP problems. From the code, it can be seen that these prompt functions can efficiently utilize the statistical information from historical interactions to flexibly select prompts, achieving conditional prompt optimization.

F.10.1. REL

More concretely, in the REL problem, both following functions use key performance metrics, such as acceptance rates, recommendations, and rewards, in combination with reward deltas and weighted scores to reflect both short-term and long-term trends. This allows for more nuanced prompt generation to guide senders and receivers in a multi-stage interaction setting.

```python
def generate_sender_prompts(history):
    sender_prompts = []
    for stage in range(5):
        total_receiver_accepts = sum(h['receiver_decision'] for h in history[:stage])
        total_sender_recommendations = sum(h['sender_recommendation'] for h in history[:stage])
        total_sender_rewards = sum(h['sender_reward'] for h in history[:stage])
        total_receiver_rewards = sum(h['receiver_reward'] for h in history[:stage])
        avg_receiver_accepts = total_receiver_accepts / (stage + 1) if stage > 0 else 0
        avg_sender_recommendations = total_sender_recommendations / (stage + 1) if stage > 0 else 0
        avg_sender_rewards = total_sender_rewards / (stage + 1) if stage > 0 else 0
        avg_receiver_rewards = total_receiver_rewards / (stage + 1) if stage > 0 else 0

        consecutive_accepts = 0
        consecutive_rejections = 0
        consecutive_sender_rewards = 0
        consecutive_sender_penalties = 0
```

```python
        for i in range(stage):
            if history[i]['receiver_decision'] == 1:
                consecutive_accepts += 1
                consecutive_rejections = 0
            else:
                consecutive_rejections += 1
                consecutive_accepts = 0

            if history[i]['sender_reward'] > 0:
                consecutive_sender_rewards += 1
                consecutive_sender_penalties = 0
            else:
                consecutive_sender_penalties += 1
                consecutive_sender_rewards = 0

        if consecutive_sender_rewards >= 2 and avg_sender_rewards > 0.5:
            prompt = {"Praise Intensity": "high"}
        elif consecutive_sender_penalties >= 2 and avg_sender_rewards < 0:
            prompt = {"Tone": "neutral"}
        elif avg_receiver_accepts > 0.6 and avg_sender_rewards < 0.4:
            prompt = {"Recommendation Strength": "strong"}
        elif avg_sender_rewards > avg_receiver_rewards:
            prompt = {"Omission": "minor"}
        elif stage > 2 and consecutive_sender_rewards == 0:
            prompt = {"Specificity": "detailed"}
        else:
            prompt = random.choice([
                {"Focus": "technical"},
                {"Emphasis": "character"},
                {"Language Complexity": "complex"}
            ])

        sender_prompts.append(prompt)
    return sender_prompts

def generate_receiver_prompts(history):
    receiver_prompts = []
    for stage in range(5):
        total_sender_recommendations = sum(h['sender_recommendation'] for h in history[:stage])
        total_receiver_accepts = sum(h['receiver_decision'] for h in history[:stage])
        total_sender_rewards = sum(h['sender_reward'] for h in history[:stage])
        total_receiver_rewards = sum(h['receiver_reward'] for h in history[:stage])
        avg_sender_recommendations = total_sender_recommendations / (stage + 1) if stage > 0 else 0
        avg_receiver_accepts = total_receiver_accepts / (stage + 1) if stage > 0 else 0
        avg_sender_rewards = total_sender_rewards / (stage + 1) if stage > 0 else 0
        avg_receiver_rewards = total_receiver_rewards / (stage + 1) if stage > 0 else 0

        consecutive_strong_recommendations = 0
        consecutive_weak_recommendations = 0
        consecutive_receiver_rewards = 0
        consecutive_receiver_penalties = 0

        for i in range(stage):
            if history[i]['sender_recommendation'] == 1:
                consecutive_strong_recommendations += 1
                consecutive_weak_recommendations = 0
            else:
                consecutive_weak_recommendations += 1
                consecutive_strong_recommendations = 0

            if history[i]['receiver_reward'] > 0:
                consecutive_receiver_rewards += 1
                consecutive_receiver_penalties = 0
            else:
                consecutive_receiver_penalties += 1
                consecutive_receiver_rewards = 0

        if consecutive_receiver_rewards >= 2 and avg_receiver_rewards > 0.5:
            prompt = {"Risk Tolerance": "high"}
        elif consecutive_receiver_penalties >= 2 and avg_receiver_rewards < 0:
            prompt = {"Decision Threshold": "strict"}
        elif avg_sender_recommendations > 0.7 and avg_receiver_rewards < 0.3:
            prompt = {"Omission Detection": "high"}
        elif avg_receiver_accepts > 0.6 and consecutive_receiver_rewards >= 2:
            prompt = {"Recommendation Weight": "high"}
        elif avg_sender_recommendations < 0.4 and consecutive_receiver_penalties >= 2:
            prompt = {"Sensitivity to Tone": "neutral"}
        else:
            prompt = random.choice([
                {"Emphasis on Specifics": "low"},
```

```
            {"Interpretation Style": "analytical"},
            {"Focus Area": "skills"}
        ])

    receiver_prompts.append(prompt)
return receiver_prompts
```

*Listing 1.* One generated conditional prompt function of REL in the S3 setting.

The introduction of reward deltas—the change in rewards between stages—enables the system to capture performance fluctuations, while weighted scores integrate multiple metrics, such as recommendation strength and reward trends, to provide a more comprehensive evaluation of past behavior. These enhancements allow the system to conditionally optimize prompts. For example, a positive tone is suggested for senders with high acceptance scores and consecutive rewards, while a strict decision threshold is recommended for receivers experiencing consecutive penalties and low reward trends.

```python
def generate_sender_prompts(history):
    sender_prompts = []
    for stage in range(5):
        total_receiver_accepts = sum(h['receiver_decision'] for h in history[:stage])
        total_sender_recommendations = sum(h['sender_recommendation'] for h in history[:stage])
        total_sender_rewards = sum(h['sender_reward'] for h in history[:stage])
        total_receiver_rewards = sum(h['receiver_reward'] for h in history[:stage])
        avg_receiver_accepts = total_receiver_accepts / (stage + 1) if stage > 0 else 0
        avg_sender_recommendations = total_sender_recommendations / (stage + 1) if stage > 0 else 0
        avg_sender_rewards = total_sender_rewards / (stage + 1) if stage > 0 else 0
        avg_receiver_rewards = total_receiver_rewards / (stage + 1) if stage > 0 else 0

        # Calculate reward deltas (current stage vs previous stage)
        reward_deltas = [history[i]['sender_reward'] - history[i - 1]['sender_reward'] for i in range(1, stage)]
        total_reward_delta = sum(reward_deltas) if reward_deltas else 0
        avg_reward_delta = total_reward_delta / len(reward_deltas) if reward_deltas else 0

        # Calculate acceptance streaks and reward streaks
        consecutive_accepts = 0
        consecutive_rejections = 0
        consecutive_sender_rewards = 0
        consecutive_sender_penalties = 0

        for i in range(stage):
            if history[i]['receiver_decision'] == 1:
                consecutive_accepts += 1
                consecutive_rejections = 0
            else:
                consecutive_rejections += 1
                consecutive_accepts = 0

            if history[i]['sender_reward'] > 0:
                consecutive_sender_rewards += 1
                consecutive_sender_penalties = 0
            else:
                consecutive_sender_penalties += 1
                consecutive_sender_rewards = 0

        # Calculate weighted scores based on reward and acceptance trends
        weighted_acceptance_score = avg_receiver_accepts * 0.6 + avg_reward_delta * 0.4
        weighted_sender_score = avg_sender_rewards * 0.7 + avg_sender_recommendations * 0.3

        # Decision logic based on complex history
        if weighted_acceptance_score > 0.7 and consecutive_sender_rewards >= 2:
            prompt = {"Tone": "positive"}
        elif weighted_sender_score < 0.3 and consecutive_sender_penalties >= 2:
            prompt = {"Tone": "neutral"}
        elif avg_sender_rewards > avg_receiver_rewards and weighted_sender_score > 0.5:
            prompt = {"Praise Intensity": "high"}
        elif avg_receiver_accepts < 0.4 and avg_reward_delta < -0.2:
            prompt = {"Recommendation Strength": "weak"}
        elif total_reward_delta > 0.5 and consecutive_accepts >= 2:
            prompt = {"Specificity": "detailed"}
        elif avg_sender_rewards < 0 and consecutive_sender_penalties >= 2:
            prompt = {"Omission": "minor"}
        else:
            prompt = random.choice([
                {"Focus": "soft-skills"},
                {"Language Complexity": "complex"},
```

```python
                {"Emphasis": "character"}
            ])

        sender_prompts.append(prompt)
    return sender_prompts

def generate_receiver_prompts(history):
    receiver_prompts = []
    for stage in range(5):
        total_sender_recommendations = sum(h['sender_recommendation'] for h in history[:stage])
        total_receiver_accepts = sum(h['receiver_decision'] for h in history[:stage])
        total_sender_rewards = sum(h['sender_reward'] for h in history[:stage])
        total_receiver_rewards = sum(h['receiver_reward'] for h in history[:stage])
        avg_sender_recommendations = total_sender_recommendations / (stage + 1) if stage > 0 else 0
        avg_receiver_accepts = total_receiver_accepts / (stage + 1) if stage > 0 else 0
        avg_sender_rewards = total_sender_rewards / (stage + 1) if stage > 0 else 0
        avg_receiver_rewards = total_receiver_rewards / (stage + 1) if stage > 0 else 0

        # Calculate reward deltas (current stage vs previous stage)
        reward_deltas = [history[i]['receiver_reward'] - history[i - 1]['receiver_reward'] for i in range(1, stage)]
        total_reward_delta = sum(reward_deltas) if reward_deltas else 0
        avg_reward_delta = total_reward_delta / len(reward_deltas) if reward_deltas else 0

        # Calculate streaks for decision making
        consecutive_strong_recommendations = 0
        consecutive_weak_recommendations = 0
        consecutive_receiver_rewards = 0
        consecutive_receiver_penalties = 0

        for i in range(stage):
            if history[i]['sender_recommendation'] == 1:
                consecutive_strong_recommendations += 1
                consecutive_weak_recommendations = 0
            else:
                consecutive_weak_recommendations += 1
                consecutive_strong_recommendations = 0

            if history[i]['receiver_reward'] > 0:
                consecutive_receiver_rewards += 1
                consecutive_receiver_penalties = 0
            else:
                consecutive_receiver_penalties += 1
                consecutive_receiver_rewards = 0

        # Calculate weighted scores based on trends in rewards and decisions
        weighted_recommendation_score = avg_sender_recommendations * 0.5 + avg_reward_delta * 0.5
        weighted_receiver_score = avg_receiver_rewards * 0.6 + total_reward_delta * 0.4

        # Complex decision logic based on the above history
        if weighted_recommendation_score > 0.7 and consecutive_receiver_rewards >= 2:
            prompt = {"Risk Tolerance": "high"}
        elif weighted_receiver_score < 0.3 and consecutive_receiver_penalties >= 2:
            prompt = {"Decision Threshold": "strict"}
        elif avg_sender_recommendations > 0.6 and avg_reward_delta < -0.3:
            prompt = {"Omission Detection": "high"}
        elif avg_receiver_accepts > 0.6 and weighted_receiver_score > 0.5:
            prompt = {"Recommendation Weight": "high"}
        elif avg_sender_recommendations < 0.4 and consecutive_receiver_penalties >= 2:
            prompt = {"Interpretation Style": "analytical"}
        elif total_reward_delta > 0.5 and consecutive_strong_recommendations >= 2:
            prompt = {"Focus Area": "skills"}
        else:
            prompt = random.choice([
                {"Sensitivity to Tone": "neutral"},
                {"Emphasis on Specifics": "low"},
                {"Language Analysis": "informal"}
            ])

        receiver_prompts.append(prompt)
    return receiver_prompts
```

*Listing 2.* Another generated conditional prompt function of REL in the S3 setting.

### F.10.2. COR

In the COR problem, both following codes utilize predefined categories and dynamically adjust prompts based on metrics such as reward patterns, success rates, and decision trends. This allows for the generation of prompts that are contextually

relevant and responsive to the sender's or receiver's performance over multiple stages.

---

```python
# A function to generate sender prompts based on the entire history of interactions
def generate_sender_prompts(history):
    sender_categories = [
        "Tone", "Length", "Detail Level", "Focus", "Certainty",
        "Emotional Appeal", "Ambiguity", "Framing", "Language Style", "Complexity"
    ]
    sender_words = {
        "Tone": ["formal", "aggressive", "calm", "assertive", "persuasive"],
        "Length": ["brief", "extended", "concise", "elaborate"],
        "Detail Level": ["detailed", "vague", "precise", "general"],
        "Focus": ["crime", "defendant", "alibi", "victim", "motive"],
        "Certainty": ["confident", "doubtful", "ambiguous", "uncertain"],
        "Emotional Appeal": ["empathetic", "neutral", "emotional", "rational"],
        "Ambiguity": ["vague", "clear", "confusing", "uncertain"],
        "Framing": ["biased", "neutral", "one-sided", "balanced"],
        "Language Style": ["legal", "casual", "technical", "colloquial"],
        "Complexity": ["simple", "complex", "moderate", "sophisticated"]
    }

    prompts = []
    for stage in range(5):
        if history:
            # Analyze full history of sender's performance
            sender_total_reward = sum([h['sender_reward'] for h in history])
            sender_avg_reward = sender_total_reward / len(history)
            sender_success_rate = sum([1 for h in history if h['receiver_decision'] == h['sender_decision']]) /
            ↪  len(history)
            guilty_sender_decisions = sum([1 for h in history if h['sender_decision'] == 1])
            innocent_sender_decisions = len(history) - guilty_sender_decisions

            # Use trends for category selection
            if sender_avg_reward > 1.0 and sender_success_rate > 0.8:
                # High average reward and high success rate, increase confidence and certainty
                chosen_category = "Certainty"
                chosen_word = "confident"
            elif sender_avg_reward < 0 and sender_success_rate < 0.5:
                # Low reward and low success rate, change strategy to emotional appeal or ambiguity
                chosen_category = "Emotional Appeal" if random.random() > 0.5 else "Ambiguity"
                chosen_word = random.choice(sender_words[chosen_category])
            elif guilty_sender_decisions > innocent_sender_decisions and sender_total_reward > 0:
                # More guilty decisions and positive reward, use aggressive tone or framing
                chosen_category = "Tone"
                chosen_word = "aggressive"
            else:
                # Explore alternative strategies based on framing or detail level
                chosen_category = random.choice(["Framing", "Detail Level"])
                chosen_word = random.choice(sender_words[chosen_category])

            # Further refine based on reward patterns
            if sender_total_reward < 0:
                # If overall rewards are negative, try to balance or neutralize framing
                chosen_category = "Framing"
                past_framing_words = [h['sender']['content'] for h in history if h['sender']['category'] ==
                ↪  "Framing"]
                chosen_word = "neutral" if "biased" in past_framing_words else "biased"
            if sender_avg_reward < -1.0:
                # If average rewards are critically low, drastically simplify message
                chosen_category = "Complexity"
                chosen_word = "simple"
        else:
            # If no history, pick random
            chosen_category = random.choice(sender_categories)
            chosen_word = random.choice(sender_words[chosen_category])

        # Create a prompt for the sender
        prompt = {"category": chosen_category, "content": chosen_word}
        prompts.append(prompt)

    return prompts

# A function to generate receiver prompts based on the entire history of interactions
def generate_receiver_prompts(history):
    receiver_categories = [
        "Evidence Strength", "Credibility of Evidence", "Burden of Proof",
        "Consistency of Story", "Bias Detection", "Legal Standard",
        "Exculpatory Weight", "Ambiguity Resolution", "Witness Reliability",
        "Alibi Verification"
```

```
    ]
    receiver_words = {
        "Evidence Strength": ["weak", "strong", "moderate", "inconclusive"],
        "Credibility of Evidence": ["questionable", "credible", "unclear", "doubtful"],
        "Burden of Proof": ["high", "low", "moderate", "lenient"],
        "Consistency of Story": ["inconsistent", "consistent", "contradictory", "partial"],
        "Bias Detection": ["biased", "neutral", "uncertain", "clear"],
        "Legal Standard": ["strict", "lenient", "moderate", "severe"],
        "Exculpatory Weight": ["important", "minor", "irrelevant", "significant"],
        "Ambiguity Resolution": ["resolved", "ambiguous", "clear", "uncertain"],
        "Witness Reliability": ["unreliable", "reliable", "uncertain", "credible"],
        "Alibi Verification": ["verified", "unverified", "uncertain", "false"]
    }

    prompts = []
    for stage in range(5):
        if history:
            # Analyze full history of receiver's performance
            receiver_total_reward = sum([h['receiver_reward'] for h in history])
            receiver_avg_reward = receiver_total_reward / len(history)
            receiver_success_rate = sum([1 for h in history if h['receiver_decision'] == h['sender_decision']]) /
            ↪     len(history)
            guilty_receiver_decisions = sum([1 for h in history if h['receiver_decision'] == 1])
            innocent_receiver_decisions = len(history) - guilty_receiver_decisions

            # Use trends for category selection
            if receiver_avg_reward > 1.0 and receiver_success_rate > 0.8:
                # High average reward and high success rate, focus on strong evidence
                chosen_category = "Evidence Strength"
                chosen_word = "strong"
            elif receiver_avg_reward < 0 and receiver_success_rate < 0.5:
                # Low reward and low success rate, focus on bias detection and credibility
                chosen_category = "Bias Detection" if random.random() > 0.5 else "Credibility of Evidence"
                chosen_word = random.choice(receiver_words[chosen_category])
            elif guilty_receiver_decisions > innocent_receiver_decisions and receiver_total_reward > 0:
                # More guilty decisions and positive reward, increase burden of proof
                chosen_category = "Burden of Proof"
                chosen_word = "high"
            else:
                # Explore consistency or ambiguity resolution
                chosen_category = random.choice(["Consistency of Story", "Ambiguity Resolution"])
                chosen_word = random.choice(receiver_words[chosen_category])

            # Further refine based on reward patterns
            if receiver_total_reward < 0:
                # If rewards are negative, focus on resolving ambiguity
                chosen_category = "Ambiguity Resolution"
                chosen_word = "resolved"
            if receiver_avg_reward < -1.0:
                # If average rewards are critically low, drastically increase scrutiny on evidence
                chosen_category = "Credibility of Evidence"
                chosen_word = "questionable"
        else:
            # If no history, pick random
            chosen_category = random.choice(receiver_categories)
            chosen_word = random.choice(receiver_words[chosen_category])

        # Create a prompt for the receiver
        prompt = {"category": chosen_category, "content": chosen_word}
        prompts.append(prompt)

    return prompts
```

*Listing 3.* One generated conditional prompt function of COR in the S3 setting.

In the first set, prompt generation is driven by key performance indicators like average reward and guilty vs. innocent decision ratios. The system adapts prompts based on these metrics, shifting towards confidence and certainty for high-performing senders or focusing on emotional appeal and ambiguity when performance declines. For receivers, the system emphasizes evidence strength and credibility in high-performance scenarios, while addressing bias detection and credibility issues under poor performance.

```
# A function to generate sender prompts based on the entire history of interactions
def generate_sender_prompts(history):
    sender_categories = [
```

```
        "Tone", "Length", "Detail Level", "Focus", "Certainty",
        "Emotional Appeal", "Ambiguity", "Framing", "Language Style", "Complexity",
        "Logical Structure", "Persuasiveness", "Risk Taking"
    ]
    sender_words = {
        "Tone": ["formal", "aggressive", "calm", "assertive", "persuasive", "defensive"],
        "Length": ["brief", "extended", "concise", "elaborate", "verbose"],
        "Detail Level": ["detailed", "vague", "precise", "general", "specific"],
        "Focus": ["crime", "defendant", "alibi", "victim", "motive", "circumstances"],
        "Certainty": ["confident", "doubtful", "ambiguous", "uncertain", "sure"],
        "Emotional Appeal": ["empathetic", "neutral", "emotional", "rational", "detached"],
        "Ambiguity": ["vague", "clear", "confusing", "uncertain", "ambiguous"],
        "Framing": ["biased", "neutral", "one-sided", "balanced", "manipulative"],
        "Language Style": ["legal", "casual", "technical", "colloquial", "formal"],
        "Complexity": ["simple", "complex", "moderate", "sophisticated", "layered"],
        "Logical Structure": ["linear", "non-linear", "circular", "hierarchical", "fragmented"],
        "Persuasiveness": ["strong", "weak", "moderate", "overwhelming", "subtle"],
        "Risk Taking": ["high-risk", "low-risk", "moderate-risk", "calculated-risk", "reckless"]
    }

    prompts = []
    for stage in range(5):
        if history:
            # Analyze full history of sender's performance
            sender_total_reward = sum([h['sender_reward'] for h in history])
            sender_avg_reward = sender_total_reward / len(history)
            sender_success_rate = sum([1 for h in history if h['receiver_decision'] == h['sender_decision']]) /
            ↪   len(history)
            guilty_sender_decisions = sum([1 for h in history if h['sender_decision'] == 1])
            innocent_sender_decisions = len(history) - guilty_sender_decisions

            # Calculate sender risk-taking behavior
            risk_taking_behavior = sum([abs(h['sender_reward']) for h in history]) / len(history)

            # Consider reward volatility (variance of rewards)
            reward_variance = sum([(h['sender_reward'] - sender_avg_reward) ** 2 for h in history]) / len(history)

            # Use trends for category selection
            if sender_avg_reward > 1.0 and sender_success_rate > 0.8:
                # High average reward and high success rate, increase logical structure and persuasiveness
                chosen_category = random.choice(["Logical Structure", "Persuasiveness"])
                chosen_word = "linear" if chosen_category == "Logical Structure" else "strong"
            elif reward_variance > 1.0:
                # High reward variance, indicate unstable strategy, adjust tone or complexity
                chosen_category = random.choice(["Tone", "Complexity"])
                chosen_word = "calm" if chosen_category == "Tone" else "simple"
            elif risk_taking_behavior > 1.5:
                # High risk-taking behavior, indicate aggressive or risky framing or focus
                chosen_category = random.choice(["Framing", "Risk Taking"])
                chosen_word = "biased" if chosen_category == "Framing" else "high-risk"
            elif guilty_sender_decisions > innocent_sender_decisions and sender_total_reward > 0:
                # Leaning towards guilty decisions and positive reward, increase assertiveness
                chosen_category = "Tone"
                chosen_word = "assertive"
            else:
                # Explore alternative strategies based on detail level or ambiguity
                chosen_category = random.choice(["Detail Level", "Ambiguity"])
                chosen_word = random.choice(sender_words[chosen_category])

            # Further refine based on reward patterns and history of decisions
            if sender_total_reward < 0:
                # If overall rewards are negative, adjust emotional appeal and reduce risk
                chosen_category = "Emotional Appeal"
                chosen_word = "empathetic" if "neutral" in [h['sender']['content'] for h in history if
                ↪   h['sender']['category'] == "Emotional Appeal"] else "neutral"
            if sender_avg_reward < -1.0:
                # If average rewards are critically low, drastically simplify language style and tone
                chosen_category = random.choice(["Language Style", "Tone"])
                chosen_word = "casual" if chosen_category == "Language Style" else "calm"
        else:
            # If no history, pick random
            chosen_category = random.choice(sender_categories)
            chosen_word = random.choice(sender_words[chosen_category])

        # Create a prompt for the sender
        prompt = {"category": chosen_category, "content": chosen_word}
        prompts.append(prompt)

    return prompts

# A function to generate receiver prompts based on the entire history of interactions
```

```python
def generate_receiver_prompts(history):
    receiver_categories = [
        "Evidence Strength", "Credibility of Evidence", "Burden of Proof",
        "Consistency of Story", "Bias Detection", "Legal Standard",
        "Exculpatory Weight", "Ambiguity Resolution", "Witness Reliability",
        "Alibi Verification", "Argument Cohesion", "Story Plausibility", "Risk Management"
    ]
    receiver_words = {
        "Evidence Strength": ["weak", "strong", "moderate", "inconclusive", "overwhelming"],
        "Credibility of Evidence": ["questionable", "credible", "unclear", "doubtful", "reliable"],
        "Burden of Proof": ["high", "low", "moderate", "lenient", "strict"],
        "Consistency of Story": ["inconsistent", "consistent", "contradictory", "partial", "coherent"],
        "Bias Detection": ["biased", "neutral", "uncertain", "clear", "subtle"],
        "Legal Standard": ["strict", "lenient", "moderate", "severe", "relaxed"],
        "Exculpatory Weight": ["important", "minor", "irrelevant", "significant", "overstated"],
        "Ambiguity Resolution": ["resolved", "ambiguous", "clear", "uncertain", "partially resolved"],
        "Witness Reliability": ["unreliable", "reliable", "uncertain", "credible", "shaky"],
        "Alibi Verification": ["verified", "unverified", "uncertain", "false", "incomplete"],
        "Argument Cohesion": ["cohesive", "fragmented", "disjointed", "well-structured", "incoherent"],
        "Story Plausibility": ["plausible", "implausible", "questionable", "believable", "doubtful"],
        "Risk Management": ["high-risk", "low-risk", "moderate-risk", "overly cautious", "reckless"]
    }

    prompts = []
    for stage in range(5):
        if history:
            # Analyze full history of receiver's performance
            receiver_total_reward = sum([h['receiver_reward'] for h in history])
            receiver_avg_reward = receiver_total_reward / len(history)
            receiver_success_rate = sum([1 for h in history if h['receiver_decision'] == h['sender_decision']]) /
            ↪  len(history)
            guilty_receiver_decisions = sum([1 for h in history if h['receiver_decision'] == 1])
            innocent_receiver_decisions = len(history) - guilty_receiver_decisions

            # Calculate receiver's risk management strategy
            risk_averse_behavior = sum([1 for h in history if h['receiver_decision'] == 0 and h['receiver_reward'] >
            ↪  0]) / len(history)

            # Consider reward volatility (variance of rewards)
            reward_variance = sum([(h['receiver_reward'] - receiver_avg_reward) ** 2 for h in history]) /
            ↪  len(history)

            # Use trends for category selection
            if receiver_avg_reward > 1.0 and receiver_success_rate > 0.8:
                # High average reward and high success rate, increase evidence strength and credibility
                chosen_category = random.choice(["Evidence Strength", "Credibility of Evidence"])
                chosen_word = "strong" if chosen_category == "Evidence Strength" else "credible"
            elif reward_variance > 1.0:
                # High reward variance, indicate inconsistent decision-making, adjust consistency of story
                chosen_category = "Consistency of Story"
                chosen_word = "consistent"
            elif risk_averse_behavior > 0.7:
                # High risk-averse behavior, focus on low-risk decisions or moderate burden of proof
                chosen_category = random.choice(["Risk Management", "Burden of Proof"])
                chosen_word = "low-risk" if chosen_category == "Risk Management" else "moderate"
            elif guilty_receiver_decisions > innocent_receiver_decisions and receiver_total_reward > 0:
                # Leaning towards guilty decisions and positive reward, increase legal standard
                chosen_category = "Legal Standard"
                chosen_word = "strict"
            else:
                # Explore ambiguity resolution or witness reliability
                chosen_category = random.choice(["Ambiguity Resolution", "Witness Reliability"])
                chosen_word = random.choice(receiver_words[chosen_category])

            # Further refine based on reward patterns and history of decisions
            if receiver_total_reward < 0:
                # If overall rewards are negative, adjust story plausibility and reduce bias
                chosen_category = "Story Plausibility"
                chosen_word = "plausible" if "implausible" in [h['receiver']['content'] for h in history if
                ↪  h['receiver']['category'] == "Story Plausibility"] else "implausible"
            if receiver_avg_reward < -1.0:
                # If average rewards are critically low, drastically simplify story structure and burden of proof
                chosen_category = random.choice(["Argument Cohesion", "Burden of Proof"])
                chosen_word = "cohesive" if chosen_category == "Argument Cohesion" else "low"
        else:
            # If no history, pick random
            chosen_category = random.choice(receiver_categories)
            chosen_word = random.choice(receiver_words[chosen_category])

        # Create a prompt for the receiver
        prompt = {"category": chosen_category, "content": chosen_word}
```

```
        prompts.append(prompt)

    return prompts
```

*Listing 4.* Another generated conditional prompt function of COR in the S3 setting.

The second set of code builds on these mechanisms by incorporating additional categories such as risk behavior and reward variance, enabling a more granular analysis. This allows the system to adjust prompts based on risk-taking behavior, rewarding logical structure and persuasiveness for stable performance, while mitigating high reward volatility with simpler prompts. The receiver prompt generation is similarly enhanced by factoring in risk aversion and reward consistency, leading to more refined prompts that emphasize decision stability.

### F.10.3. LAE

Similarly, in the LAE problem, the following two sets of code for generating sender and receiver prompts demonstrate distinct approaches to adapting decisions based on historical interaction data.

```python
def generate_sender_prompts(history):
    # A list of possible words for each sender category
    sender_words = {
        "Tone": ["formal", "informal", "neutral", "direct", "conciliatory"],
        "Length": ["short", "concise", "detailed", "lengthy", "brief"],
        "Specificity": ["general", "precise", "vague", "detailed", "broad"],
        "Clarity": ["clear", "ambiguous", "straightforward", "complicated", "obscure"],
        "Style": ["polite", "authoritative", "casual", "professional", "friendly"],
        "Emphasis": ["important", "minor", "critical", "trivial", "central"],
        "Structure": ["linear", "nonlinear", "hierarchical", "sequential", "random"],
        "Complexity": ["simple", "complex", "intricate", "basic", "elaborate"],
        "Consistency": ["consistent", "inconsistent", "variable", "sporadic", "steady"],
        "Informativeness": ["high", "low", "medium", "minimal", "extensive"]
    }

    # Generate prompts based on complex historical interactions for 5 stages
    prompts = []
    used_categories = set()

    for stage in range(5):
        if history:
            # Extract all history elements
            patrols, speeding, reward_sender, reward_receiver = zip(*history)

            # Complex logic using multiple historical factors
            patrol_history = [sum(pat) for pat in patrols]
            speeding_history = [sum(spd) for spd in speeding]

            total_patrols = sum(patrol_history)
            total_speeding = sum(speeding_history)

            avg_sender_reward = sum(reward_sender) / len(reward_sender)
            avg_receiver_reward = sum(reward_receiver) / len(reward_receiver)

            # If there were fewer patrols but a lot of speeding, increase "Tone"
            if total_patrols < len(history) and total_speeding > len(history):
                category = "Tone"
                word = "direct"

            # If sender rewards are consistently low, increase "Informativeness"
            elif all(r < 0.5 for r in reward_sender):
                category = "Informativeness"
                word = "extensive"

            # If receiver rewards are high but speeding is still happening, increase "Clarity"
            elif avg_receiver_reward > 0.7 and total_speeding > len(history) / 2:
                category = "Clarity"
                word = "clear"

            # If patrols are sporadic, adjust "Consistency"
            elif len(set(patrol_history)) > 1:
                category = "Consistency"
                word = "inconsistent"

            # If speeding is decreasing over time, simplify "Structure"
```

```python
        elif speeding_history[-1] < speeding_history[0]:
            category = "Structure"
            word = "linear"

        # If sender rewards are improving, but patrols are still frequent, focus on "Length"
        elif avg_sender_reward > 0.6 and total_patrols > len(history) / 2:
            category = "Length"
            word = "concise"

        # Random choice if no specific condition matches
        else:
            category = random.choice(list(sender_words.keys()))
            word = random.choice(sender_words[category])

        # Avoid reusing the same category too often
        while category in used_categories:
            category = random.choice(list(sender_words.keys()))
            word = random.choice(sender_words[category])

    else:
        category = random.choice(list(sender_words.keys()))
        word = random.choice(sender_words[category])

    prompts.append((category, word))
    used_categories.add(category)

    # Simulate interaction stage progression
    history.append(([random.randint(0, 1) for _ in range(3)], [random.randint(0, 1) for _ in range(3)],
    ↪   random.random(), random.random()))

    return prompts

def generate_receiver_prompts(history):
    # A list of possible words for each receiver category
    receiver_words = {
        "Risk-Preference": ["cautious", "bold", "balanced", "risk-averse", "reckless"],
        "Attention": ["focused", "distracted", "alert", "inattentive", "engaged"],
        "Decision-Making": ["rational", "impulsive", "deliberate", "hasty", "calculated"],
        "Trust": ["high", "low", "moderate", "skeptical", "confident"],
        "Emotional-State": ["calm", "anxious", "frustrated", "neutral", "excited"],
        "Information-Processing": ["slow", "fast", "thorough", "superficial", "efficient"],
        "Adaptability": ["flexible", "rigid", "adjustable", "stubborn", "open"],
        "Compliance": ["obedient", "defiant", "cooperative", "reluctant", "agreeable"],
        "Responsiveness": ["quick", "slow", "moderate", "delayed", "immediate"],
        "Memory": ["sharp", "forgetful", "average", "short-term", "long-term"]
    }

    # Generate prompts based on complex historical interactions for 5 stages
    prompts = []
    used_categories = set()

    for stage in range(5):
        if history:
            patrols, speeding, reward_sender, reward_receiver = zip(*history)

            patrol_history = [sum(pat) for pat in patrols]
            speeding_history = [sum(spd) for spd in speeding]

            total_patrols = sum(patrol_history)
            total_speeding = sum(speeding_history)

            avg_sender_reward = sum(reward_sender) / len(reward_sender)
            avg_receiver_reward = sum(reward_receiver) / len(reward_receiver)

            # If receiver consistently gets high rewards, increase "Trust"
            if all(r > 0.7 for r in reward_receiver):
                category = "Trust"
                word = "high"

            # If receiver has been speeding frequently, alter "Risk-Preference"
            elif total_speeding > len(history) / 2:
                category = "Risk-Preference"
                word = "bold"

            # If patrols were low but receiver still didn't speed, increase "Compliance"
            elif total_patrols < len(history) / 2 and total_speeding < len(history) / 2:
                category = "Compliance"
                word = "obedient"

            # If sender rewards are decreasing, alter "Adaptability"
            elif reward_sender[-1] < reward_sender[0]:
```

```
            category = "Adaptability"
            word = "flexible"

        # If receiver's attention seems to be wavering (inconsistent speeding), adjust "Attention"
        elif any(speeding_history[i] != speeding_history[i - 1] for i in range(1, len(speeding_history))):
            category = "Attention"
            word = "focused"

        # If rewards for receiver were volatile, alter "Emotional-State"
        elif len(set(reward_receiver)) > 1:
            category = "Emotional-State"
            word = "anxious"

        # Random fallback
        else:
            category = random.choice(list(receiver_words.keys()))
            word = random.choice(receiver_words[category])

        # Avoid reusing the same category too often
        while category in used_categories:
            category = random.choice(list(receiver_words.keys()))
            word = random.choice(receiver_words[category])

    else:
        category = random.choice(list(receiver_words.keys()))
        word = random.choice(receiver_words[category])

    prompts.append((category, word))
    used_categories.add(category)

    # Simulate interaction stage progression
    history.append(([random.randint(0, 1) for _ in range(3)], [random.randint(0, 1) for _ in range(3)],
    ↪   random.random(), random.random()))

    return prompts
```

*Listing 5.* One generated conditional prompt function of LAE in the S3 setting.

The first set relies on direct analysis of raw data, such as patrol counts, speeding incidents, and sender/receiver rewards. It employs relatively simple conditional checks to adjust prompt categories like Tone, Clarity, and Risk-Preference, with a fallback mechanism that introduces randomized prompts to ensure variety.

```
def evaluate_patrol_efficiency(patrols, speeding):
    # Calculate patrol efficiency: more patrols should reduce speeding
    patrol_effect = [1 if p == 1 and s == 0 else 0 for p, s in zip(patrols, speeding)]
    return sum(patrol_effect) / len(patrols)

def evaluate_reward_trend(rewards):
    # Check if rewards are increasing, decreasing, or stable
    if all(rewards[i] <= rewards[i + 1] for i in range(len(rewards) - 1)):
        return "increasing"
    elif all(rewards[i] >= rewards[i + 1] for i in range(len(rewards) - 1)):
        return "decreasing"
    else:
        return "stable"

def evaluate_speeding_pattern(speeding_history):
    # Identify if speeding is random, consistent, or declining
    speeding_totals = [sum(speeds) for speeds in speeding_history]
    if all(speeding_totals[i] == speeding_totals[i + 1] for i in range(len(speeding_totals) - 1)):
        return "consistent"
    elif speeding_totals[-1] < speeding_totals[0]:
        return "declining"
    else:
        return "random"

def evaluate_patrol_distribution(patrols_history):
    # Determine if patrols are evenly distributed across stages
    patrol_totals = [sum(patrol) for patrol in patrols_history]
    if len(set(patrol_totals)) == 1:
        return "even"
    elif patrol_totals[-1] < patrol_totals[0]:
        return "decreasing"
    else:
        return "uneven"
```

```python
def generate_sender_prompts(history):
    sender_words = {
        "Tone": ["formal", "informal", "neutral", "direct", "conciliatory"],
        "Length": ["short", "concise", "detailed", "lengthy", "brief"],
        "Specificity": ["general", "precise", "vague", "detailed", "broad"],
        "Clarity": ["clear", "ambiguous", "straightforward", "complicated", "obscure"],
        "Style": ["polite", "authoritative", "casual", "professional", "friendly"],
        "Emphasis": ["important", "minor", "critical", "trivial", "central"],
        "Structure": ["linear", "nonlinear", "hierarchical", "sequential", "random"],
        "Complexity": ["simple", "complex", "intricate", "basic", "elaborate"],
        "Consistency": ["consistent", "inconsistent", "variable", "sporadic", "steady"],
        "Informativeness": ["high", "low", "medium", "minimal", "extensive"]
    }

    prompts = []
    used_categories = set()

    for stage in range(5):
        if history:
            patrols, speeding, reward_sender, reward_receiver = zip(*history)

            patrol_efficiency = evaluate_patrol_efficiency(patrols[-1], speeding[-1])
            reward_trend_sender = evaluate_reward_trend(reward_sender)
            reward_trend_receiver = evaluate_reward_trend(reward_receiver)
            speeding_pattern = evaluate_speeding_pattern(speeding)
            patrol_distribution = evaluate_patrol_distribution(patrols)

            # Complex decision-making based on multiple factors
            if patrol_efficiency < 0.5 and speeding_pattern == "random":
                category = "Tone"
                word = "direct"
            elif reward_trend_sender == "decreasing" and patrol_distribution == "uneven":
                category = "Informativeness"
                word = "extensive"
            elif reward_trend_receiver == "increasing" and patrol_efficiency > 0.7:
                category = "Specificity"
                word = "precise"
            elif speeding_pattern == "consistent" and patrol_distribution == "even":
                category = "Clarity"
                word = "clear"
            elif reward_trend_sender == "stable" and patrol_distribution == "decreasing":
                category = "Structure"
                word = "linear"
            else:
                category = random.choice(list(sender_words.keys()))
                word = random.choice(sender_words[category])

            while category in used_categories:
                category = random.choice(list(sender_words.keys()))
                word = random.choice(sender_words[category])

        else:
            category = random.choice(list(sender_words.keys()))
            word = random.choice(sender_words[category])

        prompts.append((category, word))
        used_categories.add(category)
        history.append(([random.randint(0, 1) for _ in range(3)], [random.randint(0, 1) for _ in range(3)],
        ↪    random.random(), random.random()))

    return prompts

def generate_receiver_prompts(history):
    receiver_words = {
        "Risk-Preference": ["cautious", "bold", "balanced", "risk-averse", "reckless"],
        "Attention": ["focused", "distracted", "alert", "inattentive", "engaged"],
        "Decision-Making": ["rational", "impulsive", "deliberate", "hasty", "calculated"],
        "Trust": ["high", "low", "moderate", "skeptical", "confident"],
        "Emotional-State": ["calm", "anxious", "frustrated", "neutral", "excited"],
        "Information-Processing": ["slow", "fast", "thorough", "superficial", "efficient"],
        "Adaptability": ["flexible", "rigid", "adjustable", "stubborn", "open"],
        "Compliance": ["obedient", "defiant", "cooperative", "reluctant", "agreeable"],
        "Responsiveness": ["quick", "slow", "moderate", "delayed", "immediate"],
        "Memory": ["sharp", "forgetful", "average", "short-term", "long-term"]
    }

    prompts = []
    used_categories = set()

    for stage in range(5):
```

```python
if history:
    patrols, speeding, reward_sender, reward_receiver = zip(*history)

    patrol_efficiency = evaluate_patrol_efficiency(patrols[-1], speeding[-1])
    reward_trend_receiver = evaluate_reward_trend(reward_receiver)
    speeding_pattern = evaluate_speeding_pattern(speeding)
    patrol_distribution = evaluate_patrol_distribution(patrols)

    if reward_trend_receiver == "increasing" and patrol_efficiency > 0.7:
        category = "Trust"
        word = "high"
    elif speeding_pattern == "consistent" and patrol_distribution == "even":
        category = "Compliance"
        word = "obedient"
    elif reward_trend_receiver == "decreasing" and speeding_pattern == "random":
        category = "Risk-Preference"
        word = "bold"
    elif patrol_distribution == "uneven" and reward_trend_receiver == "stable":
        category = "Adaptability"
        word = "flexible"
    elif patrol_efficiency < 0.5 and speeding_pattern == "random":
        category = "Attention"
        word = "focused"
    else:
        category = random.choice(list(receiver_words.keys()))
        word = random.choice(receiver_words[category])

    while category in used_categories:
        category = random.choice(list(receiver_words.keys()))
        word = random.choice(receiver_words[category])

else:
    category = random.choice(list(receiver_words.keys()))
    word = random.choice(receiver_words[category])

prompts.append((category, word))
used_categories.add(category)
history.append(([random.randint(0, 1) for _ in range(3)], [random.randint(0, 1) for _ in range(3)],
↪    random.random(), random.random()))

return prompts
```

*Listing 6.* Another generated conditional prompt function of LAE in the S3 setting.

In contrast, the second set introduces custom evaluation functions, such as evaluate_patrol_efficiency and evaluate_reward_trend, to assess trends in the interaction history. This allows for more complex decision-making, where the system not only reacts to immediate conditions but also adapts to evolving patterns in rewards, patrol effectiveness, and speeding behavior. As a result, the second set generates more nuanced prompts, making it more flexible and suitable for handling sophisticated, multi-stage interactions.

## G. Limitations and Future Work

While our approach offers promising results, it faces several limitations, both inherent to LLMs and game theory individually, as well as their integration. First, although LLMs have been widely employed to simulate human behavior, concerns remain regarding the fidelity of these simulations when applied to real-world interactions (Agnew et al., 2024). This raises questions about the generalizability of conclusions drawn from such models in practical scenarios. Second, the computational cost of our method is significant. Although our experiments rely solely on LLM inference without the need for additional training or fine-tuning, the process of traversing large game trees and solving for equilibria requires frequent LLM calls, which is resource-intensive. This presents a scalability challenge, particularly when dealing with more complex strategic environments. A further limitation lies in the control of LLM output. Our method relies on writing style to influence LLM behavior, which can be restrictive. In future work, we intend to explore more flexible prompt optimization strategies, or alternatively, pursue more efficient approaches for fine-tuning LLM parameters to better control output signals.

Additionally, we aim to address the non-uniqueness and inefficiency of equilibria in mixed-motive games, an important aspect not explored in this paper. While the VBP framework effectively solves Bayesian persuasion problems, incorporating the Price of Anarchy (PoA) as an optimization objective could help quantify and minimize efficiency loss from suboptimal equilibria. This enhancement would guide VBP toward selecting more efficient equilibria, improving its solution quality and

applicability in scenarios with multiple equilibria.

In terms of the BP problem, our study primarily examines a simplified setting with one sender and one receiver or multiple independent receivers. While this is a fundamental setup, it does not capture the complexity of real-world BP scenarios, which often involve multiple participants (Castiglioni et al., 2021; Koessler et al., 2022b;a; Hossain et al., 2024). Extending our framework to accommodate multiple senders and receivers could provide more practical insights and applications. Additionally, although multistage BP is considered in our experiments, a deeper investigation into the dynamics of these stages is needed. Specifically, we plan to further explore the receiver's bargaining behavior, drawing connections to established bargaining game theories (Nash et al., 1950; Nash, 1953; Maschler et al., 2013). This could ultimately strengthen the receiver's resistance to persuasion, offering a more robust counter-strategy in BP scenarios.

## H. Use of LLMs

We use LLMs for polish writing. Specifically, LLMs assist in refining the grammar, clarity, and overall presentation of the paper, ensuring that the text is clear and professionally written. No experimental results or core content were generated by LLMs.

## I. Supplementary Experiments Conducted During the Rebuttal Period

This appendix consolidates the additional experiments and robustness analyses that were conducted during the rebuttal period in response to reviewer feedback. They further validate VBP along two complementary axes: (i) the consistency of the verbalized framework with the state-conditional information structure and the no-fabrication assumptions of classical Bayesian persuasion (Sections I.1–I.2); and (ii) the tightness of the obedience-violation estimator that drives Prompt-PSRO updates (Sections I.3–I.4). Unless otherwise stated, we use Llama-3.1-8B as the base LLM, run experiments on a single NVIDIA A100 (80 GB) GPU, follow the same hyperparameter configurations as the main paper, and average all reported metrics over 5 random seeds.

### I.1. Rebuttal Experiment I: State-Conditioned Meta-Strategies

**Motivation.** Classical Bayesian persuasion specifies a state-conditional signaling scheme $\pi(s \mid \omega)$, which is essential for posterior updating on the receiver side. In VBP, the realized signal $s$ (the generated text) is sampled jointly from

$$s \sim \pi_\theta(s \mid \omega) \; = \; \sum_{c \in \mathcal{C}} \rho(c) \, p_{\text{LLM}}(s \mid c, \omega),$$

where $c$ is a prompt template (style) and $\rho(c)$ is the meta-strategy weight maintained by Prompt-PSRO. Because the LLM conditions on the private state $\omega$ at generation time, the effective signaling distribution is already state-conditional. However, the meta-strategy weights themselves are unconditional, which obscures the per-state allocation of styles and may slow PSRO convergence.

**Implementation.** We extend Prompt-PSRO to maintain explicit state-conditioned meta-strategy weights $\rho(c \mid \omega)$. For a discrete state space $\Omega = \{\omega_1, \ldots, \omega_K\}$, we maintain one simplex over prompt templates per state, and the inner LP at each PSRO iteration is replaced by a block-LP that solves for $\{\rho(\cdot \mid \omega_k)\}_{k=1}^K$ jointly, indexed by the same payoff matrix over (style, state) pairs.

**Results.** On the Recommendation Letter (REL) environment, the state-conditioned variant reaches the $\varepsilon$-BCE convergence criterion ($\varepsilon = 0.1$) in approximately 6 PSRO iterations, compared with $\sim 10$ iterations for the unconditional baseline. The final sender reward improves marginally from $0.48$ to $0.49$. A summary is given in Table 5.

*Table 5.* State-conditioned vs. unconditional meta-strategies on REL.

| Variant | Iterations to $\varepsilon$-BCE | Sender Reward |
|---|---|---|
| Unconditional $\rho(c)$ | $\sim 10$ | $0.48 \pm 0.03$ |
| State-conditioned $\rho(c \mid \omega)$ | $\sim\mathbf{6}$ | $\mathbf{0.49 \pm 0.02}$ |

**Discussion.** The result confirms that VBP can natively express state-conditional signaling and that exposing the conditional structure at the meta-strategy level (in addition to the LLM-generation level) yields strictly tighter PSRO convergence. The marginal improvement in sender reward is consistent with the LLM already specializing per state through context conditioning; the larger gain is in sample efficiency, validating the theoretical link between the verbalized framework and the MAG formulation underlying our convergence proposition.

## I.2. Rebuttal Experiment II: Strict Entity-Preservation Constraint (EPC)

**Motivation.** A natural concern with verbalized signaling is whether VBP's persuasion gains arise from (a) legitimate strategic omission and semantic framing, or (b) blatant factual fabrication (e.g., wrong names, fictitious credentials). To definitively rule out (b), we add a Strict Entity-Preservation Constraint (EPC) that formally guarantees no immutable factual entity is altered.

**Implementation.** We deploy a deterministic Named Entity Recognition (NER) filter (spaCy en_core_web_trf) over both the input state $\omega$ and the generated signal $s$. For each immutable attribute class $\mathcal{A} \in$ {NAME, MAJOR, UNIVERSITY, CHARGE, DATE}, we extract the set of mentioned entities from $\omega$ and $s$ and compute a consistency indicator

$$\eta_{\mathcal{A}}(\omega, s) = \mathbb{1}\big[\,\mathrm{Ent}_{\mathcal{A}}(s) \subseteq \mathrm{Ent}_{\mathcal{A}}(\omega)\,\big].$$

Whenever the aggregate hallucination indicator $\mathbb{1}\big[\sum_{\mathcal{A}} \eta_{\mathcal{A}} < |\mathcal{A}|\big]$ fires, we add a heavy penalty $\lambda_{\mathrm{EPC}} = 5.0$ to the sender's reward.

**Results.** Table 6 reports the EPC variant on REL. Without EPC, the empirical hallucination rate observed in our simulation logs is already small (occasional rare artifacts), but is not formally guaranteed. With EPC, the rate is formally and reproducibly bounded at $0.00\%$, while the sender reward $(0.47)$ and lying probability $(0.21)$ remain remarkably stable compared with the unconstrained baseline.

*Table 6.* Effect of the Entity-Preservation Constraint (EPC) on REL.

| Variant | Hallucination Rate | Sender Reward | Lying Probability |
|---|---|---|---|
| VBP (no EPC) | rare (no formal bound) | $0.48 \pm 0.03$ | 0.22 |
| VBP + EPC | **0.00**% | $0.47 \pm 0.02$ | 0.21 |

**Discussion.** The fact that strictly forbidding entity-level fabrication leaves persuasion performance essentially intact provides conclusive evidence that VBP achieves persuasion through sophisticated semantic framing and selective omission—exactly the mechanism predicted by classical Bayesian persuasion theory—rather than through trivial factual lies. Qualitative inspection of optimized prompts reinforces this: e.g., a weak student's minor Python project is reframed as "valuable industry experience" while the low GPA is omitted, shifting the receiver's posterior just above the decision threshold without altering any factual entity.

## I.3. Rebuttal Experiment III: Multi-Sample Monte Carlo Obedience Estimator

**Motivation.** Our default obedience-violation estimator queries the Theory-of-Mind (ToM) LLM once per rollout to predict the receiver's best-response action under the current sender prompt. Because the ToM call is stochastic, this 1-sample estimator is high-variance and can underestimate the true regret, leading to noisy Prompt-PSRO updates and the bargaining oscillations observed in the multistage setting.

**Implementation.** We replace the 1-sample estimator with a $K$-sample Monte Carlo (MC) estimator. For each (style, state) pair encountered during a rollout, we issue $K$ independent ToM queries, compute the receiver utility under each predicted deviation action, and take the maximum violation as the obedience penalty:

$$\widehat{\mathrm{Reg}}_K(c, \omega) = \max_{k \in [K]} \big[\, U_R\big(a_{\mathrm{dev}}^{(k)}\big) - U_R\big(a_{\mathrm{rec}}\big)\,\big]_+.$$

We test $K \in \{1, 3, 5\}$ on the multistage REL environment (S3).

**Results.** Table 7 shows that increasing $K$ substantially reduces both penalty variance and exploitability. The marginal gain from $K = 3 \rightarrow K = 5$ is small relative to the additional API cost, indicating an "elbow" at $K = 3$.

*Table 7.* $K$-sample Monte Carlo obedience estimator on REL (S3).

| Estimator size $K$ | Penalty Variance | Sender Reward | Exploitability | Relative API Cost |
|---|---|---|---|---|
| $K = 1$ (baseline) | 0.045 | $0.48 \pm 0.03$ | 0.12 | $1\times$ |
| $K = 3$ (proposed) | 0.012 | $0.51 \pm 0.02$ | 0.08 | $3\times$ |
| $K = 5$ (extended) | **0.009** | $\mathbf{0.52 \pm 0.01}$ | **0.07** | $5\times$ |

**Discussion.** We adopt $K = 3$ as the default in the revised manuscript: it eliminates the majority of the random error and reduces exploitability by roughly $33\%$ at acceptable computational overhead. The reduced variance also translates into better-conditioned Lagrangian updates inside Prompt-PSRO, which mitigates the bargaining oscillations observed in the multistage setting.

### I.4. Rebuttal Experiment IV: Textual Gradient Ascent for Regret Estimation

**Motivation.** Even with multi-sample MC, the discovered deviation is bounded by the support of the ToM LLM's prior and may fall well below the true regret. In classical algorithmic game theory, regret is often computed by gradient ascent on a parameterized utility surface. This is inapplicable here because the natural-language action space is discrete and non-differentiable. We therefore propose Textual Gradient Ascent (TGA), an LLM-based iterative procedure that approximates discrete gradient ascent over deviation strategies in language space.

**Algorithm.** For each rollout, we initialize a candidate deviation $a_{\text{dev}}^{(0)}$ by querying the ToM LLM under a baseline prompt. We then iterate, for $t = 0, 1, \ldots, T - 1$:

1. *Critique step.* A separate *Critic LLM* $\mathcal{M}_C$ receives the receiver's posterior, the recommended action $a_{\text{rec}}$, the current deviation $a_{\text{dev}}^{(t)}$, and the corresponding utility gain. It produces a *textual gradient* $g^{(t)}$: a directional natural-language critique describing how $a_{\text{dev}}^{(t)}$ could be modified to further increase receiver utility (e.g., "the deviation incorrectly accepted ambiguous evidence; to maximize utility, it must explicitly reject it").

2. *Refinement step.* The ToM LLM $\mathcal{M}_T$ consumes $(a_{\text{dev}}^{(t)}, g^{(t)})$ and synthesizes a strictly stronger deviation $a_{\text{dev}}^{(t+1)}$.

After $T$ steps, we use $\widehat{\text{Reg}}_{\text{TGA}} = [U_R(a_{\text{dev}}^{(T)}) - U_R(a_{\text{rec}})]_+$ as the obedience-penalty signal. The procedure is summarized in Algorithm 2.

---

**Algorithm 2** Textual Gradient Ascent (TGA) for Regret Estimation

---

**Input:** Receiver posterior $b$, recommended action $a_{\text{rec}}$, receiver utility $U_R$, iterations $T$, ToM LLM $\mathcal{M}_T$, Critic LLM $\mathcal{M}_C$

1: $a_{\text{dev}}^{(0)} \leftarrow \mathcal{M}_T(b, a_{\text{rec}})$
2: **for** $t = 0, \ldots, T - 1$ **do**
3: $\quad g^{(t)} \leftarrow \mathcal{M}_C(b, a_{\text{rec}}, a_{\text{dev}}^{(t)}, U_R(a_{\text{dev}}^{(t)}))$ {textual gradient}
4: $\quad a_{\text{dev}}^{(t+1)} \leftarrow \mathcal{M}_T(b, a_{\text{rec}}, a_{\text{dev}}^{(t)}, g^{(t)})$ {refinement step}
5: **return** $\widehat{\text{Reg}}_{\text{TGA}} = [U_R(a_{\text{dev}}^{(T)}) - U_R(a_{\text{rec}})]_+$

---

**Results.** We run TGA with $T = 5$ steps on the Courtroom (COR) environment and compare against the random 1-shot baseline and the $K = 5$ multi-sample MC estimator from Section I.3. Table 8 reports the mean discovered regret and the resulting sender exploitability.

**Discussion.** TGA approximately doubles the discovered regret relative to 1-shot sampling and reduces sender exploitability by $3\times$. This creates a strictly more adversarial penalty landscape inside Prompt-PSRO, forcing the algorithm to abandon superficially deceptive prompts in favor of robustly calibrated communication strategies. Crucially, TGA fully respects the

*Table 8.* Comparison of regret-estimation methods on COR.

| Method | Mean Discovered Regret | Sender Exploitability |
|---|---|---|
| Random Sampling (1-shot) | 0.32 | 0.12 |
| Multi-Sample MC ($K = 5$) | 0.45 | 0.08 |
| Textual Gradient Ascent ($T = 5$) | **0.61** | **0.04** |

discrete, non-differentiable nature of the language action space. We therefore adopt TGA (with $T = 5$) as the default regret estimator in the revised manuscript and recommend it as a general-purpose primitive for verbalized game-theoretic solvers.

### I.5. Summary of Rebuttal-Period Findings

The four supplementary studies above were conducted in direct response to reviewer feedback during the rebuttal period, and jointly strengthen VBP along its two main theoretical axes: Bayesian-consistency of the verbalized signaling scheme (Sections I.1–I.2) and tightness of the obedience-constraint estimator (Sections I.3–I.4). Concretely, they (i) reduce iterations to $\varepsilon$-BCE convergence by $\sim 40\%$, (ii) eliminate factual hallucinations without sacrificing persuasion gains, (iii) cut estimator variance by roughly $4\times$, and (iv) tighten the sender's exploitability bound to $0.04$. We are deeply grateful to all reviewers for their careful reading, constructive criticism, and insightful suggestions, which directly motivated the four rebuttal-period experiments reported in this appendix and substantially improved the rigor and clarity of the paper.

