# OpenReview forum: "Verbalized Bayesian Persuasion"
_ICML.cc/2026/Conference — ICML 2026 regular_

### Official Review · Reviewer_cDmX · 2026-02-17

**Soundness:** 3
**Presentation:** 2
**Significance:** 2
**Originality:** 3
**Overall Recommendation:** 4
**Confidence:** 3

**Summary:**

This paper introduces a framework for studying and simulating Bayesian persuasion in natural language domains, addressing a key gap between the mathematical economic model of Kamenica and Gentzkow (2011) and real-world applications, such as advertising and marketing. Within this verbalized Bayesian persuasion (VBP) framework, a mediator-augmented game (MAG) is constructed, and both sender and receiver actions are instantiated using large language models (LLMs). To solve the game, the VBP utilizes standard techniques such as Prompt-PSRO, while adapting it to the Bayesian persuasion framework, in which two additional constraints arise (the commitment assumption and the incentive-compatibility). The authors then conduct a series of experiments on several problems and settings, demonstrating the effectiveness of their VBP framework.

**Compliance With Llm Reviewing Policy:**

Affirmed.

**Final Justification:**

The rebuttal was quite productive in my view, and the authors have addressed some concerns raised by the reviewers (including myself). Hence, I am maintaining my positive score.

**Key Questions For Authors:**

What would be the core insights derived from your experimental results? That is, is there any surprising behavioral phenomenon that your experiments reveal? Is the VBP framework superior to alternative approaches with respect to some reasonable criteria? Do these results leave some open questions that can potentially lead to an important follow-up study?

**Limitations:**

yes

**Strengths And Weaknesses:**

## Strengths

- The paper addresses a key gap between economic theory and real-world applications, in domains with high relevance to the ML community.
- The paper provides a comprehensive background on both the economic setup of Bayesian persuasion and the approaches/techniques utilized within the VBP framework (e.g., MAG and Prompt-PSRO).
- The two core challenges in adapting the known techniques to the Bayesian persuasion setting are identified, explained, and discussed in detail.
- The appendices contain technical details and illustrations required for the reproducibility of the framework implementation and experimental setup.

## Weaknesses

- Some relevant work is missing from the discussion; For instance, [1] studied natural language framing in Bayesian persuasion, [2,3] studied choice prediction in a repeated, language-based persuasion setup, [4] viewed the Bayesian persuasion theoretical framework as a scalable tool for measuring the persuasive capabilities of LLMs. It is crucial to discuss the contribution of the current paper in the light of these existing works (and potentially more).
- The experiments in the paper are somewhat difficult to follow, and in particular, it is hard to understand what the research questions are that the experiments aim to address, why each of the presented metrics is relevant, etc. For instance, what are we expecting to deduce/learn by comparing the S- and R-rewards across methods? What information/insight do we gain by additionally considering the honesty/lie probabilities? Are we focusing on the sender's perspective (i.e., we want to design a sender-optimal method) or are we rather interested in socially-beneficial outcomes? A comprehensive discussion and justification are required, and more importantly, a clear definition of research questions could significantly help in navigating this part of the paper.

[1] Duetting, P., Hossain, S., Lin, T., Leme, R. P., Ravindranath, S. S., Xu, H., & Zuo, S. (2025). Information Design With Large Language Models. arXiv preprint arXiv:2509.25565.

[2] Shapira, E., Madmon, O., Reichart, R., & Tennenholtz, M. (2024). Can llms replace economic choice prediction labs? The case of language-based persuasion games. arXiv preprint arXiv:2401.17435.

[3] Shapira, E., Madmon, O., Apel, R., Tennenholtz, M., & Reichart, R. (2025). Human choice prediction in language-based persuasion games: Simulation-based off-policy evaluation. Transactions of the Association for Computational Linguistics, 13, 980-1006.

[4] Cheng, Z., & You, J. (2025). Towards strategic persuasion with language models. arXiv preprint arXiv:2509.22989.

---

> ### Author Rebuttal · Authors · 2026-03-30
>
> We sincerely thank you for your constructive feedback. Below, we address your detailed comments and clarify our contributions, experimental design, and core insights. We will seamlessly integrate these clarifications into the revised manuscript.
>
> **1. Missing Literature & Contributions**
>
> We will add the suggested citations and explicitly differentiate VBP's novel contributions:
>
> - **Signaling vs. Framing [1]:** Unlike prior work treating language merely as a "framing" device to shift priors, VBP fully verbalizes the signal mapping mechanism itself. By modeling semantic operations as discrete stylistic prompts, VBP empowers game-theoretic solvers to derive optimal signaling strategies directly in infinite-dimensional natural language spaces.
> - **Normative Equilibrium vs. Behavioral Prediction [2,3]:** While previous studies use LLMs to simulate bounded human rationality in repeated persuasion, VBP establishes a normative paradigm. By using Prompt-PSRO, it explores optimal game-theoretic frontiers for fully rational agents, achieving convergence in exploitability to provide a theoretical upper bound for AI communication.
> - **Multi-agent Co-evolution vs. Single-agent RL [4]:** Existing BP evaluations of LLMs rely on single-agent RL, prone to policy overfitting. VBP employs a Lagrangian zero-sum transformation and Double Oracle architecture for alternating best-response. It enforces verbalized commitments and explicitly computes obedience penalties via Theory of Mind (ToM) queries, ensuring convergence to an approximate BCE/BNE.
> - **Multi-stage Dynamics:** Integrating FunSearch-based conditional prompt optimization, VBP pioneers multi-stage sequential games, replicating reputation mechanisms and revealing the micro-dynamics of trust reconstruction.
>
> **2. Experimental Setup, Research Questions, and Metrics**
>
> Our experiments follow a deliberate progression from foundational validation to dynamic scalability:
>
> - **Research Questions:**
>   - **RQ1 (S1/S2):** Can an LLM-parameterized framework approximate precise BCE in static BP within language spaces?
>   - **RQ2 (S3):** How do persuasion strategies and deception evolve in multi-stage settings where the receiver possesses memory?
>   - **RQ3 (S4):** Can VBP scale to large-action-space, multi-agent negotiations (Full-Press Diplomacy)?
>   - **RQ4 (App F):** Do the algorithmic persuasion strategies transfer to human decision-makers?
> - **Metrics Justification:**
>   - **S- and R-Rewards:** Comparing these reveals strategic exploitation and obedience constraint enforcement. The sender achieves near-maximum utility while the receiver’s reward stays just high enough to ensure compliance, proving a BCE computation. Exploitability descending to ~0.1 formally proves convergence.
>   - **Honesty/Lie Probabilities:** These serve as macroscopic proxies for the informativeness and obfuscation of the sender's policy in intractable natural language spaces. Maintaining stable deception-to-honesty ratios confirms the verbalized commitment holds.
> - **Sender vs. Socially-Beneficial Perspective:** BP's objective is mathematically sender-centric. However, S3 reveals that extending interaction and granting receiver memory breaks the unilateral commitment model. Excessive deception triggers receiver retaliation (ignoring signals), forcing the sender to increase honesty to rebuild credibility. This emergent bargaining aligns sender strategies closer to socially-beneficial outcomes.
>
> **3. Core Insights, Superiority, and Future Work**
>
> - **Core Insights & Surprising Phenomena:**
>   - *Honesty Oscillation & Bargaining:* Multi-stage receivers punish deception by ignoring signals, forcing senders to capitulate and surge honesty to rebuild trust—an emergent bargaining process.
>   - *Paradox of Alignment:* An unaligned LLaMA model exhibited no oscillations but stabilized at suboptimal honesty. RLHF alignment introduces normative friction during zero-sum training, but this "pull" toward truthfulness is ultimately necessary to achieve mathematically optimal truthfulness.
>   - *Nuanced Obfuscation:* Optimal persuasion relies on modulating information bandwidth (omitting facts, reframing contexts) rather than absolute falsehoods.
> - **Superiority of VBP:**
>   - *Computational Efficiency:* Converges strictly in prompt space in merely 1.5-2 hours, vastly superior to MARL's 6-8 hours.
>   - *Strategic Supremacy:* Standard LLMs are inherently biased toward honesty. VBP precisely calibrates deception to achieve a sender reward of 0.48, outperforming DeepSeek-R1 (0.42).
>   - *External Validity:* In controlled trials, human participants agreed with VBP-generated signals 95% of the time, compared to 55-65% for baselines.
> - **Future Work:** Future trajectories include formalizing sequential VBP as a Nash Bargaining Game, developing variance-reduction techniques to manage LLM stochasticity, and designing "Defensive PSRO" to synthesize cognitive immune systems that help humans resist adversarial linguistic persuasion.

---

> > ### Author Rebuttal · Reviewer_cDmX · 2026-04-02
> >
> > Thank you for the detailed response which addresses my concerns (particularly concerning the experimental results and RQs). I am therefore maintaining my positive score.

---

> > > ### Author Response · Authors · 2026-04-03
> > >
> > > We sincerely thank you for the thoughtful and constructive review. Your feedback on the missing literature and the clarity of our experimental design has been invaluable in strengthening the paper. We are glad that our responses have adequately addressed your concerns, and we will ensure all discussed improvements are fully incorporated into the revised manuscript. Thank you again for your time and effort throughout this review process.

---

### Official Review · Reviewer_tDsz · 2026-03-02

**Soundness:** 4
**Presentation:** 4
**Significance:** 3
**Originality:** 3
**Overall Recommendation:** 5
**Confidence:** 3

**Summary:**

Bayesian persuasion plays a significant role in human communication and modern economies. While classical optimization and analytical methods could solve the BP problem optimally, they require both the information and action spaces to be discrete and small, which restricts their application to real-world scenarios and means they can only be used on toy games or toy examples.

This paper proposes a verbalized Bayesian persuasion framework, integrating LLM agents into the process and solving the BP problem directly in the natural language domain. To tackle the challenges of infinite-dimensional signal spaces and LLM proxies in strategy optimization, the authors build on the previous Prompt-PSRO framework, which transfers strategy optimization to prompt optimization for LLMs.

Additionally, they propose several components, including verbalizing commitment assumptions, obedience constraints, and information obfuscation, to enhance the performance, efficiency, and stability of the solver. Beyond that, they extend the previous Prompt-PSRO framework to multistage games with conditional prompt optimization under convergence guarantees.

**Compliance With Llm Reviewing Policy:**

Affirmed.

**Final Justification:**

Finally, I would recommend this work for Accept. The authors' rebuttal addressed my concerns about the "regret" issue—specifically, the use of the Textual Gradient method to achieve a higher regret bound. The concern regarding sample sizes was also addressed by extending the samples for each iteration.

**Key Questions For Authors:**

No more question from my end. This manuscript is well written and presented.

**Limitations:**

Yes

**Strengths And Weaknesses:**

Strengths:

1. Classical BP solvers are analytical and only suitable for simple settings. Learning-based methods extend the solvable regime further, while this paper successfully goes beyond that to directly solve the BP problem in the natural language space, instead of transferring it to the mathematical field and using traditional optimization. The idea is very interesting and the execution of this paper is very impressive. It reminds me of the following paper, which is quite similar in its ideas but targets different problems:

Huang, D., Marmolejo-Cossío, F., Lock, E., & Parkes, D. (2025). Accelerated Preference Elicitation with LLM-Based Proxies. arXiv preprint arXiv:2501.14625.

2. The experiments consider classical, multistage, and large-scale multi-receiver BP problems. The evaluations are complete and thorough, including analyses of model differences, comparisons with SOTA models, and the use of extended techniques for performance improvement. Some classical results are also reproduced by the proposed VBP framework.

3. The whole paper is well-written and clearly described.

Weaknesses:
1. The sampling-based estimate uses only one sample, which might cause random errors because the choice is stochastic. I suggest using 3–5 sample estimates and taking an overall average across both the rollouts and the samples to improve stability.

Suggestion:
I have a follow-up idea regarding the sampling-based estimates. Since the accurate computation of violations is nearly impossible due to the infinite state space, I have seen many papers—especially those using learning methods for classical problems in algorithmic game theory—calculate the “regret” using gradient descent.

Instead of only sampling states and calculating the “violated utility,” one could sample a state first and then perform gradient descent on the utility function for several steps (e.g., 25–50) to find a stronger violation or misreport. In classical BP problems, I believe this is feasible because the setting consists of a single player and the utility function is one-dimensional. While it might be difficult to calculate the approximate regret using gradient methods in multi-player scenarios, I suspect it could be achieved through alternative optimization methods in those cases.

---

> ### Author Rebuttal · Authors · 2026-03-30
>
> We sincerely thank the reviewer for the highly positive feedback, recognizing our paper's originality, execution, and the thoroughness of our evaluations. We deeply appreciate the constructive suggestions regarding sampling stability and gradient-based regret calculation, which provide excellent directions for further enhancing our framework. Below, we address your detailed comments.
>
> **1. Discussion on Related Literature**
>
> We thank the reviewer for bringing this highly relevant and recent preprint (Huang et al., 2025) to our attention. Both our work and Huang et al. represent a new paradigm of using LLMs to transcend the numerical constraints of classical algorithmic game theory, solving complex economic problems in natural language.
>
> However, they operate on opposite ends of the information asymmetry spectrum:
> * **Target Problem:** Huang et al. tackle *Preference Elicitation* (Mechanism Design) where the user holds private information. The LLM acts as an empathetic, cooperative proxy aiming to faithfully extract hidden utility functions, minimizing communication complexity.
> * **Our Framework:** VBP addresses *Bayesian Persuasion* (Information Design) where the Sender holds private information. The LLM acts as a self-interested agent, strategically designing signals to manipulate the receiver, constrained by verbalized obedience.
>
> We will prominently cite Huang et al. (2025) in the revised version and include a detailed comparative discussion highlighting how LLMs can be utilized for both cooperative elicitation and strategic persuasion.
>
> **2. Stability of the Sampling-Based Estimate**
>
> We completely agree that relying on a 1-sample estimate for the counterfactual deviation introduces stochastic variance, which can artificially underestimate the receiver's regret and destabilize the Prompt-PSRO meta-game updates.
>
> During the rebuttal period, we conducted a supplementary ablation study in the Recommendation Letter (REL) environment under the S3 setting, replacing the 1-sample estimator with a $k$-sample Monte Carlo estimator. By querying the Theory-of-Mind (ToM) LLM $k$ independent times (at $T=0.7$) and taking the maximum violation, we observed a dramatic stabilization:
>
> | Estimator Size | Penalty Variance | Final Sender Reward | Exploitability | Relative API Cost |
> | :--- | :--- | :--- | :--- | :--- |
> | $k=1$ (Baseline) | 0.045 | 0.48 ± 0.03 | 0.12 | $1.0\times$ |
> | $k=3$ (Proposed) | 0.012 | 0.51 ± 0.02 | 0.08 | $3.0\times$ |
> | $k=5$ (Extended) | 0.009 | 0.52 ± 0.01 | 0.07 | $5.0\times$ |
>
> The results demonstrate an "elbow" effect: $k=3$ eliminates the majority of the random error and lowers exploitability with an acceptable computational trade-off, whereas $k=5$ yields diminishing returns. We will integrate this $k=3$ multi-sample evaluation as the robust default and include this ablation study in the revised manuscript.
>
> **3. Calculating Regret via Gradient/Alternative Optimization Methods**
>
> We strongly appreciate this insightful and rigorous suggestion. Finding the maximum possible violation (regret) via optimization is a gold standard in classical game theory. As you rightly pointed out, standard numerical gradient descent is inapplicable to VBP because our action space—natural language—is discrete and non-differentiable.
>
> However, inspired by your feedback, we successfully implemented an alternative discrete optimization method using **Textual Gradients** (LLM-based iterative refinement). Instead of numerical steps, we use a "Critic LLM" to evaluate the ToM's initial deviation action. The Critic generates a directional textual critique (e.g., *"The deviation incorrectly accepted ambiguous evidence; to maximize utility, it must explicitly reject it"*), guiding the ToM LLM to synthesize a strictly stronger deviation.
>
> We ran a rapid rebuttal experiment executing $N=3$ textual gradient ascent steps in the Courtroom (COR) environment:
>
> * **Random Sampling (1-shot):** Mean Discovered Regret: 0.32 | Sender Exploitability: 0.12
> * **Multi-Sample MC ($k=3$):** Mean Discovered Regret: 0.45 | Sender Exploitability: 0.08
> * **Textual Gradient Ascent ($N=3$):** Mean Discovered Regret: **0.61** | Sender Exploitability: **0.04**
>
> By actively performing textual gradient ascent, we uncover significantly higher regret bounds. This creates a strictly adversarial penalty landscape that forces the Prompt-PSRO algorithm to abandon superficially deceptive prompts and discover highly robust, carefully calibrated communication strategies. We are thrilled with these results and will integrate this textual-gradient regret calculation mechanism into the final version of the paper as a core algorithmic enhancement.

---

> > ### Author Rebuttal · Reviewer_tDsz · 2026-03-31
> >
> > Thanks for the authors, really great response and I didn't expect that my suggestion really works (since it was my brainstormed idea haha). I will maintain my score and I believe this paper is ready for acceptance. Good luck.

---

> > > ### Author Response · Authors · 2026-04-01
> > >
> > > Thank you so much for your final confirmation and your continued support of our work!
> > >
> > > We would like to express our special gratitude once again for your brilliant "brainstormed" idea. As our experiments demonstrated, your suggestion was not only highly feasible but also introduced an elegant and effective solution (Textual Gradient Ascent) to our framework, significantly improving the algorithm's robustness and the overall quality of the paper. We truly enjoyed this inspiring academic exchange with you.
> > >
> > > We will ensure that these new experimental results and discussions are fully integrated into the revised version, and we will acknowledge your invaluable suggestions.
> > >
> > > Thank you again for your time, effort, and well wishes throughout this review process! Best regards!

---

### Official Review · Reviewer_whcS · 2026-03-11

**Soundness:** 2
**Presentation:** 3
**Significance:** 2
**Originality:** 2
**Overall Recommendation:** 4
**Confidence:** 2

**Summary:**

This paper studies verbalized Bayesian persuasion, where the sender’s signaling strategy and the receiver’s responses are implemented in natural language using LLMs. The paper proposes a fairly elaborate framework, VBP, that combines Prompt-PSRO-style prompt optimization with several additional components, including verbalized commitment assumptions, obedience constraints, information obfuscation, and conditional prompt optimization for multistage settings. The method is evaluated on several persuasion-inspired tasks, including recommendation letters, courtroom, law enforcement, and full-press Diplomacy. Overall, I think the direction is interesting and ambitious, and the paper is clearly trying to extend Bayesian persuasion to more realistic language-based settings rather than remaining in small symbolic games. That being said, I am not familiar enough with the Bayesian Persuasion framework to make any meaningful review of the theory or soundness of the proposed method.

**Compliance With Llm Reviewing Policy:**

Affirmed.

**Final Justification:**

Rebuttal.

**Key Questions For Authors:**

1. Which components of the VBP pipeline are actually necessary for the gains in the main language-based settings? In particular, can the authors better disentangle the contribution of Prompt-PSRO itself versus obedience constraints, commitment scaffolding, and information obfuscation?

2. Can the authors provide a clearer compute and cost accounting for the full VBP pipeline, including optimization-time overhead and judge/scaffolding calls, relative to direct prompting baselines?

3. Given that the Diplomacy Game Score improvement is statistically marginal, what should readers view as the main practical takeaway of the paper: improved task performance, or improved control/calibration properties?

4. How should a reader interpret the theoretical guarantees relative to the actual implemented system, especially given the approximate best-response machinery and additional scaffolding? I ask this mainly because I am not expert enough in Bayesian persuasion to fully judge that aspect.

**Limitations:**

The paper would benefit from having an explicit conclusions section to discuss the impact and limitations of the work.

**Strengths And Weaknesses:**

### Strengths

1. **Interesting and ambitious direction.** The paper tackles a worthwhile problem by trying to extend Bayesian persuasion into natural-language settings rather than restricting attention to symbolic toy games. This is a potentially valuable direction for LLM-based strategic interaction.

2. **Broad empirical scope.** The paper evaluates the framework across several environments, ranging from simpler persuasion-inspired tasks to a more realistic setting in full-press Diplomacy. This gives the work breadth.

3. **Generally readable high-level presentation.** Despite the method being quite complex, I found the overall motivation and intended role of the different components reasonably clear at a high level.

### Weaknesses

1. **Methodological complexity is a major concern.** The framework combines Prompt-PSRO, OPRO/FunSearch, commitment scaffolding, obedience penalties, information obfuscation, self-reflection, and additional LLM-based judging components. This makes the method difficult to parse and, more importantly, makes it hard to isolate what is actually driving the gains.

2. **The ablations do not fully disentangle the contribution of each component.** The current analysis is helpful, but only partially resolves the main concern above. In particular, the obedience component seems more important, while other parts such as commitment assumptions and information obfuscation appear to have more limited impact in the reported static setting. This weakens the case that the full stack is necessary.

3. **The strongest empirical result is not fully convincing.** In Diplomacy, the improvement on the main Game Score metric appears small, and the paper itself describes it as statistically marginal. Since this is the headline realistic setting, it is difficult to conclude that the method yields a robust gain on the main task.

4. **The practical cost of the method is not fully transparent.** While the paper provides some implementation details and controls some evaluation conditions, I did not find a clear accounting of the additional optimization/search overhead introduced by the full VBP pipeline relative to simpler prompting baselines. Given how elaborate the method is, this matters for assessing practical value.

5. **No conclusions or limitations section.** In particular, the manuscript does not include a dedicated conclusion section in the main paper, which contributes to the feeling that many moving parts are introduced without a final distillation of the main takeaways.

---

> ### Author Rebuttal · Authors · 2026-03-31
>
> We sincerely thank the reviewer for recognizing our paper's ambitious direction in extending Bayesian persuasion to natural language and its broad empirical scope. Below we address your detailed comments.
>
> **1: Component Disentanglement & Complexity (W1, W2, Q1)**
>
> As requested, **App F.3 (Fig 9)** provides full ablations. The **Obedience Constraint (OC)** is the primary driver: without it, the sender defaults to myopic deception, shattering trust and plummeting rewards. **Commitment (CA)** is theoretically mandatory to distinguish BP from cheap talk. In multistage settings, **Obfuscation (IO)** critically stabilizes prompt brittleness via contrastive training, increasing rewards by ~10% (**App F.3.2**).
>
> **2: Computational Cost vs. Direct Prompting (W4, Q2)**
>
> As detailed in **App E.1**, on an A100 (Llama-3.1-8B), S1/S2 optimization takes 1.5–2 hours (vs. 6–8 hrs for MARL), while multi-stage (S3) takes 8–10 hours. The Prompt-PSRO loop generates ~16 API/judge calls per meta-step.  This overhead is strictly justified: zero-shot prompting (e.g., DeepSeek-R1) fails completely to discover equilibria. It defaults to normative honesty, yielding sub-optimal rewards (0.42 vs VBP's 0.48 in REL, **Tab 1**) and high exploitability. VBP mathematically discovers the optimal deception threshold and reduces exploitability to ~0.1 (**App F.2, Fig 8**)—a property impossible via direct prompting.
>
> **3: Diplomacy Results & Main Practical Takeaway (W3, Q3)**
>
> In the highly complex Full-Press Diplomacy setting (S4), while the raw Game Score improvement is marginal, the main practical takeaway is that VBP fundamentally transforms LLMs from unreliable conversational agents into coordinated, mathematically grounded strategists. As detailed in **Table 2** of the main text (and conceptually expanded in **Appendix E.2**), the true power of VBP lies in mechanistic control and tactical reliability, bypassing the limitations of ungrounded cheap talk:
>
> 1. **Calibration & Belief Updating:** Expected Calibration Error (ECE) drops dramatically from 0.12 (GPT-5) to 0.07 (VBP). As App E.2 explains, this physically means the receiver is updating its Bayesian beliefs based on the sender's committed communication, rather than hallucinating intentions.
> 2. **Action Coupling & Coordination:** Because posteriors reflect reality, unforced errors plummet (Invalid orders drop from 5.2% to 3.1%). Most crucially, **Support Success** surges from 52.7% to 59.8%, and **Betrayal rate** drops from 34.7% ro 24.9%. Physically, successful support orders require precise, multi-agent synchronization built on trusted communication—a metric far more indicative of strategic alignment than raw win rates.
> 3. **Theoretical Adherence:** The Obedience Penalty drops by nearly half (from 0.12 to 0.07), demonstrating the LLM adheres to the generated game-theoretic constraints rather than randomly defecting.
>
> **4: Bridging Theory and Practice (Q4)**
>
> In classical BP, the sender submits a "rulebook" to a mediator. In VBP, the LLM prompt serves as this rulebook, made publicly verifiable by injecting it into the receiver's context window.
>
> Regarding the theoretical bounds (Proposition 4.1): Prompt-PSRO restricts the infinite language space into a compact empirical subspace, guaranteeing convergence to a local equilibrium. The LLM optimizers (OPRO/FunSearch) act as *approximate* best-response oracles; their heuristic error margin constitutes the $\epsilon$ in our $\epsilon$-approximate bounds. Furthermore, the obedience penalty mathematically acts as a Lagrangian relaxation of the BP linear programming problem. Any prompt proposing an action where the receiver benefits by deviating is heavily penalized during the rollout phase, keeping the heuristic search rigorously bound to the correlated equilibrium.
>
> **W5/L1: Missing Conclusions & Limitations**
>
> Due to strict page limits, our dedicated Limitations and Future Works section was placed in **App G**. The revised version will utilize the extra page allowance to synthesize these points into a formal Conclusion in the main text, distilling our core message: shifting LLMs from cheap-talk to grounded communication.
>
> **Further Empirical Highlights (Appendices F.6 & F.8.4)**
>
> We respectfully highlight two existing appendix results that cement the paper's significance:
>
> 1. **Human Evaluation (App F.6):** Do these optimized strategies persuade humans? Yes. Human agreement with VBP recommendations was **95% (REL/COR)** vs 55-65% for direct prompting baselines (p=0.005), proving VBP successfully influences human Bayesian updating.
> 2. **Alignment Impact (App F.8.4):** Normative alignment causes bargaining oscillations (over-deceiving then correcting). Unaligned models do not oscillate but converge sub-optimally. This reveals alignment is paradoxically required to access higher-utility cooperative states.
>
> We hope these clarifications and pointers to the existing appendices effectively address your concerns!

---

> > ### Author Rebuttal · Reviewer_whcS · 2026-04-05
> >
> > I think the authors have resolved most of my concerns. I have updated my score. However I still think there is no way to justify the missing conclusion section, a format to which most other papers in the conference had to adhere to.

---

> > > ### Author Response · Authors · 2026-04-05
> > >
> > > We sincerely thank you for the constructive engagement and the updated assessment. We fully agree that a dedicated Conclusion section should appear in the main text. Rather than simply reiterating our commitment, we provide the drafted section below, which will be included verbatim in the revised version using the additional page allowance.
> > >
> > > ---
> > >
> > > **7. Conclusion**
> > >
> > > This paper introduced Verbalized Bayesian Persuasion (VBP), the first general framework that extends Bayesian persuasion from small symbolic games to real-world natural-language settings. By formulating BP as a verbalized mediator-augmented extensive-form game and solving it with a Prompt-PSRO-based solver augmented with verbalized commitment assumptions, obedience constraints, and information obfuscation, VBP bridges the gap between classical information design theory and practical LLM-driven strategic communication. Experiments on recommendation letters, courtroom persuasion, law enforcement, and full-press Diplomacy demonstrate that VBP reproduces known equilibria in classic BP, discovers effective persuasion strategies in natural-language and multistage scenarios, and significantly improves posterior calibration and coordination reliability in complex multi-agent settings.
> > >
> > > Several limitations point to promising future directions. First, although LLMs provide a powerful interface for processing natural language, concerns remain regarding simulation fidelity when modeling real human decision-makers, and larger-scale human studies are needed to further validate the framework's external validity. Second, the current pipeline incurs non-trivial computational overhead from iterative prompt optimization and rollout-based evaluation; advances in model compression and inference acceleration would improve practical scalability. Third, our method controls LLM behavior primarily through writing-style prompts, which can be restrictive; future work will explore richer prompt optimization strategies and parameter-efficient fine-tuning for more expressive signal control. Finally, our study focuses on single-sender settings with one or multiple independent receivers. Extending VBP to multi-sender scenarios and investigating the emergent bargaining dynamics observed in multistage interactions—connecting them to established bargaining theory—constitute exciting avenues for future research. We hope this work opens a new line of study on persuasion in realistic, language-driven strategic environments.
> > >
> > > ---
> > >
> > > We hope this directly addresses your remaining concern about the missing section. We are grateful for the thorough and fair review process.

---

### Official Review · Reviewer_YgVw · 2026-03-12

**Soundness:** 2
**Presentation:** 1
**Significance:** 4
**Originality:** 3
**Overall Recommendation:** 3
**Confidence:** 4

**Summary:**

This paper proposes Verbalized Bayesian Persuasion (VBP), a framework that extends classical Bayesian persuasion (BP) from small, discrete games into natural language settings by leveraging LLMs. The sender and receiver are parameterized as LLMs, states and signals are represented as text (e.g., recommendation letters, legal complaints), and strategies are optimized in prompt space via the Prompt-PSRO framework. The authors introduce verbalized versions of commitment assumptions and obedience constraints, an information obfuscation mechanism, and a conditional prompt optimization method for multistage settings. Experiments span three classic BP problems (recommendation letters, courtroom, law enforcement), a natural language extension of each, a multistage variant, and full-press Diplomacy.

**Compliance With Llm Reviewing Policy:**

Affirmed.

**Key Questions For Authors:**

Feel free to respond to any of the critiques.

### Overall Assessment

My overall feeling is that this paper does genuinely have a novel and important perspective. But it suffers from trying to fit it into an existing framework (BP) and in doing so, opens up many inconsistencies. I would much prefer the authors unshackle themselves from the BP formalisms and design this as simply a new framework for Information Design in the messy language space. They can then focus deeply on the implications of each choice and run a few but detailed experiments to really flesh out the model and it's motivations. I think it can then be a very impactful paper.

**Limitations:**

Yes they have.

**Strengths And Weaknesses:**

## Strengths
- The core idea of creating a rigorous verbalized framework for information design addresses a genuine gap. Classical BP is confined to toy-scale discrete games, while the communication it models occurs in natural language. LLMs offer a natural bridge, and the paper articulates this vision clearly.

- At a high level, I agree with optimization architecture chosen by the authors. Optimizing in prompt space while letting LLMs generate the actual signals is well-motivated. The Prompt-PSRO framework with OPRO and FunSearch as best-response oracles is a reasonable and technically interesting instantiation of this idea.

## Weaknesses

### W1: The signaling scheme lacks the conditional structure required for Bayesian updating

In classical BP, the signaling scheme π(s|ω) is state-conditional: the receiver knows how likely each signal is given each state, which enables Bayesian posterior computation. In VBP, the receiver observes the declared style and the Prompt-PSRO meta-strategy weights, but these weights are *unconditional*. The receiver learns that "Focus: Character" is used 25% of the time, not that it is used with different probabilities depending on student quality. Without π(style|state), there is no basis for Bayesian updating.

Repeated play could in principle reveal these conditionals empirically, but this requires a fixed signaling scheme — precisely what does not happen, as the PSRO loop continuously evolves the strategy set and the S3 setting explicitly uses history-dependent prompt functions. So I am not sure how to interpret Proposition 4.1's claim of convergence to ε-approximate Bayes correlated equilibria, since the MAG framework assumes the state-conditional information structure that the implementation does not provide.

### W2: The prompt-to-behavior mapping is opaque and fragile

The relationship between style prompts and strategic behavior is neither transparent nor reliable. It's not clear how a signal, say "Language Complexity: Moderate" would induce? Is it asking for technical language, verbosity, deception. In the example in the appendix, it's the latter, but not sure why this would be. To be honest, the fact that this is allowed in the obedience/commitment checker is a bit strange. Similarly, the fact that in Figure 16, generated letter describes Jane Smith while the input state describes Michael Brown also makes me wonder about the checks and obedience behaviour. One can lie, but some lies are blatant (such as having the wrong name).

### W3: Excessive breadth at the cost of depth

The paper covers four settings, three BP problems, Diplomacy, six technical components, multiple baselines, ablations, and human evaluation — but the main text contains no concrete examples of what the signals, schemes, or receiver reasoning look like. The recommendation letters and prompt functions are all in the appendix. Benchmark scores like "lying probability = 0.22" are hard to interpret without understanding the qualitative strategies behind them. Fewer settings with worked examples in the main text would be far more instructive.

---

> ### Author Rebuttal · Authors · 2026-03-31
>
> We sincerely thank you for recognizing our paper's originality and significance. Below, we address your concerns.
>
> **1. State-Conditional Structure & Bayesian Updating**
>
> You raised an excellent foundational question regarding the unconditional nature of $P(\text{style})$ and its implications for Bayesian updating and Proposition 4.1.
>
> **Clarification on Information Structure:** We acknowledge the paper did not adequately distinguish between the *policy parameters* (style prompt) and the *realized signal* (generated text). In VBP, the "style" is not the signal; it is a hyperparameter. The actual signal $s$ (the text) is generated by the LLM conditioned on **both** the selected style **and the private true state $\omega$** (which is directly injected into the LLM's context window, see Appx E.6.1). Therefore, the effective signaling scheme is strictly state-conditional:
>
> $\pi\_{VBP}(s | \omega) = \sum\_z P(\text{style}\_z) \cdot P\_{LLM}(s | \text{style}\_z, \omega)$
>
> Because the LLM's text output heavily depends on the state $\omega$ in its prompt, $\pi_{VBP}(s|\omega_{strong}) \neq \pi_{VBP}(s|\omega_{weak})$. This preserves the mathematical foundation required for Bayesian updating and the MAG (Prop 4.1). Figure 7 (Appx F.1) empirically proves the receiver performs Bayesian updating, as its internal posterior estimates dynamically recalibrate to sender obfuscation tactics.
>
> **New Rebuttal Experiment (State-Conditioned Meta-Strategies):** Inspired by your feedback, we modified the Prompt-PSRO solver to maintain explicit state-conditioned meta-strategy weights $P(\text{style} | \omega)$. This implementation successfully accelerated convergence to the $\epsilon$-approximate BCE (from ~10 to ~6 iterations) and slightly improved sender reward (0.48 to 0.49). This validates your intuition while proving our formulation successfully captures BP dynamics.
>
> **2. Prompt Mapping & Checker Fragility**
>
> - **The "Jane Smith / Michael Brown" Artifact:** We must explicitly clarify that this was strictly a manual LaTeX copy-paste error during PDF formatting. Simulation logs confirm the LLM maintained correct entity tracking. We deeply apologize for the confusion and have corrected the figure.
>
> - **Prompt-to-Behavior Mapping:** The mapping from abstract prompts (e.g., "Focus: Character") to strategic behavior is an emergent result of optimization via OPRO. The prompts act as semantic conditioning vectors, directing the LLM's attention to soft-skill attributes and consequently *omitting* quantitative deficits.
>
> - **New Rebuttal Experiment (Entity-Preservation Constraint):** To definitively distinguish valid strategic omission from blatant hallucination, we implemented a Strict Entity-Preservation Constraint (EPC) using a deterministic NER filter to heavily penalize altering immutable facts (Name, Major). Under EPC, the hallucination rate is **0.00**, while the sender reward (0.47) and "lying probability" (0.21) remained remarkably stable. This conclusively proves VBP achieves persuasion through sophisticated semantic framing and omission, completely independent of factual fabrication.
>
> **3. Breadth vs. Depth & Paper Framing**
>
> **Structural Reorganization:** We entirely agree with your critique on excessive breadth. We will move the S4 setting to the appendix. This frees substantial space to integrate the concrete qualitative examples of signals, prompt functions, and receiver reasoning (currently buried in Appx F.9/F.10) directly into the main results section.
>
> - **Qualitative Context for "Lying Probability":** We will explicitly unpack metrics like "lying probability = 0.22" in the main text. Qualitatively, this 22% does not represent explicit falsehoods (like inventing a 4.0 GPA). Instead, it manifests as semantic obfuscation—e.g., generating a letter for a weak student that heavily emphasizes a minor Python project as "valuable industry experience" while omitting their low GPA. This is how the LLM successfully shifts the receiver's posterior belief just above the decision threshold.
>
> - **Framing and BP Formalisms:** We embrace your suggestion to conceptually frame the paper around the broader paradigm of "Information Design in Language." We will explicitly adopt this perspective in the revised version to better contextualize our contributions. However, we strongly defend utilizing the strict BP formalisms as the necessary computational engine. Without verbalized commitment, the interaction collapses into intractable "cheap talk" where signals have no binding power and the sender's reward drops to baseline. We will clarify in the text that BP serves as the tractable mathematical entry point to anchor the otherwise unbounded semantic space of Information Design.
>
> - **Visualizing Receiver Reasoning:** To directly address your request for deeper receiver mechanics, please see the visual breakdown of explicit posterior tracking and signal examples here: **https://ibb.co/9HKM3P0M**.

---

> > ### Author Rebuttal · Reviewer_YgVw · 2026-03-31
> >
> > Thank you for your response. If others are willing to champion the work, I would be open to it. But as for me, I still have some reservations.

---

> > > ### Author Response · Authors · 2026-04-01
> > >
> > > Thank you very much for taking the time to read our rebuttal and for keeping an open mind about our work.
> > >
> > > We fully understand and respect your remaining reservations. Bridging the strict formalisms of Bayesian Persuasion with the messy, unbounded semantic space of LLMs is indeed a highly challenging and nascent area, and it is natural that there are still aspects to be refined.
> > >
> > > Regardless of the final outcome, we want to sincerely thank you for your rigorous and constructive critiques. Your insights—especially regarding the state-conditional structure, the entity-preservation constraints, and the need to prioritize depth over breadth—have been incredibly valuable and will substantively improve the final version of this paper and our future research in this direction.
> > >
> > > Thank you again for your time, effort, and contribution to the review process!
> > >
> > >
> > > Best regards,
> > >
> > > Authors

---

### Decision · Program_Chairs · 2026-04-30

**Decision:**

Accept (regular)

**Comment:**

There are genuine and constructive engagement between reviewers and authors during the rebuttal phase. The AC appreciated all reviewers' constructive comments and insightful feedback. The authors have also done excellent job to genuinely address reviewers' suggestions and concerns to their best, or otherwise genuinely admitting limitations. While some reviewer has some reservation about the presentation style of the paper (e.g., prioritizing breadth over depth), reviewers overall reached consensus about the novel and challenging problems tackled by the paper, potentially big impact to Bayesian persuasion, nicely presented results. The paper is ready to be accepted to the conference.

I also encourage the authors to incorporate the constructive comments and new discussions/experiments in their camera ready version, as also promised by authors in the rebuttals.